**Stratocumulus Cloud Clearings: Statistics from Satellites, Reanalysis Models, and Airborne Measurements**

Hossein Dadashazar[1], Ewan Crosbie[2,3], Mohammad S. Majdi[4], Milad Panahi[5], Mohammad A. Moghaddam[5], Ali Behrangi[5], Michael Brunke[5], Xubin Zeng[5], Haflidi H. Jonsson[6], Armin Sorooshian[1,5*]

[1]Department of Chemical and Environmental Engineering, University of Arizona, Tucson, AZ, USA
[2]Science Systems and Applications, Inc., Hampton, VA, USA
[3]NASA Langley Research Center, Hampton, VA, USA
[4]Department of Electrical and Computer Engineering, University of Arizona, Tucson, AZ, USA
[5]Department of Hydrology and Atmospheric Sciences, University of Arizona, Tucson, AZ, USA
[6]Naval Postgraduate School, Monterey, CA, USA

[*]Corresponding author: armin@email.arizona.edu

Abstract
This study provides a detailed characterization of stratocumulus clearings off the U.S. West
Coast using remote sensing, reanalysis, and airborne in situ data. Ten years (2009-2018) of
Geostationary Operational Environmental Satellite (GOES) imagery data are used to quantify the
monthly frequency, growth rate of total area ($GR_{Area}$), and dimensional characteristics of 306 total
clearings. While there is interannual variability, the summer (winter) months experienced the most
(least) clearing events with the lowest cloud fractions being along coastal topographical features
along the central to northern coast of California including especially just south of Cape Mendocino
and Cape Blanco. From 09:00 to 18:00 (PST), the median length, width, and area of clearings
increased from 680 to 1231 km, 193 to 443 km, and ~67,000 to ~250,000 km$^2$, respectively.
Machine learning was applied to identify the most influential factors governing the $GR_{Area}$ of
clearings between 09:00-12:00 PST, which is the time frame of most rapid clearing expansion.
The results from Gradient Boosted Regression Tree (GBRT) modeling revealed that air
temperature at 850 hPa ($T_{850}$), specific humidity at 950 hPa ($q_{950}$), sea surface temperature ($SST$),
and anomaly in mean sea level pressure ($MSLP_{anom}$) were probably most impactful in enhancing
$GR_{Area}$ using two scoring schemes. Clearings have distinguishing features such as an enhanced
Pacific high shifted more towards northern California, offshore air that is warm and dry, stronger
coastal surface winds, enhanced lower tropospheric static stability, and increased subsidence.
Although clearings are associated obviously with reduced cloud fraction where they reside, the
domain-averaged cloud albedo was actually slightly higher on clearing days as compared to non-
clearing days. To validate speculated processes linking environmental parameters to clearing
growth rates based on satellite and reanalysis data, airborne data from three case flights were
examined. Measurements were compared on both sides of the clear-cloudy border of clearings at
multiple altitudes in the boundary layer and free troposphere, with results helping to support links
suggested by this study's model simulations. More specifically, airborne data revealed the
influence of the coastal low-level jet and extensive horizontal shear at cloud-relevant altitudes that
promoted mixing between clear and cloudy air. Vertical profile data provide support for warm and
dry air in the free troposphere additionally promoting expansion of clearings. Airborne data
revealed greater evidence of sea salt in clouds on clearing days, pointing to a possible role for, or
simply the presence of, this aerosol type in clearing areas coincident with stronger coastal winds.

## 1. Introduction

Stratocumulus clouds play an important role in both global and regional climate systems. Stratocumulus clouds are the dominant cloud type over marine environments based on annual mean of area covered (Warren et al., 1986; Hahn and Warren, 2007). In coastal areas, these clouds can impact industries such as agriculture, transportation (e.g., aviation), military operations, coastal ecology, and biogeochemical cycles of nutrients. Stratocumulus clouds also play an important role in the global radiation budget due to their high albedo contrast with the underlying ocean surface (Hartmann and Short, 1980; Herman et al., 1980; Stephens and Greenwald, 1991). Challenges in accurately simulating the presence and properties of stratocumulus clouds include the difficulty in separating the influence of microphysical and dynamical factors and the existence of multiple feedbacks in cloud systems (Brunke et al., 2019). Therefore, accurate characterization of cloud formation and evolution is critical.

Numerous studies have examined the behavior of clouds off the United States (U.S.) West Coast (e.g., Coakley et al., 2000; Durkee et al., 2000; Stevens et al., 2003; Lu et al. 2009; Painemal and Minnis, 2012; Modini et al., 2015; Sanchez et al., 2016). The persistence of the cloud deck in this region, especially during the summer, makes it a key location for studying marine stratocumulus clouds. Furthermore, the prevalence of freshly-emitted aerosols from ships provides an optimal setting for field measurements of aerosol-cloud-precipitation interactions because of the relative ease of finding strong aerosol perturbations, from which cloud responses can be robustly quantified (e.g., Russell et al., 2013). Over the decades of research conducted in the aforementioned study region and two other major stratocumulus regions (Southeast Pacific Ocean off the Chile-Peru coasts and Southeast Atlantic Ocean off the Namibia-Angola coasts), one feature that has not received sufficient attention is large scale stratocumulus clearings that are easily observed in satellite imagery and often exceed 100 km in width (Fig. 1). Perhaps the most obvious impact of these clearings is the change in albedo as an otherwise cloudy area would be highly reflective. Improving understanding of factors governing clearings has implications for modeling of marine boundary layer clouds and for operational forecasting of weather and fog along coastlines.

Previous studies have documented the existence of large scale cloud clearings off the U.S. West Coast (e.g., Kloesel, 1992). During the 2013 Nucleation in Cloud Experiment (NiCE), three case study flights with the Center for Interdisciplinary Remotely-Piloted Aircraft Studies (CIRPAS) Twin Otter examined clearings off the California coast, with a focus on diurnal behavior and contrasting aerosol and thermodynamic properties across the cloud-clearing interface (Crosbie et al., 2016). Based on a multi-day event, they showed that a clearing expanded during the day and contracted at night towards the coast with oscillations between growth and decay over the multi-day clearing lifetime. They observed that small scale processes (~1 km) at the clearing-cloud border are influential in edge dynamics that likely upscale to more climatologically influential scales, which is why reanalysis data cannot accurately replicate the spatial profile of cloud fraction ($CF$) and cloud liquid water path ($LWP$) when compared to satellite data. One of their three events was associated with a so-called "southerly surge", also referred to as a coastally-trapped disturbance (CTD). CTD events were recently characterized off the U.S. West Coast by Juliano et al. (2019a,b). Clearing events have been examined over the southeast Atlantic Ocean with the catalyst for cloud erosion shown to be atmospheric gravity waves (Yuter et al., 2018). While these aforementioned studies have explained details associated with clearings in different coastal regions, there are many unanswered questions remaining and a need for more statistics associated with clearings to build more robust conclusions.

The goal of this work is to build upon cloud clearing studies over the U.S. West Coast to provide a more comprehensive analysis using the synergy of data from satellite remote sensors, reanalysis products, and airborne in-situ measurements. We first examine a decade of satellite data to report on statistics associated with the temporal and spatial characteristics of clearings. These characteristics are then studied in conjunction with environmental properties from reanalysis products and machine learning simulations to identify factors potentially contributing to the formation and evolution of clearings. Lastly, airborne in situ data are used to validate findings from the aforementioned analyses and to gain more detailed insight into specific events that otherwise would not be possible with reanalysis and satellite products. The most significant implications of our results are linked to modeling of fog and boundary layer clouds, with major implications for a range of societal and environmental issues such as climate, military operations, transportation, and coastal ecology.

## 2. Experimental Methods
### 2.1 Satellite Datasets

Long-term statistics associated with clearings were obtained using Geostationary Operational Environmental Satellite (GOES) visible band (~0.6 μm) images. Visual imagery data were obtained from GOES-11 for 2009 through 2011 and from GOES-15 between 2012 and 2018 (data products summarized in Table 1). Images were analyzed for the spatial domain bounded by 115°-135° W and 30°-50° N. The following steps led to the identification of individual clearings using GOES images, of which a total of 306 were identified between 2009 and 2018:

(i) GOES-11 and GOES-15 visible images were obtained from the National Oceanic and Atmospheric Administration (NOAA) Comprehensive Large Array-data Stewardship System (CLASS) database (http://www.class.noaa.gov).

(ii) Each day's sequence of GOES images were visually inspected to identify if a clearing event was present. This involved utilizing the following general guidelines: (i) there had to be sufficient cloud surrounding the clearing area that the clearing's borders could be approximately identified, which excluded cases with highly broken cloud deck; (ii) clearings that were not connected to land between 30°-50° N in any of daily images were excluded; (iii) days with the cloud deck completely detached from the coast between 30°-50° N were not considered; and (iv) only clearings with a maximum daily area of greater than 15,000 km$^2$ (which translates to a clearing length on the order of 100 km) were considered. Consequently, the statistics presented in Section 3.1.1 represent a lower limit of clearing occurrence in the study region. However, it is expected that the qualitative trends discussed in Section 3.1.1 are representative of clearing behavior in the study region.

(iii) For each clearing event, four images were selected to both quantify clearing properties and characterize diurnal variability: (i) Image 1 after sunrise, between 14:15 UTC (7:15 Pacific Standard Time (PST)) and 16:45 UTC (09:45 PST) with a median at ~16:00 UTC (09:00 PST); (ii) Image 2 at a time relevant to the Moderate Resolution Imaging Spectroradiometer (MODIS) Terra overpass over the study region, between 18:45 UTC (11:45 PST) and 20:45 UTC (13:45 PST) with a median at ~19:00 UTC (~12:00 PST); (iii) Image 3 at a time relevant to the MODIS Aqua overpass over the study region, ranging from 19:45 UTC (12:45 PST) to 22:15 UTC (15:15 PST) with a median at ~22:00 UTC (~15:00 PST); and (iv) Image 4 before sunset, ranging from 22:45 UTC (15:45 PST) to

02:15 UTC (19:15 PST) with a median at ~01:00 UTC (~18:00 PST). For the purposes of
subsequent discussion, local times (PST) will be used.
(iv)   A custom-made cloud mask algorithm was applied consisting of the following steps: (i)
each visible image was converted to an 8-bit integer gray-scale image with values assigned
to each pixel ranging from 0 (black) to 255 (white); (ii) continental areas were masked
from the analysis (i.e., green regions in Fig. 1), meaning that their values were not included
in subsequent steps; (iii) a histogram of values for all pixels over the ocean was calculated
for each image obtained in the previous step and then Otsu's method (Otsu 1979) was
applied on the obtained histogram to compute a global threshold to categorize each pixel
as either clear or cloudy; (iv) a MATLAB image processing toolbox was used to extract
the clearing as an object, including the pixels at the clearing-cloud border and pixels inside
the clearing; (v) information contained within the clear pixels was then used to estimate
clearing dimensions such as width, length, area, and centroid for the spatial domain
bordered by 115°-135° W and 30°-50° N; and (vi) a MATLAB application was written to
automate all of the aforementioned steps to process data for a decade (2009-2018).

Data were used from the MODIS on the Terra and Aqua satellites to characterize cloud
properties on clearing and non-clearing days in the spatial domain of analysis defined above. Daily
Level 3 data (Hubanks et al., 2019) with spatial resolution 1°×1° were downloaded from the
LAADS DAAC distribution system (https://ladsweb.modaps.eosdis.nasa.gov/). The key daytime
parameters (Table 1) retrieved for this study relevant to liquid clouds included the following, which
were retrieved at 2.1 μm and selected based on their importance for marine boundary layer (MBL)
cloud studies: *CF* obtained from the MODIS cloud mask algorithm (Platnick et al., 2003), cloud
optical thickness ($\tau$), *LWP*, and cloud droplet effective radius ($r_e$). Detailed information about these
MODIS products is described elsewhere (Platnick et al., 2003; Platnick et al., 2017; Hubanks et
al., 2019).
Although MODIS Level 3 data parameters do not include cloud droplet number
concentration ($N_d$), previous studies estimated $N_d$ using retrievals of $\tau$ and $r_e$ with assumptions
(Bennartz, 2007; Painemal and Zuidema, 2010; McCoy et al., 2017). We use the following
equation from Painemal and Zuidema (2010) to estimate $N_d$:
$N_d = \frac{(\Gamma_{ad})^{\frac{1}{2}}}{k} \frac{10^{\frac{1}{2}}}{4\pi\rho_w^{\frac{1}{2}}} \frac{\tau^{\frac{1}{2}}}{r_e^{\frac{5}{2}}}$                            (1)
where $\rho_w$ is the density of liquid water, $\Gamma_{ad}$ is the adiabatic lapse rate of liquid water content
(*LWC*), and the parameter *k* is representative of droplet spectral shape as the cube of the ratio
between the volume mean radius and the effective radius. $\Gamma_{ad}$ is a function of temperature and
pressure (Albrecht et al., 1990). In this study, cloud top temperature and pressure, provided by
MODIS, are used to estimate $\Gamma_{ad}$ following the methodology described in Braun et al. (2018). A
constant value of 0.8 (Martin et al. 1994) is assigned to *k* in Equation 1. Similar to our previous
study on clearings (Crosbie et al., 2016), cloud top albedo (*A*) was quantified using $\tau$ in the
following relationship (Lacis and Hansen 1974):
$A = \frac{\tau}{\tau + 7.7}$                                            (2)

## 2.2 Reanalysis Data

Various products from Modern-Era Retrospective analysis for Research and Applications, Version 2 (MERRA-2; Gelaro et al., 2017) were used to gain insight into possible mechanisms influencing the formation and evolution of clearings off the U.S. West Coast. MERRA-2 data were downloaded from the NASA Goddard Earth Sciences Data and Information Services Center (GES DISC; https://disc.gsfc.nasa.gov/). Table 1 summarizes MERRA-2 parameters used in this work, including detailed information such as their product identifier and temporal resolution. The parameters were chosen based on their ability to provide a sufficient view of atmospheric conditions in which MBL clouds form, evolve, and dissipate. Various vertical levels were used for some MERRA-2 products as a way of obtaining representative information for different layers of the MBL and free troposphere (FT). Of note is that the MERRA-2 aerosol reanalysis relies on the GEOS-5 Goddard Aerosol Assimilation System (Buchard et al., 2015) for which the Goddard Chemistry, Aerosol, Radiation, and Transport (GOCART) model (Chin et al., 2002) simulates 15 externally mixed aerosol tracers including sulfate, dust (five size bins), sea salt (five size bins), and hydrophobic and hydrophilic black carbon and organic carbon. Of relevance to this study, GOCART applies wind-speed dependent emissions for sea salt. Furthermore, the dominant removal mechanisms for aerosols include gravitational settling, dry deposition, and wet scavenging.

## 2.3 Airborne In-Situ Data

Motivated by the three case study research flights (RFs) probing clearings during the NiCE campaign (Crosbie et al., 2016), the Fog and Stratocumulus Evolution Experiment (FASE) was carried out with nearly the same payload on the Center for Interdisciplinary Remotely-Piloted Aircraft Studies (CIRPAS) Twin Otter between July and August 2016 (Sorooshian et al., 2018). Data were used from three case RFs examining clearings: RF08 on 2 August 2016, and RF09A/RF09B on 3 August 2016. The back-to-back flights on 3 August afforded an opportunity to examine the evolution of clearing properties at the clear-cloudy interface over a span of a few hours. Figure 2 shows GOES imagery and the flight pattern for RF09A, which is representative of the other two shown in Figs. S1-S2. The same flight strategy from NiCE (Crosbie et al., 2016) was used in the FASE RFs and included the following set of maneuvers (Fig. 2c): (i) spiral profiles on both sides of the clear-cloudy interface; (ii) level legs extending on both sides of the clear-cloudy interface near the ocean surface (~30 m; called "surface leg"), above cloud base, and mid-cloud; (iii) a series of sawtooth maneuvers up and down between ~60 m below and above the cloud top on both sides of the clear-cloudy interface; and a (iv) level leg in the FT at ~1 km altitude. The typical aircraft speed was 55 m s$^{-1}$.

Commonly used instruments provided dynamic, thermodynamic, and navigational data (Crosbie et al., 2016; Dadashazar et al., 2017; Sorooshian et al., 2018). Of relevance to this study are 10 Hz measurements of wind speeds, air temperature, and humidity. Setra pressure transducers attached to a five-hole gust probe radome provided three components of wind speeds after correction for aircraft motion, which was obtained by a C-MIGITS-III GPS/INS system. Ambient air temperature was measured by a Rosemount Model 102 total temperature sensor. Also, humidity data were collected with an EdgeTech Vigilant chilled mirror hygrometer (EdgeTech Instruments, Inc.).

Cloud micro/macrophysical parameters were measured at 1 Hz with various instruments. Size distributions of cloud droplets and rain droplets were characterized using the Forward

Scattering Spectrometer Probe (FSSP; $D_p \sim$ 2-45 μm) and Cloud Imaging Probe (CIP; $D_p \sim$ 25-1600 μm). Cloud base rain rate was quantified using the size distributions of drizzle drop ($D_P > 40$ μm) obtained from CIP in the bottom third of clouds along with documented relationships between fall velocity and drop size (Wood 2005a). *LWC* data were obtained using a PVM-100 (Gerber et al., 1994), which were vertically integrated during sounding profiles to quantify cloud *LWP*. Aerosol concentration data are reported here from the passive cavity aerosol spectrometer probe (PCASP; $D_p \sim$ 0.11–3.4 μm; Particle Measuring Systems (PMS), Inc.; modified by Droplet Measurement Technologies, Inc.) at 1 Hz time resolution. Cloud water composition data were obtained using a modified Mohnen slotted-rod collector (Hegg & Hobbs, 1986) that was manually placed out of the aircraft during cloud passes to collect cloud water. The collected samples were analyzed for water-soluble ions using ion chromatography (IC; Thermo Scientific Dionex ICS-2100 system) and water-soluble elements using triple quadrupole inductively coupled plasma mass spectrometry (ICP-QQQ; Agilent 8800 Series). Liquid-phase concentrations of species were converted to air-equivalent units (µg m$^{-3}$) via multiplication with the sample-averaged *LWC*. The reader is referred to other works for more extensive discussion about cloud water collection and sample analysis from FASE and other recent CIRPAS Twin Otter campaigns (Crosbie et al., 2018; Prabhakar et al., 2014; Sorooshian et al., 2013a; Wang et al., 2016; Youn et al., 2015).

Ten Hz measurements of environmental parameters were used to estimate turbulent variance and covariance flux values, which may be relevant to the understanding of clearing formation and evolution based on past work (Crosbie et al., 2016). To perform the aforementioned calculations, collected data for wind speed and temperature were de-trended using a 2-km wide high pass filter that utilizes a minimum order-filter with a stopband attenuation of 60 dB and transition band steepness of 0.95. Friction velocity ($u^*$) was calculated from the surface leg following the method provided in Stull (1988) and Wood (2005b). In addition, convective velocity ($w^*$) was estimated by implementing the buoyancy integral method (Nicholls and Leighton, 1986). Turbulent kinetic energy (*TKE*) in the MBL is generated by two main mechanisms, specifically shear and buoyancy generation. Following Wood (2005b), the ratio of the MBL depth ($z_i$) to the Monin–Obukhov length ($L_{MO}$) was estimated as a way to determine the relative influence of shear versus buoyancy in values of *TKE*. Large positive values of the ratio ($-z_i/L_{MO}$) are associated with the turbulence in the MBL governed more with buoyancy production, while small or negative values are associated with the dominance of shear production.

Properties relevant to the inversion layer were estimated from sawtooth maneuvers above and below the cloud top, which typically coincided with the inversion base altitude (Fig. 2c). The inversion base height was defined as the altitude where the ambient temperature first reached its minimum above the sea surface (Crosbie et al., 2016). Inversion top was defined as the highest altitude at which $d\theta_l/dz$ exceeded 0.1 K m$^{-1}$, where $\theta_l$ is liquid water potential temperature and $z$ is altitude. $d\theta_l/dz$ was calculated from linear fits over a moving window of 75 points from 10 Hz data. The following characteristics were estimated and reported for the inversion layer: (i) inversion base height; (ii) inversion top height; (iii) inversion depth; (iv) jump in liquid water temperature ($\Delta\theta_l$); (v) maximum gradient of the potential temperature (($d\theta_l/dz$)$_{max}$); (vi) drop in the total moisture ($\Delta q_t$); and (vii) change in the horizontal wind speed ($\Delta U$).

## 2.4 Clearing Growth Modeling Using Machine Learning

A Gradient Boosted Regression Tree (GBRT) model approach was implemented to investigate the impact of environmental parameters on the evolution of clearing events (Friedman 2001). GBRT models have been successfully used in past work to study low-level clouds (Fuchs

et al., 2018). The Scikit-Learn library (Pedregosa et al., 2011) was used for careful parameter tuning in order to accurately represent the data and desired relationships without overfitting the model (Fuchs et al., 2018).

We apply the GBRT model to analyze clearing growth rates of total area ($GR_{Area}$) obtained from the comparative analysis between GOES Image 1 (~9:00 PST) and Image 2 (~12:00 PST) for each of the 306 events. As will be shown, the most rapid clearing growth occurs between 9:00 and 12:00 PST among the three time increments between Images 1-4 (i.e., 09:00 - 18:00 PST). Here we describe how the predictor values were obtained. A rectangular box was placed around the larger of the clearing areas from Image 1 or 2 for each clearing event using the maximum and minimum values of both latitude and longitude. The same size rectangular box was then placed on the other image using identical latitude and longitude bounds. MERRA-2 data were then obtained for each $0.5° \times 0.625°$ grid within the rectangular area for the two images, and then averaged for the pair of images. Each grid was also assigned the value of the clearing $GR_{Area}$ for the entire clearing (i.e., each grid had the same value of $GR_{Area}$ assigned to it). Parameters used in the modeling included those relevant to aerosol (aerosol optical depth ($AOD$)), thermodynamics (air temperature ($T$), air specific humidity ($q$), and sea-surface temperature ($SST$), and dynamic variables (mean sea level pressure anomaly ($MSLP_{anom}$), zonal wind speed ($U$), meridional wind speed ($V$), planetary boundary layer height ($PBLH$), and vertical pressure velocity ($\omega$)). Most of the aforementioned variables were first analyzed at different vertical levels including the surface, 950 hPa, 850 hPa, and 700 hPa in order to then filter variables out to keep only the most appropriate input parameters.

Model simulation results are reported in terms of a parameter termed 'partial dependence' ($PD$) following methods in earlier works (e.g., Friedman, 2001; Fuchs et al., 2018). $PD$ plots represent the change of the clearing $GR_{Area}$ relative to a selected parameter by marginalizing over the remaining predictors. For each given value of a selected parameter ($x_s$), partial dependence ($PD(x_s)$) can be obtained by computing the average of model outputs using the training data as shown in Equation 3:

$$PD(x_s) = \frac{1}{n}\sum_{i=1}^{n} \hat{f}(x_s, x_R^{(i)}) \tag{3}$$

where $\hat{f}$ is the machine learning model, $x_R$ are the remaining parameters, and $n$ is the number of instances in the training data. $PD$ profiles were computed between the 1st and 99th percentile of each selected parameter.

While $PD$ plots are not flawless in capturing the influence of each variable in the model, especially if the input variables are strongly correlated, they provide useful information for interpretation of GBRT results (Friedman and Meulman 2003; Elith et al., 2008). To decrease the undesired influence of correlated variables on $PD$ profiles, an arbitrary $r^2$ threshold of 0.5 was used based on the linear regressions between prospective input parameters. For instance, there were three choices of air temperature (i.e., at 950, 850, and 700 hPa), but based on the $r^2$ criterion, only one ($T_{850}$) was used in the model to minimize the unwanted impact of dependent input parameters. Lower tropospheric stability ($LTS$: defined as the difference between the potential temperature of the FT (700 hPa) and the surface) is the stability parameter that has been widely used as a key factor controlling the coverage of stratocumulus clouds. However, in this study, the effects of stability were examined by putting $T_{850}$ and $SST$ into the model without explicitly including $LTS$. The correlation between $LTS$ and $T_{850}$ prevented them to be used as input parameters simultaneously. Using $T_{850}$ and $SST$ instead of $LTS$ is advantageous because the results can be more informative by revealing different impacts of the two individual parameters on the model's output rather than just one parameter in the form of $LTS$. In addition, the mean sea level

pressure anomaly ($MSLP_{anom}$) was used as an input parameter, which was calculated in reference to the average values of $MSLP$ for the summer months for the study period. In the end, the following 11 predicting variables from MERRA-2 were used as input parameters for the GBRT simulations, with data product details summarized in Table 1: $AOD$, $T_{850}$, $q_{950}$, $q_{850}$, $q_{700}$, $SST$, $MSLP_{anom}$, $U_{850}$, $V_{850}$, $PBLH$, and $\omega_{700}$. It is important to note that the results of extensive sensitivity tests led to the selection of the set of parameters presented in this study. Also, these sensitivity tests confirmed that the general conclusions presented here were preserved regardless of using different sets of the input parameters.

To train, test, and validate the statistical models, the dataset was split into random parts. The training set was comprised of 75% of the data points, 30% of which were randomly selected for validation. This process helped reduce variance and increase model robustness. The remaining 25% of the data points comprised the test dataset. The model setup was tuned using training data, for which different scenarios were tested that were specified by a parameter grid through a 10-fold cross-validated search. The model was run on the dataset 30 times to achieve robust results. To qualitatively rank the input parameters based on their influence on growth rates, two scoring metrics were calculated over 30 runs: (i) differences between the maximum and minimum of $PD$ ($\Delta PD$); and (ii) the relative feature importance following the method developed by Friedman (2001), which is determined by the frequency that a variable is chosen for splitting, weighted by the gained improvement due to each split and averaged over all trees (Friedman and Meulman 2003; Elith et al., 2008).

**3. Results and Discussion**

**3.1 Temporal and Spatial Profile of Clearings**
**3.1.1 Monthly and Interannual Trends**

The frequency of clearing events was quantified for the three summer months (June – July – August, JJA) of each year from 2009 through 2018 (Fig. 3a). Note that if a clearing event lasted multiple days as in the case of the 11-day clearing probed by Crosbie et al. (2016), it was counted separately for each individual day rather than assigned a value of one for a multi-day period. There was considerable interannual variability, with clearing events ranging between a minimum of 14 in 2017 and a maximum of 45 in 2011. The relative percentage of total days in the summer season having clearings ranged from 15.2% – 48.9% with a mean ± standard deviation of 33.3 ± 10.9 days. The specific month with the most clearing events varied between years, with August typically having the least number of events among the summer months. The most recent year of the decade examined, 2018, was used to more closely examine the distribution of clearing events as a function of all 12 months. Daily probabilities of clearing events are shown for each month, with the highest probability between May and September (> 0.2), especially June (~0.42) (Fig. 3b). Daily probabilities were lowest in the winter season, with January having no clearings.

To identify if the monthly profile of clearings is biased by the monthly profile of $CF$, Figs. S3-S4 show the mean annual cycle of MODIS $CF$ for 2018 and 2009-2018, respectively. The range in $CF$s for 2018 and 2009-2018 were 0.59-0.76 and 0.60-0.74, respectively, with the mean values being 0.69 ± 0.05 and 0.68 ± 0.04. This is indicative of relatively low variability. A reasonable question is if August had the lowest clearing daily probability of the summer months because it potentially had the lowest $CF$. Figs. S3-S4 do not show significant variations in $CF$ between the summer months, with mean values in 2018 for June, July, and August being 0.71, 0.72, and 0.72, respectively. Also, the lowest mean daily probability in 2018 was for January and February, but those months do not exhibit the lowest $CF$ (January = 0.76, February = 0.67). Rather, September

exhibited the lowest *CF* (0.59). Finally, *CF* decreased from 0.72 to 0.59 from August to September
2018, but the daily probability of clearings actually increased slightly. Thus, the systematic
changes in *CF* between months are not the primary cause for inter-monthly variation in clearing
formation.
**3.1.2 Diurnal**
Dimensional characteristics of cloud clearings as a function of time of day are summarized
here. The median width of clearings was smallest in the morning at 09:00 (193 km), with an
increase between 09:00 and 12:00, and then a leveling off in expansion until 18:00 (443 km) (Fig.
4). Clearing length and area followed the same qualitative trend in growth with an initial increase
and then leveling off.  The median length and area of clearings at 09:00 were 680 km and ~67,000
$km^2$, respectively, with values at 18:00 being ~1231 km and ~250,000 $km^2$. The aspect ratio
(width:length) was of interest to quantify how long such clearings are relative to their width
throughout the day, with results indicating a minor increase that was more linear than asymptotic
(from ~0.32 at 09:00 to ~0.37 at 18:00). Although the range in median values was very small, there
was significant variability at each of the four time steps shown. Figure S5 quantifies the *GR* of
total area, width, and length by comparing 12:00 to 09:00, 15:00 to 12:00, and 18:00 to 15:00. The
*GRs* for clearing length, width, and area are expectedly lowest from 15:00 to 18:00 and highest
from 09:00 to 12:00.
Figure 5 shows *CF* maps for the times corresponding to panels 1 – 4 for all 306 events
between 2009 and 2018. The spatial maps show that the centroid of the clearings is generally
focused on the coastal topographical features along the central to the northern coast of California
including especially just south of Cape Mendocino and Cape Blanco. Less pronounced is a centroid
of reduced *CF* by Point Conception, where similar mechanisms may be at work. The 09:00 map
most clearly shows that those two topographical features potentially serve as 'trigger points' for
the majority of clearings, and as a typical clearing day develops, the *CF* gets reduced around those
points by moving farther south and to the west. The significance of these capes is discussed in
many previous studies (Beardsley et al., 1987; Haack et al., 2001; Juliano et al., 2019a,b) pointing
to their ability to alter local dynamics, cloud depth, and various microphysical processes such as
entrainment. Cloud thinning in the vicinity of the capes due to an expansion fan effect is reported
for both northerly and southerly flow (Beardsley et al., 1987; Juliano et al., 2017).
**3.2 Contrasting Clearing and Non-Clearing Cases**
Large-scale dynamic and thermodynamic characteristics were contrasted (parameters in
Table 1) between clearing and non-clearing days (Fig. 6). Sub-daily data were averaged up to daily
resolution for parameters of interest, which were subsequently used to produce a climatology for
non-clearing (614 days) and clearing (306 days) cases for the summers between 2009 and 2018. It
is important to note that non-clearing cases include those summer days (e.g., June, July, and
August) from 2009 through 2018 that were not categorized as clearing days. We further calculated
the difference between clearing and non-clearing conditions.
The Pacific high usually sets up ~1000 km west of California during the summertime,
which promotes northerly flow near the surface along the coastline (e.g., Juliano et al., 2019a). As
compared to non-clearing cases, clearing days are characterized by having an enhanced Pacific
high shifted more towards northern California (Fig. 6a). The presence of Pacific high over the
ocean and thermal low over the land, especially for the summer months, are the main synoptic
components contributing to the formation of coastal low-level jets (CLLJs) along the California

coast (Beardsley et al., 1987; Parish 2000). California CLLJs are characterized by vertically narrow regions of intensified coast-parallel winds in low altitudes near the MBL top (Burk and Thompson 1996) with an average strength of ~15 m s$^{-1}$ (Lima et al., 2018). In contrast, CLLJs have a relatively large horizontal offshore extent of up to a couple of hundred kms, which is determined by the Rossby radius of deformation (Ranjha et al., 2013). In both cases (clearing and non-clearing), the cross-coast gradient in *MSLP* and 850 hPa geopotential height gradients are the highest in northern California and directed away from the coast. Due to the displacement of the Pacific high towards the northeast part of the study region on clearing days, these gradients are much more profound on clearing days as compared to non-clearing days. The zonal pressure gradient is the main parameter controlling the intensity and occurrence of California CLLJs (Zemba and Friehe 1987; Parish 2000; Lima et al., 2018). The probability of CLLJ incidents is most likely greater on clearing days as a response to the enhanced pressure gradients near the coast. This is also supported by low level wind fields shown in Fig. 7, which exhibit a 2-5 m s$^{-1}$ increase in northerly surface wind speed (Fig. 7a) between 35°N and 45°N. Looking at the 850 hPa wind field (Fig. 7b), there is also a ~2-5 m s$^{-1}$ increase in wind speed but in this case more in a northeasterly direction, which equates to having offshore flow from the northern California coast. The tightening of the 850 hPa geopotential height gradient on clearing days results in strong offshore flows by Cape Blanco and Cape Mendocino (Fig. 7b) where *CF* minima are observed (Fig. 5). In addition, Beardsley et al. (1987) reported periods of low cloudiness along the California coast as a response to the synoptic scale features, an increase in the pressure gradient along the coast, and enhanced wind speeds. In other studies, over the southeast Pacific (Garreaud and Munoz 2005; Zuidema et al., 2009), dissipation of the coastal stratocumulus cloud deck was observed over the jet regions. Average conditions at 500 hPa indicate mostly westerly flow on both clearing and non-clearing days. Non-clearing days exhibited a weak trough offshore, while during clearing days a ridge is present at 500 hPa farther offshore. Displacement and strengthening of the high-pressure system on clearing days can be associated with the passage of mid-latitude ridges (Garreaud and Munoz 2005).

The difference in air temperature between clearing and non-clearing cases at the surface reaches up to ~0.7 K on the western edge of the study domain (Fig. 6a). Clearing cases exhibited cooler temperatures closer to the coast where the clearings develop and evolve. *SST* shows a similar pattern as air temperature at the surface (Fig. 8a). Faster offshore winds at the surface can promote ocean upwelling and thus cooler *SST*s (Lima et al., 2018), as was also observed for CTD events in the same region (Juliano et al., 2019a). Furthermore, the generally high *CF*s during clearing days for the entire spatial domain reduces radiative transfer to the ocean, also acting to reduce *SST* over the broader study region. Cloudiness and surface winds play a major role in influencing *SST*s (e.g., Klein et al., 1995). In contrast, air temperatures at higher levels (850 and 500 hPa) are enhanced adjacent to the coastline in clearing cases. Air temperature at 850 hPa is higher (lower) to the south (north) of Cape Blanco and Cape Mendocino (Fig. 5) in clearing cases as compared to non-clearing cases, with the difference reaching as high as ~2 K. The enhanced offshore flow of warm and dry air in in the vicinity of Cape Blanco and Cape Mendocino likely contributes to why many of the clearings geographically are centered by these coastal topographical features (Fig. 5). It is noteworthy that over the west coast of subtropical South America, cloud dissipation over and upstream of the coastal jet region was reported (Garreaud and Munoz 2005; Zuidema et al., 2009), whereas downstream there was enhanced *CF*, which appears to be analogous to this study.

The changes in synoptic-scale conditions, including relocation/strengthening of the Pacific high, on clearing days in comparison to non-clearing days can alter large-scale subsidence. This is

indeed confirmed in Fig. 8b using $\omega_{700}$ as the proxy variable, with the strongest difference between
clearing and non-clearing days (up to ~ 0.1 Pa s$^{-1}$) off the coast by Cape Blanco and Cape
Mendocino and geographically coincident with where the sharpest gradients occur for *MSLP*
between clearing and non-clearing cases (Fig. 6a). It is interesting to note that the maximum *LTS*
values coincide spatially with enhanced values of $\omega_{700}$ on non-clearing days, in contrast to clearing
days when the peak value of $\omega_{700}$ is farther north from where *LTS* peaks (Fig. 8c). Consistent with
the results presented here (Fig. 8b), modeling studies (Burk and Thompson 1996; Munoz and
Garreaud 2005) reported enhanced subsidence for the entrance regions of the Chilean and
California CLLJs in response to coastal features. These studies also reported the generation of a
warm layer above the MBL due to coastal mechanisms especially downstream of coastal points
and capes. This is also the case in this study where higher air temperature at 850 hPa was observed
to the south of Cape Blanco and Cape Mendocino on clearing days (Fig. 6b). In addition, higher
*LTS* values on clearing days by up to ~2 K (Fig. 8c) are largely associated with the presence of
warmer layer above the MBL south of Cape Blanco and Cape Mendocino. It is likely that reduced
*SST*s and greater subsidence contributed to generally higher *LTS* on clearing days versus non-
clearing days (Fig. 8c). Other works have pointed to the connection between cooler *SST*s, higher
boundary layer cloud amount, and increased stability in the lower atmosphere (Klein and Hartman
1993; Norris and Leovy 1994).
Another key environmental parameter related to MBL cloud coverage is the *PBLH*.
Consistent with previous studies (Neiburger et al., 1961; Wood and Bretherton 2004), regardless
of whether clearings were present, *PBLH* generally increases with distance from the coast (Fig.
8d), where warmer *SSTs* lead to deeper MBLs by weakening the inversion (Bretherton and Wyant
1997). The shallowing of the MBL near the California coast is also notable with enhanced
gradients on clearing days. The aforementioned MBL shallowing is believed to be a crucial
element in development of the coastal jet off the California coast (Zemba and Friehe 1987; Parish
2000). Previous studies (Beardsley et al., 1987; Edwards et al., 2001; Parish 2000; Zuidema et al.,
2009) also reported MBL height adjustment in the vicinity of coast due to hydraulic adaptation to
coastal topography, thermally driven circulation, and geostrophic adjustment in the cross-coast
direction in response to the contrast in surface heating between ocean and land. There is also a
strong gradient in *PBLH* along the shoreline in the vicinity of Cape Blanco (Fig. 8d). While the
presence of a similar gradient in *SST* (Fig. 8a) may partly explain the observed gradient in *PBLH*,
coastally induced processes could also play a role.
Comparing clearing with non-clearing days, *PBLH* tends to be higher on clearing days,
with the largest differences (~200 m) observed to the north off the coasts of Washington and British
Columbia, which re-emphasizes the important role of coastal topography near Cape Blanco and
Cape Mendocino in mesoscale dynamics (Beardsley et al., 1987; Haack et al., 2001). Zuidema et
al. (2009) suggested that dynamical blocking of the surface winds by the southern Peruvian Andes
contributed to boundary layer thickening by encouraging mesoscale convergence. Enhanced
dynamical blocking of surface winds by coastal topography near Cape Blanco, as suggested by
greater wind speeds on clearing days (Fig. 7a), can lead to a deeper MBL in the coastal regions
north and northwest of Cape Blanco. In contrast, coastal areas south of Cape Blanco, exhibit
negligible differences in *PBLH* between clearing and non-clearing days. In the aforementioned
regions, enhanced hydraulic response (i.e., expansion fan (Parish et al., 2016)) to coastal
topography, may cause slightly shallower MBL on clearing days.

Higher MBL depths in the offshore regions of clearing days is noteworthy to discuss. Parameters influencing MBL depth include entrainment rates, vertical velocity at the top of MBL, and horizontal advection of MBL (Wood and Bretherton 2004; Rahn and Garreaud 2010). Although on clearing days there may be greater subsidence rates offshore (Fig. 8b) promoting a shallower MBL, the sum of entrainment and horizontal advection terms counteract the aforementioned effect resulting in a deeper MBL. Wood and Bretherton (2004) showed for the Northeast and Southeast Pacific that entrainment and subsidence were the most influential terms in the MBL prognostic equation, which acted in the opposite manner. It is also likely that entrainment processes resulting from changes in small scale turbulence contributed to elevated *PBLH* on clearing days (Randall 1984; Rahn and Garreaud 2010). The maps of *CF* from MODIS Terra (Fig. 9a) can provide at least one possible explanation for the spatial differences in *PBLH* between clearing and non-clearing days. Cloud fraction is generally higher for the broad study region on clearing days, which leads to more opportunity for cloud top radiative cooling to then fuel turbulence in MBL (Wood 2012). Greater turbulence can lead to a deeper MBL by promoting greater entrainment at the top of MBL (Randall 1984; Wood 2007).

Figure 8e shows spatial maps of specific humidity at 10 m above the sea surface ($q_{10m}$), which serves as a proxy of available moisture in MBL. Assuming a shallow and well-mixed MBL, $q_{10m}$ represents moisture levels in the MBL. Similar to *SST*, $q_{10m}$ increases to the south of the study region with especially reduced values immediately adjacent to the California coast. Comparing clearing and non-clearing days, the former is less humid in the MBL (up to -0.6 g kg$^{-1}$). This is at least partly attributed to offshore flow and entrainment of dry continental air. Specific humidity was also examined at 850 hPa, which is closer to the vertical layer more relevant to air impacting cloud top close to the coastline. Figure 8f shows that $q_{850}$ was substantially lower (up to ~-1.2 g kg$^{-1}$) in the clearing cases, especially in the regions where most of the clearings occur. Drier air above cloud top will decrease cloudiness through entrainment processes. It is interesting to note that the area of greatest $q_{850}$ difference (Fig. 8f) corresponds to the area of greatest northeasterly winds in the difference plot of the wind field at 850 hPa (Fig. 7b). These pieces of evidence point to the role of dry continental air in contributing to the formation and sustenance of clearings via offshore flow.

Another important parameter influencing MBL clouds is nuclei of the cloud droplets, specifically the cloud condensation nuclei (*CCN*). *CCN* in the region originate from a blend of sources, including natural ones (sea spray, marine and continental biogenic emissions, terrestrial dust), biomass burning, ship exhaust, and continental anthropogenic sources (Hegg et al., 2010; Coggon et al., 2014; Wang et al., 2014; Maudlin et al., 2015; Mardi et al., 2018). As a representation of the general level of aerosol pollution in the region, spatial maps are shown for Aerosol Optical Depth (*AOD*), which is a columnar measurement of aerosol extinction (Fig. 8g). In general, regions closer to the shore exhibit higher values of *AOD* on non-clearing days, with especially higher levels north of 40° N. It is unclear as to why this is, since stronger winds on clearing days along the coast have the potential for more emissions from marine biogenic sources (via upwelling), sea spray, and offshore continental flow. Although based on speculation, one of many possible explanations could be that stronger fluxes of sea spray on clearing days have the potential to expedite the drizzle formation process in polluted clouds via broadening of cloud droplet size distributions, which leads to wet scavenging of aerosols in the study region (Dadashazar et al., 2017; Jung et al., 2015; MacDonald et al., 2018; Sorooshian et al., 2013b). South of Cape Blanco and Cape Mendocino on clearing days, there were pockets of high *AOD*

relative to other coastal locations, which is presumed to be linked to stronger winds and offshore continental flow; this is analogous to how CTD events exhibit more pollution north of these coastal features when there is southerly flow (Juliano et al., 2019a). That the greatest *AOD* differences occur close to the coast warrants additional research as such differences may be suggestive of variations in ocean-land-atmosphere interactions that result from the movement and strengthening of the Pacific high during clearing events. Future work should examine if such *AOD* differences on clearing versus non-clearing days are linked to differences in MBL sources and sinks (i.e., wet scavenging), or FT processes.

Spatial maps of cloud microphysical variables provide consensus that clearing days generally have higher $N_d$ and reduced values of $r_e$, $\tau$, and *LWP* near the California coast where clearings form and evolve (Fig. 9). Figure S6 shows the same qualitative results based on MODIS Aqua data for cloud microphysical parameters. Lower *LWP* values on clearing days near the coast are consistent with offshore flow of dry and warm air eroding clouds. The combination of higher $N_d$ and lower *LWP* by the coastline results in smaller $r_e$ on clearing days. The more polluted clouds along the coastline during clearing days, especially south of major capes, is analogous to CTD clouds being more polluted during southerly wind regimes in the study region (Juliano et al., 2019a,b). An intriguing aspect of clearing days was that although a significant section of the study region was cloud-free, the mean cloud albedo (*A*) over the entire study domain was actually slightly higher than on non-clearing days (Fig. 9f). More specifically, the domain-averaged *A* values based on MODIS Terra data (and using Eq. 2) were 0.50 and 0.53 for non-clearing and clearing cases, respectively. The corresponding values using MODIS Aqua data were 0.48 and 0.50, respectively. It is possible that the method used to identify clearing led to the greater *CF* and *A* on clearing days in distant offshore regions. It is difficult to identify the root cause of greater *CF* and *A* on clearing days versus non-clearing days, but Garreaud and Munoz (2005) also demonstrated that the cloud deck tends to dissipate over CLLJ regions in contrast to an increase in cloudiness downstream of the jet core. This is also the case in this study as large scale conditions such as an intensified Pacific high and greater *LTS* on clearing days are in favor of the preservation of cloud deck in the regions except for coastal areas impacted by a CLLJ.

**3.3 Modeling of Clearing Growth Rates**

It has been already shown (Figs. 4-5) that clearings exhibit diurnal variability in dimensional characteristics, with rapid growth between 09:00 and 12:00 PST (Fig. S5). It is of interest now to examine what environmental parameters control the growth within this 3 h period based on the 306 clearing cases between 2009 and 2018. The GBRT modeling method was used to this end based on the method described in Section 2.4.

The coefficient of determination ($r^2$) between predicted and observed clearing growth rates for the 30 randomly selected testing datasets ranged between 0.52 to 0.77 with an average of 0.65. A multivariate linear regression model using the LASSO method (Tibshirani, 1996) was also applied to the obtained dataset to assess the performance of the GBRT model in comparison to the linear model. The $r^2$ value of the linear model varied between 0.08 and 0.11 with an average of 0.10, revealing the poor performance as compared to the GBRT model. As noted in at least one previous study (Klein 1997), linear models can explain less than 20% of the variance in low cloud amount on daily time scales. This is in contrast to monthly time scales for which such models perform much better and can explain over 50% of the variance (Klein and Hartmann, 1993; Norris and Leovy, 1994). Part of the success of the GBRT model to reproduce clearing growth rates can

be attributed to the complexity of the model, specifically its ability to capture non-linearity
between clearing growth rates and environmental parameters.
The range of $PD$s for each individual environmental parameter and the relative feature
importance are used here as two proxies for the sensitivity of clearing growth rates to that specific
parameter. Higher $PD$ ranges translate to a higher sensitivity of $GR_{Area}$ to that specific parameter,
indicating that it is likely a major influential factor. In addition, the relative feature importance
indicates how useful each parameter was in building the GBRT model. The range of $PD$ of clearing
growth rates and relative feature importance for all the parameters included in the GBRT model
are provided in Fig. 10, moving from left to right in order of highest to lowest influence in the
model. While it is expected that the results of these two methods of rankings do not match perfectly
(Fig. 10a and 10b), certain characteristics are similar between these two proxies: (i) using both
proxies, $T_{850}$ and $\omega_{700}$ appeared as the top and lowest ranking parameters, respectively; (ii) $q_{950}$
emerges as one of the most important parameters, being second and third place according to the
range of $PD$ and relative feature importance proxies, respectively; (iii) $AOD$ and $q_{700}$ emerged
among the four lowest-ranking parameters; and (iv) $SST$ and $V_{850}$ appear next to each other in the
ranking using both scoring proxies. There are some distinct differences among the ranking of
parameters as shown in Fig. 10. For instance, while $MSLP_{anom}$ appeared as a moderately influential
parameter in $GR_{Area}$ according to $PD$ proxy, this parameter turned out to be the second most
important variable using the relative feature importance proxy. In another example, $q_{850}$ has the
second least important rank according to relative importance feature proxy, but it is moderately
important based on the $PD$ range (Fig. 10a). The observed discrepancies between the results of
two proxies can stem from underlying differences in the methods used to quantify the relative
significance of each parameter. Moreover, the relative feature importance proxy may be less
susceptible to the unwanted influence of highly correlated input predictors on the ranking outcome
(Hastie et al., 2009).
Figure 11 shows the profiles of $PD$ for $GR_{Area}$ ($PD_{GRArea}$) relative to each individual
parameter tested, where increasing values of $PD_{GRArea}$ indicate that the corresponding change on
the x-axis for the value of the specific parameter is conducive to faster clearing growth. Note that
the 5th, 25th, 50th, 75th, and 95th percentiles of input parameter values are denoted in Figure 11 to
caution that sharp slopes in the bottom and top 5th percentiles are based on few data points and that
robust conclusions should not stem from those outer bounds. The response of $PD_{GRArea}$ to the
changes in $T_{850}$ is shown in Figure 11a. $T_{850}$ is closely linked to inversion strength variables such
as $LTS$ (Klein and Hartmann, 1993) and estimated inversion strength ($EIS$) (Wood and Bretherton,
2006). At constant $SST$, higher $T_{850}$ translates to higher $EIS$ and LTS values. It is well-established
that inversion strength plays a key role in controlling MBL cloud coverage (Klein and Hartmann,
1993). It is expected that higher $T_{850}$ decreases (increases) $GR_{Area}$ (cloud amount) by enhancing
stability. Figure 11a shows that up to 290 K, the profile of $PD$ exhibits a downward trend as $T_{850}$
increases. Above 290 K, $PD$ of $GR_{Area}$ starts to show the opposite trend with increasing $T_{850}$. As
noted in Brueck et al. 2015, "…increased stability is a necessary but not a controlling factor for
cloudiness, especially not when it is already sufficiently large. A further increase in inversion
strength may thus further limit cloudiness, because it increases the entrainment of relatively drier
and warmer air…". Figure 6b showed that $T_{850}$ was enhanced off the California coast on clearing
days, pointing to the high potential for warm continental air to impact the underlying cloud deck
via entrainment. It is important to note that, when the model was run with the same set of
parameters but replacing $T_{850}$ with $LTS$, the $PD$ profile of $LTS$ exhibited a qualitatively similar
trend to what was presented for $T_{850}$ in Fig. 11a.
The $PD_{GRArea}$ profile of $q_{950}$ shows increasing values as $q_{950}$ decreases below 8 g kg$^{-1}$ (Fig.
11b), coincident with dry air that can dissipate clouds and aid in clearing formation and expansion.
Similarly, the $PD$ profile of growth rate generally decreases as $q_{850}$ increases (Fig. 11f). In contrast
to the other level heights, the $PD_{GRArea}$ profile of $q_{700}$ exhibits an opposite trend but a smaller
influence on $GR_{Area}$ (Fig. 11j). This can be partly due to the fact that this layer of the FT is not as
close to the cloud layer, which in turn can permit other factors besides the entrainment process to
stand out. These various humidity parameters clearly show that conditions of dry air close to the
MBL top help clearings form and expand, with the most likely source being continental air. The
positive relationship between humidity at the level of clouds and low-level cloud amount was
reported in earlier studies (Albrecht 1981; Wang et al., 1993; Bretherton et al., 1995).
As previously explained, lower $SST$ values are associated with cloudiness (Fig. 11c) and
increased $LTS$ (Norris and Leovy 1994, Klein and Hartman 1993). Figure 11d displays the
dependence of $PD_{GRArea}$ on $V_{850}$, which is representative of flow in the FT. As discussed already,
clearings coincided with CLLJs and strong northerly flow at 850 hPa, which is consistent with the
sharp increase in $PD_{GRArea}$ as northerly wind speeds increased above 10 m s$^{-1}$ while otherwise
being flat for lower speeds. Stronger northerly flow is associated with offshore flow of dry and
warm air that can reside above the cloud top, which can dissipate the cloud layer after entrainment
and via enhanced shearing (via Kelvin-Helmholtz instability) and mixing of cloudy parcels with
warm and dry air in the FT (e.g., Rahn et al., 2016). As will be shown later, aircraft data showed
that typical wind speeds parallel to clear-cloudy interfaces were near or greater than 10 m s$^{-1}$ (Fig.
657    12).

For $PBLH$, Figure 11e suggests that above ~600 m, $PD_{GRArea}$ is relatively insensitive to
positive perturbations in PBLH, but below ~600 m, the shallower the MBL, the lower the value of
$PD_{GRArea}$. This potentially can be attributed to the fact that a shallower MBL could be more well-
mixed and moisture can get transported from the ocean surface to the cloud layer which promotes
cloudiness (Albrecht et al., 1995). Figure 11g shows that for $MSLP_{anom}$ between ~ -560 Pa and
~450 Pa, perturbations do not have much impact on $GR_{Area}$. However, above ~450 Pa, $GR_{Area}$ is
more susceptible to positive perturbations in $MSLP$. This confirms that stronger Pacific high
conditions in the study region promote the expansion of clearing events during the day. Based on
the $PD_{GRArea}$ profiles in Fig. 11h, clearings expanded faster as $U_{850}$ increased above 0 m s$^{-1}$ and
decreased below -3 m s$^{-1}$. Clearing growth due to negative zonal winds can be explained by the
offshore flow component, however, the reason for growth during periods of positive zonal winds
is unclear.
There was low variability in the range of $PD_{GR}$ for the rest of the parameters shown in Fig.
10: $AOD$ and $\omega_{700}$. Figure 11i shows a decrease in $PD_{GRArea}$ as $AOD$ increases up to the value of
~0.12, above which $PD_{GRArea}$ increases as a function of $AOD$. While it is expected that stronger
northerly winds associated with clearing expansion promote higher sea salt fluxes (i.e., higher
AOD), future work is warranted to investigate as to whether this process subsequently depletes
cloud water and thins out clouds via expedited drizzle production via broadening of cloud droplet
size distributions, as already suggested in Section 3.2.
The relationship between $\omega$ at 700 hPa and $PD_{GRArea}$ is complex. Brueck et al. (2015)
suggested that enhanced $\omega_{700}$ promotes cloudiness due to its link to higher $LTS$. Myers and Norris
(2013) further showed that stronger subsidence can reduce *CF* (at fixed inversion strength) by
pushing down the top of the MBL, which is also supported by Bretherton et al. (2013). The
*PD$_{GRArea}$* profile of $\omega_{700}$ exhibited a minimum point near a value of $0 - 0.2$ Pa s$^{-1}$, with increases
in *GR$_{Area}$* below and above that range. The increase in *PD$_{GRArea}$* with $\omega$ values above 0.2 Pa s$^{-1}$ can
be attributed to the negative influence of subsidence on lower *CF* (via pushing down the top of the
MBL) as discussed by Myers and Norris (2013). Conversely, the increase in *GR$_{Area}$* with
decreasing $\omega$ values below 0 Pa s$^{-1}$ can be due to upward motion reducing the strength of the
inversion capping the MBL, which is important to sustain the cloud deck. Vertical motions
represented by the $\omega_{700}$ parameter could also induce dynamical circulations affecting cloud top
processes such as shear and entrainment.
It is important to caution that the interpretation of results from the GBRT simulations are
speculative and rooted in documented physical relationships between the various parameters
shown in Figs. 10-11 and low cloud behavior. One way to try to validate some of the conclusions
above is with airborne data for case studies. For instance, in situ data can help confirm the nature
of factors discussed above during clearing events, including vertically-resolved winds, primary
marine aerosol fluxes in different wind regimes, humidity and temperature of air within and
above the MBL, and potential for mixing of air above and below the MBL top. The next section
is an attempt to conduct this exercise using three airborne case studies.
**3.4 Airborne Case Studies**
To gain a more detailed perspective on clearings in the study region, three case flights are
examined from the 2016 FASE airborne campaign. For context, Crosbie et al. (2016) examined
three different case flights during the 2013 NiCE campaign and provided the following insights,
which motivated the FASE flights for further statistics: (i) two of the three clearings (RF19 on 1
August 2013, RF23 on 7 August 2013) were immediately adjacent to the coastline and had reduced
specific humidity in the MBL on the clearing side, suggestive of dry continental offshore wind
laterally mixing into and dissipating clouds; (ii) the latter two cases also had enhanced temperature
in the clear column at cloud-relevant altitudes, which help explain the lack of clouds in the clear
column; and (iii) the other clearing flight (RF16 on 29 July 2013) had the clearing positioned to
the west of a cloud deck, which was associated with a CTD event along the coastline to the east of
the clearing (i.e., southerly surge). The latter case exhibited warmer temperatures in the clear
column only in the top 100 m of the MBL with similar specific humidity profiles, but with cooler
and moister air above the inversion base in the clear column. This case was suspected to be linked
to entrainment and mixing of dry air into the cloud deck to produce the clearing, but it was not a
case of subsidence/divergence, otherwise the air in the clear column would have been warmer and
drier above the inversion base.
For the three FASE case flights, the clearing was always situated to the west of a cloud
deck touching the coastline (Figs. 2, S1-S2). This positioning is reminiscent of NiCE RF16, which
was less sensitive to lateral entrainment of continental air in comparison to the other two NiCE
flights. Wind data were decomposed into *u* and *v* components to represent speeds that are
perpendicular and parallel, respectively, to the clear-cloudy interface. Figure 2d illustrates an
example of how these two components of winds varied during RF09A. There were substantial
changes in *v* on the two sides of the clear-cloud border, with stronger northerly winds on the clear

side, reaching as high as 20 m s$^{-1}$, in contrast to about half that magnitude on the cloudy side. Wind speed with the intensity of as high as 20 m s$^{-1}$ is close to the values reported in previous studies associated with California CLLJs (Parish 2000; Ranjha et al., 2013; Lima et al., 2018). Furthermore, wind profiles obtained from soundings (Fig. 12) exhibit the structure similar to CLLJ on clearing columns with enhanced horizontal wind speed at the altitude near the MBL top. It is noteworthy that the cloud edge tends to reside in the transition region where the near cloud top flow becomes similar to CLLJ (Figs. 2d and 12). The same substantial change in $v$ across the interface was also present in RF08 and RF09B with stronger $v$ winds always on the clear side. There was no substantial change in the $u$ component of wind speed between the two columns in each of the three flights.

To extend upon the possibility of shearing effects, absolute changes in $v$ ($/v/$) were calculated for level legs performed at the clear-cloudy border for the three research flights (Table 2). For consistency, these calculations were based on level legs of a constant length of ~40 km with relatively equal spacing on both sides of the clear-cloudy border. $/v/$ was calculated by multiplying 40 km by the slope of the linear fit of $v$ versus distance from cloud edge, where negative (positive) x values represent distance away from the edge on the clear (cloud) side. The results reveal that the horizontal wind shear was strongest somewhere between mid-cloud and cloud top altitudes, with the lowest values at the FT level. The lowest values in the MBL were observed in the surface legs. This can be attributed to turbulent transport of the momentum (Zemba and Friehe 1987) to the surface and the consequent drop in CLLJ wind speeds in the clear column. In addition, Fig. S7 shows absolute horizontal shear ($|dv/dx|$) as a function of distance from the cloud boundary for the parallel component of horizontal wind speed. Horizontal shear profiles for all research flights (Fig. S7) are slightly noisy especially at the surface legs, but they show the presence of the greatest horizontal wind gradient within 5 km length away from clear-cloudy edge. Shear at the clear-cloudy edge, especially at cloud levels, can support clearing growth through enhancing the mixing of cloudy and clear air. Crosbie et al. (2016) also showed using the case of NiCE RF19 that that mixing of cloudy air with adjacent clear air can be an important contributor to cloud erosion and thus expansion of clearings. To probe deeper into the clearing cases, the subsequent discussion compares vertically-resolved data on both sides of the clear-cloudy border based on soundings and level legs.

### 3.4.1  RF08

RF08 (2 August 2016) represented a case similar to the NiCE RF16 (29 July 2013) case study in Crosbie et al. (2016) where cooler and moister air above the inversion in the clear column was speculated to be due to entrainment and mixing eroding the cloud rather than subsidence and divergence catalyzing cloud dissipation. Of note is that there was rapid infill of cloud the night of the NiCE FR16 flight. FASE RF08 data showed that potential temperature was warmer (~1 K) in the MBL of the clear column as compared to the cloudy column, while in the FT, the air was slightly warmer on the cloudy side (Fig. 12). *SST* was also approximately 0.4 K higher in the clear column (Table 3). Specific humidity was almost identical in the MBL on both sides, but air was moister above the inversion base on the clear side. As noted above, vertical profiles of $u$ revealed little difference between the two columns, but $v$ values were nearly twice as high in the clear column extending from the surface to approximately 200 m above cloud top. Surface wind speeds were also enhanced on the clear side, which resulted in greater friction velocity ($u* = 0.40$ m s$^{-1}$ vs 0.15 m s$^{-1}$ on the cloudy side).

An important feature was the wind maximum in and above the inversion layer on the clear side, which resulted in larger vertical shear across the inversion on the clear side (5.44 m s$^{-1}$) compared with the cloudy side (0.8 m s$^{-1}$) (see $\Delta U$, Table 3). The strong shear on the clear side likely facilitated mixing of MBL air with drier and warmer FT air. This is supported by a lower temperature gradient $(\Delta\theta_l/\Delta z)_{max}$ in the inversion layer of the clear column (0.32 K m$^{-1}$ versus 0.38 K m$^{-1}$), which was thicker than the cloudy column (82 m versus 55 m). The wind maximum in the clearing also enhanced moisture advection, which counteracted the accumulation of moisture caused by mixing induced by vertical shear. This was most significant at the cloud top level as seen in the largest difference in the edge-parallel wind $/v/$ (Table 2). In the absence of cloud, the effects of longwave radiative cooling close to the cloud top level would be subdued allowing shear-induced mixing to erode the sharpness of the inversion. Redistribution of moisture into the inversion also serves to insulate lower layers from longwave cooling, further delaying the formation of cloud. The difference in $/v/$ was smallest close to the surface, indicating that the wind maximum in the clearing had a (comparatively) reduced effect in enhancing surface moisture fluxes. Satellite imagery confirms that later in the day, the cloud layer filled-in partially where the clearing was with the presumed help of nocturnal radiative forcing.

The cloud layer in RF08 was the thinnest (131 m) with the shallowest MBL among all three cases. In addition, the lowest $N_d$ (107 cm$^{-3}$), largest $r_e$ (6.6 µm), and highest cloud base rain rate (0.48 mm day$^{-1}$) was measured in RF08 of all three cases. The enhanced rain can likely explain why the surface aerosol concentrations from the PCASP were lowest in RF08 (106-108 cm$^{-3}$ vs 186-236 cm$^{-3}$ for the other two flights) even though surface winds were highest, specifically due to efficient wet scavenging of aerosols. This possibility is at least linked to the speculation reported earlier in Sections 3.2 and 3.3 that stronger northerly winds linked to the growth of clearings result in sea salt expediting rain formation in clouds and thus thinning them out. In support of this notion, cloud water composition results are of relevance as they provide an indication of the relative influence of giant $CCN$ ($GCCN$) in the form of sea salt, as previously demonstrated in the region by Dadashazar et al. (2017). The combined concentration of sodium ($Na^+$) and chloride ($Cl^-$) was 60 µg m$^{-3}$, 33 µg m$^{-3}$, and 64 µg m$^{-3}$ for RF08, RF09A, and RF09B, respectively. In contrast, the average combined sum of $Na^+$ and $Cl^-$ for all samples collected in FASE was 14 µg m$^{-3}$. Based on a two-tailed student's t-test with 95% confidence, the means of RF08 and RF09B were significantly different than the mean of all FASE samples. The $Cl^-:Na^+$ mass ratios in all three FASE clearing flights (RF08 = 1.80, RF09A = 1.78, RF09B = 1.79) were very close or matching that of pure sea salt (1.81), providing more confidence that sea salt was impacting these clouds via serving as $CCN$. The cloud water results are in support of $GCCN$ enhancing drizzle in RF08 and thus thinning out clouds and removing aerosol underneath the cloud base. It is unclear with this dataset though as to what role the impact of sea salt in depleting clouds of their water had to do with the actual clearing, but at least there is support for this process potentially impacting the cloudy column.

Figure S8 shows vertical profiles of aerosol concentrations on both sides of the clearing border, highlighting differences above cloud top level especially in RF09A and RF09B with higher values in the cloudy column. Higher aerosol concentrations were also observed in the cloud column in the sub-cloud layer even though surface wind speeds were always higher in the clear column for all three flights. Surface winds and thus sea spray production do not exclusively influence the aerosol concentrations. A likely explanation of higher concentrations in the MBL in the cloudy column is that there could be entrainment of more polluted free tropospheric aerosol as has been reported to be a common occurrence during the FASE flights (Mardi et al., 2019). As

also reported during FASE, there can be sub-cloud evaporation of drizzle resulting in droplet residual particles that contribute to the aerosol concentration budget in the cloudy column (Dadashazar et al., 2018).

Figure 13 displays turbulence parameters such as variance in the three components of wind speed (Fig. 13a-c), turbulent kinetic energy (Fig. 13d), and buoyancy flux (Fig. 13e). Stronger horizontal wind speed gradients, and consequently stronger shear production, near the surface on the clear side resulted in greater variance in the horizontal wind components at all MBL levels. Both $\overline{u'^2}$ and $\overline{v'^2}$ exhibit a general downward trend with increasing altitude, which is also supportive of shear driven turbulence. On the other hand, $\overline{w'^2}$, which is closely associated with cloud layer properties, exhibits a different trend on the cloudy side as it increases from cloud base to mid-cloud level. For surface and above cloud base levels, $\overline{w'^2}$ is higher in the clear column likely due to the combined influence of shear and buoyancy terms on the turbulence budget. On the other hand, in the mid-cloud layer, $\overline{w'^2}$ is slightly higher (Fig. 13c) in the cloudy column as compared to clear column, which can be attributed to the buoyancy flux (Fig. 13e). It is also interesting to note that RF08 is the only flight with a minimum in $\overline{w'^2}$ being at the level above cloud base in the cloudy column relative to other MBL levels. This is most likely due to lower buoyancy production in the cloud layer of RF08 as compared to the other flights.

To further investigate the relative role of each buoyancy and shear term in the turbulence budget, the $-z_i/L_{MO}$ ratio was compared between the two columns (Table 3). This ratio is an order of magnitude greater in the cloudy column as compared to clear one due to the latter column having stronger shear and reduced buoyancy flux. This confirms that shear is most likely the dominant mechanism for turbulence production in the clear column in the absence of the cloud layer.

### 3.4.2  RF09A and RF09B

The two flights on 3 August 2016 allowed for an opportunity to contrast clearing properties at two different times on the same day at roughly the same location (~20 km apart). Owing to their similarities, they are discussed together here. The clearing module in RF09A was performed between 11:00 and 12:30 PST, while that during RF09B was performed between 15:00 - 17:00 PST. Similar to RF08, MBL air in the clear column of RF09A and RF09B was slightly warmer than the cloudy column; however, the magnitude of the temperature difference (clear – cloudy) decreased from RF09A (~1.1K) to RF09B (~0.8K). *SST* was also greater by 0.4 K in the clear column of RF09A as compared to the cloud column, while it was slightly cooler by 0.1 K in the clear column of RF09B.

Specific humidity profiles in RF09A/RF09B exhibit more subtle differences as compared to RF08. In contrast to RF08, air in RF09A above the inversion base was drier and warmer in the region immediately above the inversion base and differences above the inversion base are less clear for RF09B. During both RF09A and RF09B, the clear profile exhibited steadily decreasing levels of water vapor with altitude, while the cloudy column was more well-mixed. The *v* component of wind speed again exhibited substantially greater values in the clear column as compared to the cloudy column for both RF09A and RF09B. Looking at the inversion layer properties (Table 3), the temperature gradient was lower and shear was greater in the clear column of RF09A and RF09B. Inversion depth was also greater in the clear column of RF09A, but less so for RF09B.

The sounding data in RF09A qualitatively resemble those from NiCE RF19 on 1 August 2013 where Crosbie et al. (2016) suspected that there was increased local subsidence and divergence in the clear column. Similar to their case, we observed the following in the clear column of RF09A: (i) warmer and drier air above and below the inversion base; (ii) the inversion base height was lower (354 m versus 375 m) with reduced temperature gradient in the inversion layer (0.33 K km$^{-1}$ versus 0.41 K km$^{-1}$); and (iii) potential temperature exhibited warming and drying in the layer equivalent to the top 100 m of cloud. The RF09B case differed in that above the inversion base, the air in the clear column was not warmer and drier but very slightly cooler and moister, similar to RF08. This potentially is due to the diurnal nature of the clearing system where there is a stronger forcing to dissipate clouds during mid-day with the help of subsidence of dry and warm air from the FT, whereas later in the afternoon that process switches to a scenario where cooler and moister air exists above the inversion base and there is a waiting process for stronger radiative forcing to form a cloud again.

The cloud layer is the thickest in RF09A (191 m) among all three case flights. The cloud layer became thinner (137 m) later in the day during RF09B as a result of a change in the lifting condensation level (*LCL*), where cloud base increased from 217 m to 265 m. Moreover, *LWP* decreased during the day from 32 g m$^{-2}$ to 18 g m$^{-2}$. It is important to note that the adiabaticity parameter, defined as the ratio of measured *LWP* to *LWP* of an adiabatic cloud, exhibited values of 0.75, 0.76, and 0.83 for RF08, RF09A, and RF09B, respectively. These adiabaticity values are close to the average value of 0.766 for the region reported in Braun et al. (2018). The clouds were quite thin near the interface based on the relatively low values of *LWP* in contrast to typical conditions observed in the region based on airborne measurements in the same campaigns (Fig. 3 of Sorooshian et al., 2019). Other cloud properties such as $N_d$, $r_e$, and rain rate were quite similar in both RF09A and RF09B. $N_d$ was greater in RF09A and RF09B as compared to RF08, corresponding to smaller values of $r_e$ and suppressed drizzle. The dataset cannot provide unambiguous evidence as to whether the higher surface aerosol concentrations in RF09A and RF09B, as compared to RF08, were due to (or led to) suppressed drizzle.

Profiles of $\overline{u'^2}$ and $\overline{v'^2}$ exhibited downward trends with increasing altitude for RF09A and RF09B, in general agreement with the findings for RF08. One contrasting aspect was the comparison of $\overline{v'^2}$ between clear and cloudy columns, which mirrored RF08 during RF09A, while in RF09B, the values of $\overline{v'^2}$ for the clear side were substantially lower. In addition, $\overline{w'^2}$ profiles during RF09A and RF09B are substantially enhanced in the cloudy column as compared to RF08, with maxima in the cloud layer. There is an accompanying increase in the buoyancy flux for these profiles suggestive of a more significant contribution of buoyancy to *TKE* production (Fig. 13e). Although more subtle, $\overline{u'^2}$ values also showed an increase in the cloudy column of RF09A and RF09B relative to the clear column, also supportive of the role of buoyancy in these cases. In addition, *TKE* profiles (Fig. 13d) were largely influenced by variances in the horizontal component of wind speed ($\overline{u'^2}$ and $\overline{v'^2}$) which led to overall greater *TKE* values in the clear column except for RF09B.

Drizzle may be an important factor in governing the differences in buoyancy between the cloudy columns of RF09A/B and RF08. While no obvious decoupling of the RF08 cloudy MBL is observed, this profile may rely more heavily on shear production to maintain a well-mixed state. The clearing persisted following RF08, while there was a rapid infilling of cloud during the night following RF09A/B, similar to the case presented by Crosbie et al. (2016), which was also non-

drizzling. While the nocturnal radiative environment has been shown to be conducive to infilling of clearings, we hypothesize that other factors that promote tighter coupling between the cloud layer and the surface (such as a lack of drizzle) may also contribute.

**4   Conclusions**

This study extends upon recent works interested in large stratocumulus clearings that significantly impact albedo and have implications for fog, cloud, and weather forecasting. We specifically reported on ten years (2009-2018) of satellite and reanalysis data to characterize the temporal behavior, spatial and dimensional characteristics, growth rates, and governing environmental properties controlling the growth of clearings off the U.S. West Coast. We also examined three case flights from the 2016 FASE campaign that probed clearings to gain a deeper insight at finer spatial scales to try to validate speculated links between environmental parameters and clearing growth rates based on machine learning simulations using satellite and reanalysis data. The major results were as follows:

(i)     Summertime (wintertime) experiences the highest (lowest) frequency of clearings as suggested by satellite retrievals.

(ii)    The centroid of clearings is located around coastal topographical features along the California coastline, specifically Cape Blanco and Cape Mendocino.

(iii)   The median length, width, and area of clearings between 09:00 and 18:00 (PST) increased from 680 km, 193 km, and ~67,000 km$^2$, respectively, to ~1231 km, 443 km, and ~250,000 km$^2$. The most growth occurred between 09:00-12:00.

(iv)    The most influential factors in clearing growth rates of total area between 09:00-12:00 were $T_{850}$, $q_{950}$, $SST$, and $MSLP_{anom}$ using two different scoring methods. Compared to non-clearing days, clearing days were characterized by having an enhanced Pacific high shifted more towards northern California, offshore air that is warm and dry, faster coastal surface winds, higher lower tropospheric static stability, and stronger subsidence.

(v)     Clearing days exhibited higher values of $N_d$ and reduced values of $r_e$, $\tau$, and $LWP$ near the California coast where clearings form and evolve. However, the mean cloud albedo over the entire study domain was actually higher on clearing days.

(vi)    Airborne data revealed that extensive horizontal shear at cloud-relevant altitudes, with much faster winds with low-level jet structure parallel to the clearing edge on the clear side as compared to the cloudy side. This helped to promote mixing and thus dissipation of clouds. Differences in sounding profiles reveal that warm and dry air in the free troposphere additionally promoted expansion of clearings.

More research is needed to further characterize clearings and the broader regions they evolve in. For instance, it remains uncertain as to if there is a physical link between the existence of clearings and a higher domain-wide cloud albedo on clearing days. More data such as those provided by GOES platforms can help understand processes occurring at the microscale that scale up to more climatologically relevant scales. The results of this work showed that there are important diurnal features that require additional examination with in situ observations. One of the hypotheses posed in this work requiring more measurements and statistical robustness is the link between sea salt aerosol and the formation and evolution of clearing events. Clearing days are characterized by having stronger northerly winds, which translate into higher sea spray fluxes and subsequently can impact clouds via faster onset of drizzle. This chain of events subsequently can thin out clouds via depletion of cloud water. Targeted experiments to examine these types of events

will help advance understanding about their nature, which can then be contrasted with clearings
along other coastal regions such as the southeastern Atlantic Ocean. Also, the nature of clearings
has direct relevance to CTD events that evolve in similar regions as discussed by Juliano et al.
(2019a,b).

**Data availability**
Airborne field data used in this work can be found on the Figshare database (Sorooshian et al.,
2017; https://figshare.com/articles/A_Multi-Year_Data_Set_on_Aerosol-Cloud-Precipitation-
Meteorology_Interactions_for_ Marine_Stratocumulus_Clouds/5099983). Also, the other data
used in this study are available at websites provided in Section 2.

**Author contributions**
EC and AS designed the study. HJ, AS, EC, and HD conducted the research flights during the
FASE field campaign. MSM and HD developed the image analysis tool to analyze GOES images.
MP, HD, and MAM ran the GBRT model. HD analyzed the collected data. AB, MB, and XZ
provided input on the results and draft. AS and HD wrote the paper. EC, MAM, AB, MB, and XZ
revised the manuscript.

**Competing interests**
The authors declare that they have no conflict of interest.

**Acknowledgments**
This work was funded by Office of Naval Research grant N00014-16-1-2567 and NASA grants
NNX14AM02G and 80NSSC19K0442, the latter of which is in support of the ACTIVATE Earth
Venture Suborbital-3 (EVS-3) investigation, which is funded by NASA's Earth Science Division
and managed through the Earth System Science Pathfinder Program Office. We acknowledge
Johannes Mohrmann and an anonymous reviewer for their constructive feedback.

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

**Table 1.** Summary of reanalysis and satellite data products used in this study. For the rows with multiple products, underlined entries
correspond to each other between different columns.

| Input coordinate for data download | Parameter | Source | Product identifier | Spatial resolution | Vertical level | Temporal resolution | Reference |
|---|---|---|---|---|---|---|---|
| 20°-60° N, 110°-160° W | Visible band imagery | GOES-11/15 imager | NA | 1 km × 1 km at nadir | NA | 30 min | Menzel and Purdom, 1994 |
| 20°-60° N, 110°-160° W | Mean sea level pressure | MERRA-2 model | M2I3NPASM | 0.5° × 0.625° | NA | 3 h | Bosilovich et al., 2016 |
| 20°-60° N, 110°-160° W | Air temperature | MERRA-2 model | M2T1NXFLX /M2I3NPASM | 0.5° × 0.625° | Sea surface, 950, 850, 700 hPa | 1 h//3 h | Bosilovich et al., 2016 |
| 20°-60° N, 110°-160° W | Geopotential height | MERRA-2 model | M2I3NPASM | 0.5° × 0.625° | 850, 500 hPa | 3 h | Bosilovich et al., 2016 |
| 20°-60° N, 110°-160° W | Wind speed | MERRA-2 model | M2T1NXFLX | 0.5° × 0.625° | Surface, 950, 850, 700 hPa | 1 h/3 h | Bosilovich et al., 2016 |
| 20°-60° N, 110°-160° W | Vertical pressure velocity | MERRA-2 model | M2I3NPASM | 0.5° × 0.625° | 700 hPa | 3 h | Bosilovich et al., 2016 |
| 20°-60° N, 110°-160° W | Planetary boundary layer height | MERRA-2 model | M2T1NXFLX | 0.5° × 0.625° | NA | 1 h | Bosilovich et al., 2016 |
| 20°-60° N, 110°-160° W | Sea surface temperature | MERRA-2 model | M2T1NXOCN | 0.5° × 0.625° | NA | 1 h | Bosilovich et al., 2016 |
| 20°-60° N, 110°-160° W | Specific humidity | MERRA-2 model | M2I1NXASM/M2I3NPASM | 0.5° × 0.625° | 10 m, 950, 850, 700 hPa | 1 h/3 h | Bosilovich et al., 2016 |
| 20°-60° N, 110°-160° W | Aerosol optical depth AOD | MERRA-2 model | M2I3NXGAS | 0.5° × 0.625° | NA | 3 h | Bosilovich et al., 2016 |
| 30°-50° N, 115°-135° W | Cloud optical thickness liquid | MODIS-Terra/Aqua | MOD08_D3/MYD08_D3 | 1° × 1° | NA | Daily | Hubanks et al., 2019 |
| 30°-50° N, 115°-135° W | Cloud fraction day | MODIS-Terra/Aqua | MOD08_D3/MYD08_D3 | 1° × 1° | NA | Daily | Hubanks et al., 2019 |
| 30°-50° N, 115°-135° W | Cloud water path liquid | MODIS-Terra/Aqua | MOD08_D3/MYD08_D3 | 1° × 1° | NA | Daily | Hubanks et al., 2019 |
| 30°-50° N, 115°-135° W | Cloud effective radius liquid | MODIS-Terra/Aqua | MOD08_D3/MYD08_D3 | 1° × 1° | NA | Daily | Hubanks et al., 2019 |



**Table 2.** Absolute changes in the parallel component of horizontal wind speed relative to the cloud
edge, $|\Delta v|$ in units of m s$^{-1}$, across various legs using FASE aircraft data. Values were calculated
based on a 40 km leg distance (approximate length of each leg). Values for the cloud top leg were
estimated using the sawtooth leg performed across the cloud top boundary. The free troposphere
level leg was not conducted in RF08 and thus left blank.

| | RF08 | RF09A | RF09B |
|---|---|---|---|
| Free troposphere | | 0.4 | 1.6 |
| Cloud top | 9.6 | 6.4 | 4.8 |
| Mid-cloud | 7.2 | 6.8 | 6.0 |
| Above cloud base | 6.8 | 5.2 | 5.2 |
| Surface | 3.6 | 2.4 | 0.0 |


**Table 3.** Summary of thermodynamic, dynamic, and cloud properties on both sides of the clear-cloudy interface for three FASE case
research flights (RFs). $U$ represents total horizontal wind speed ($U = \sqrt{u^2 + v^2}$) across the depth of the inversion layer.


| | Cloudy | | | Clear | | |
|---|---|---|---|---|---|---|
| | RF08 | RF09A | RF09B | RF08 | RF09A | RF09B |
| $SST$ (K) | 286.6 | 287.1 | 287.3 | 287.0 | 287.5 | 287.2 |
| Surface wind (m s$^{-1}$) | 11.3 | 11.1 | 11.6 | 13.2 | 12.3 | 11.5 |
| $u^*$ (m s$^{-1}$) | 0.15 | 0.19 | 0.11 | 0.40 | 0.32 | 0.25 |
| $w^*$ (m s$^{-1}$) | 0.44 | 0.64 | 0.68 | 0.44 | 0.53 | 0.38 |
| $-Z_i/L_{MO}$ | 9.8 | 15.7 | 49.1 | 0.8 | 2.2 | 1.4 |
| Inversion-base height (m) | 367 | 375 | 391 | 359 | 354 | 386 |
| Inversion-top height (m) | 422 | 441 | 457 | 443 | 440 | 455 |
| Inversion depth (m) | 55 | 66 | 66 | 82 | 86 | 69 |
| $\Delta\theta_l$ (K) | 7.4 | 8.6 | 7.0 | 7.3 | 7.6 | 5.4 |
| $(\Delta\theta_l/\Delta z)_{Max}$ (K m$^{-1}$) | 0.38 | 0.41 | 0.25 | 0.32 | 0.33 | 0.23 |
| $\Delta q_T$ (g kg$^{-1}$) | -3 | -3.2 | -2.6 | -2.9 | -3.3 | -2.6 |
| $\Delta U$ (m s$^{-1}$) | 0.80 | 1.35 | 1.35 | 5.44 | 2.50 | 5.32 |
| Cloud base (m) | 242 | 217 | 265 | | | |
| Cloud top (m) | 372 | 408 | 401 | | | |
| Cloud depth (m) | 131 | 191 | 137 | | | |
| Cloud $LWP$ (g m$^{-2}$) | 15 | 32 | 18 | | | |
| $R_{cb}$ (mm day$^{-1}$) | 0.48 | 0.09 | 0.07 | | | |
| $r_e$ (μm) | 6.6 | 6.0 | 5.9 | | | |
| $N_d$ (cm$^{-3}$) | 107 | 141 | 148 | | | |
| Surface PCASP (cm$^{-3}$) | 108 | 206 | 236 | 106 | 186 | 207 |



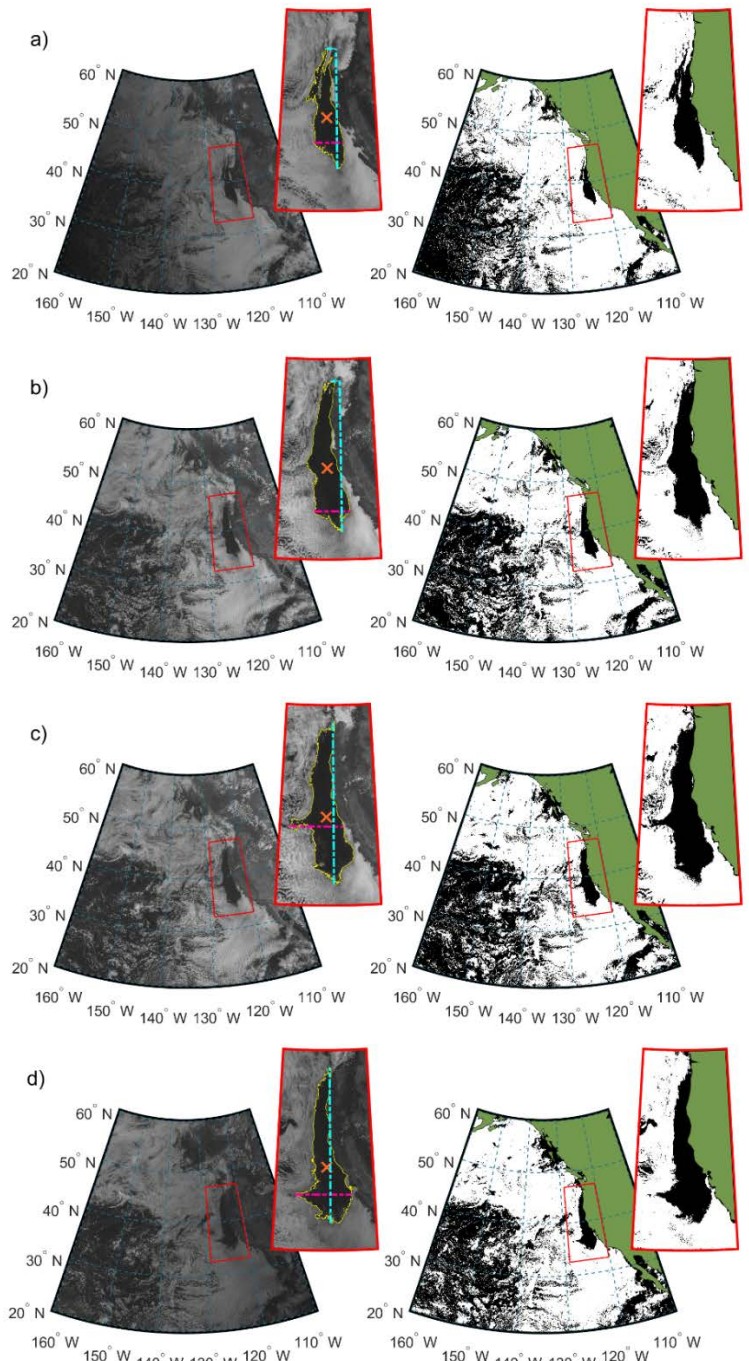

**Figure 1.** Sequence of data processing with GOES imagery at four times during a day: (i) 16:15
UTC 09 August 2011; (ii) 19:15 UTC 09 August 2011; (iii) 20:45 UTC 09 August 2011; and (iv)
01:15 UTC 10 August 2011. Left panels show visible-band images of a clearing event obtained
from GOES-11 data, while the right panel is produced using cloud masking. Note that the clearing
border, centroid, and lengths (x and y) are overlaid on the GOES images. Local time (PST) requires
subtraction of seven hours from UTC time.

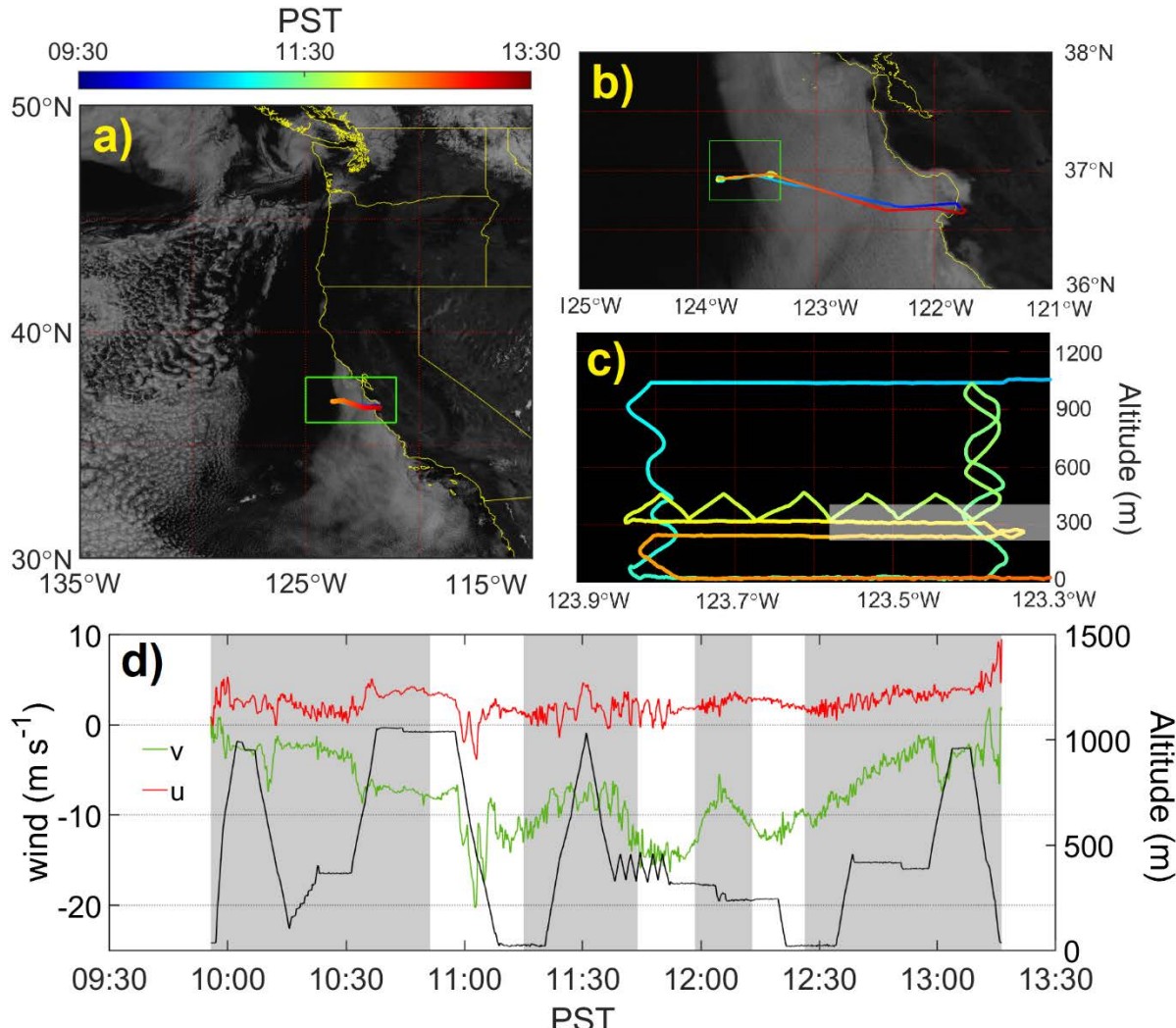

**Figure 2.** a) GOES 15 visible band image (11:45 (18:45) PST (UTC) on 03 Aug 2016) with the overlaid flight path of FASE RF09A. b) Zoomed-in view of the satellite image to highlight the clear-cloudy border. c) Aircraft flight strategy at the cloudy-clear interface for the green box highlighted in b). Cloud borders are denoted by a shaded box. d) Time series of flight altitude and horizontal wind speed, which is decomposed into two components that are perpendicular (*u*) and parallel (*v*) to the cloud edge. Wind speeds were smoothed using low-pass filtering. Parts of the flight that sampled air on the cloudy side of the clear-cloudy border are shaded in grey.

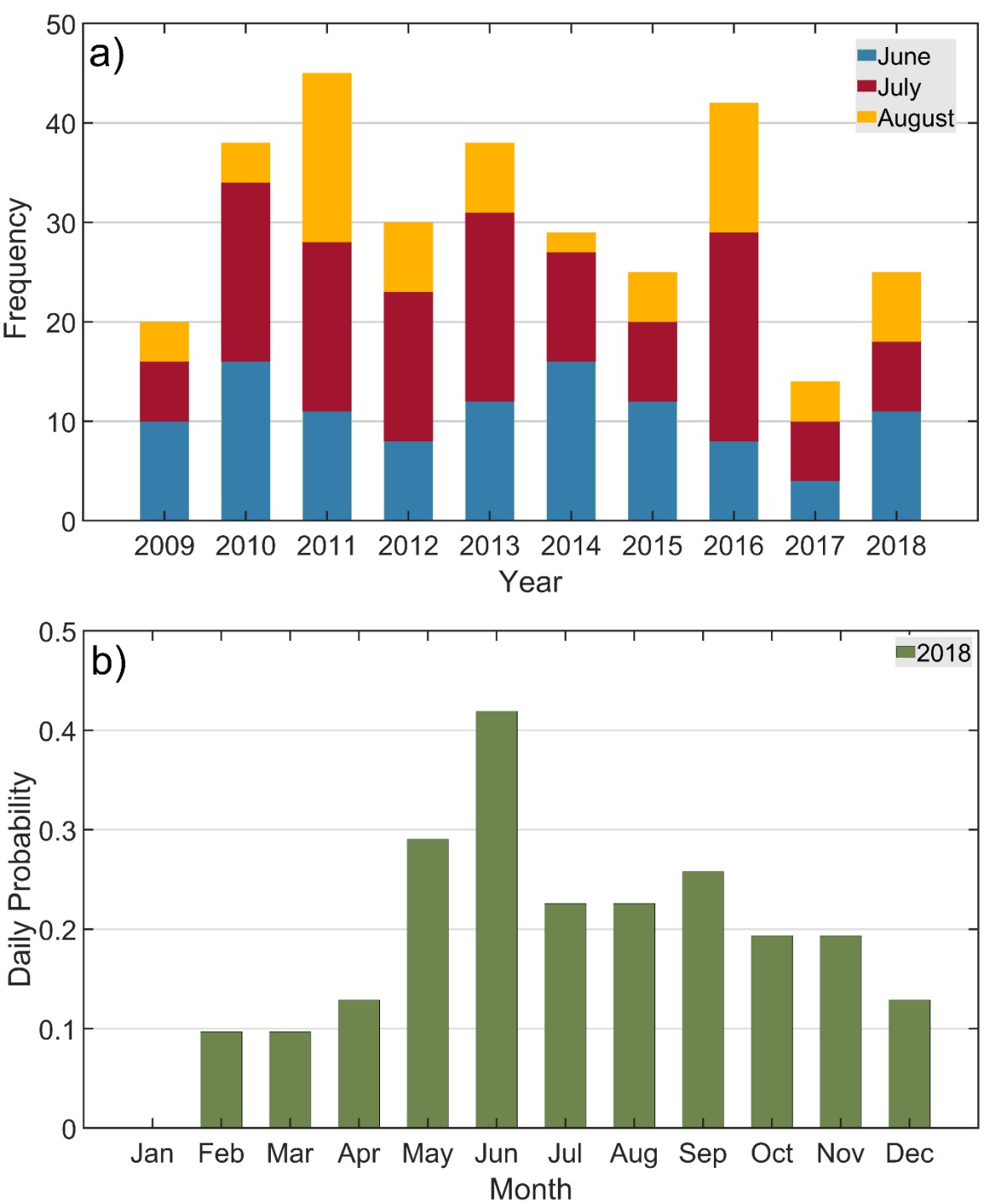

**Figure 3.** a) Frequency of clearing events in the study region for each summer month between
2009 and 2018. b) Daily probability of clearing events (i.e., days with clearings divided by total
days in that month) in each month of a representative year, 2018.

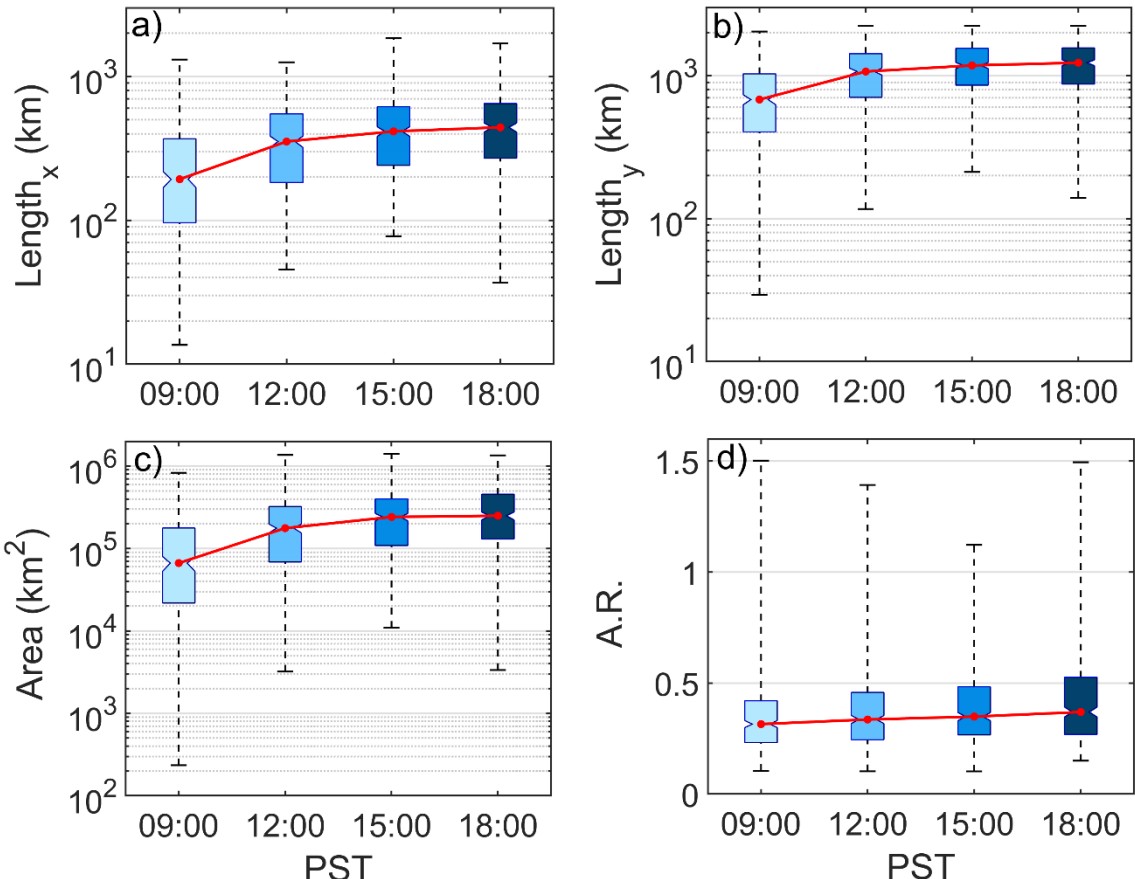


**Figure 4.** Diurnal profiles of (a) widest point of clearings at a fixed latitudinal value, (b) longest
dimension between the maximum and minimum latitudinal coordinates of a clearing regardless of
longitudinal value, (c) total clearing area, and (d) aspect ratio of clearing (i.e., width divided by
length using the maximum values as described by panels a-b). The box and whisker plots show
the median values (red points), the 25[th] and 75[th] percentile values (bottom and top of boxes,
respectively), and minimum and maximum values (bottom and top whiskers, respectively).

1461

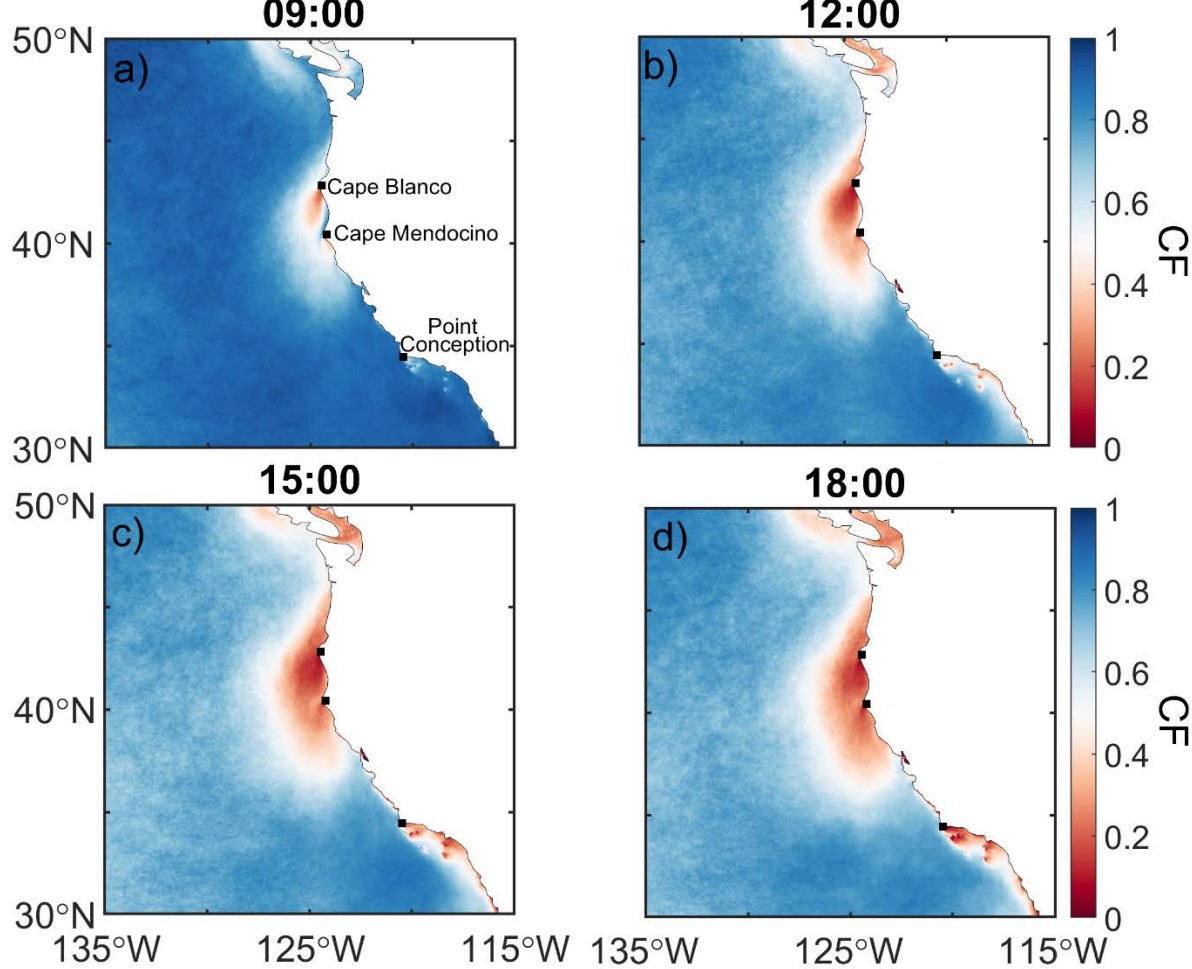

**Figure 5.** Diurnal profiles (PST times shown; add 7 h for UTC) of cloud fraction (*CF*) in the study region based on GOES imagery data from 306 clearing cases between 2009 and 2018 during JJA months.

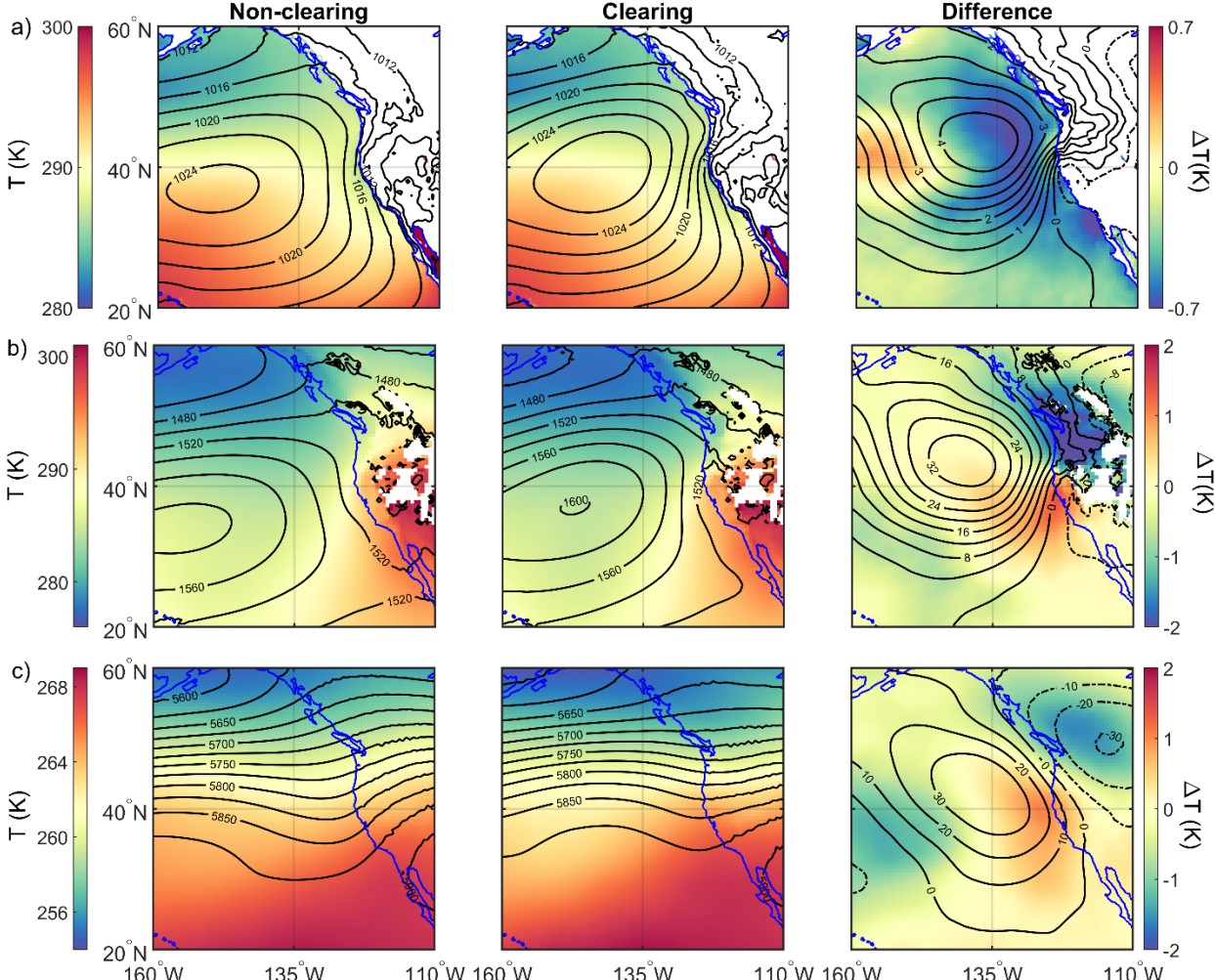

1468
**Figure 6.** Climatology of non-clearing and clearing days as well as their differences (clearing minus non-clearing) during the summers (JJA) between 2009 and 2018 for a) mean sea level pressure (contours in hPa) and air temperature (color map) at sea surface, b) 850 hPa geopotential heights (contours in m) and air temperature (color map), and c) 500 hPa geopotential heights (contours in m) and air temperature (color map). The data were obtained from MERRA-2 reanalysis. Differences (clearing minus non-clearing) are shown in the farthest right column with separate color scales. White areas indicate no data were available.

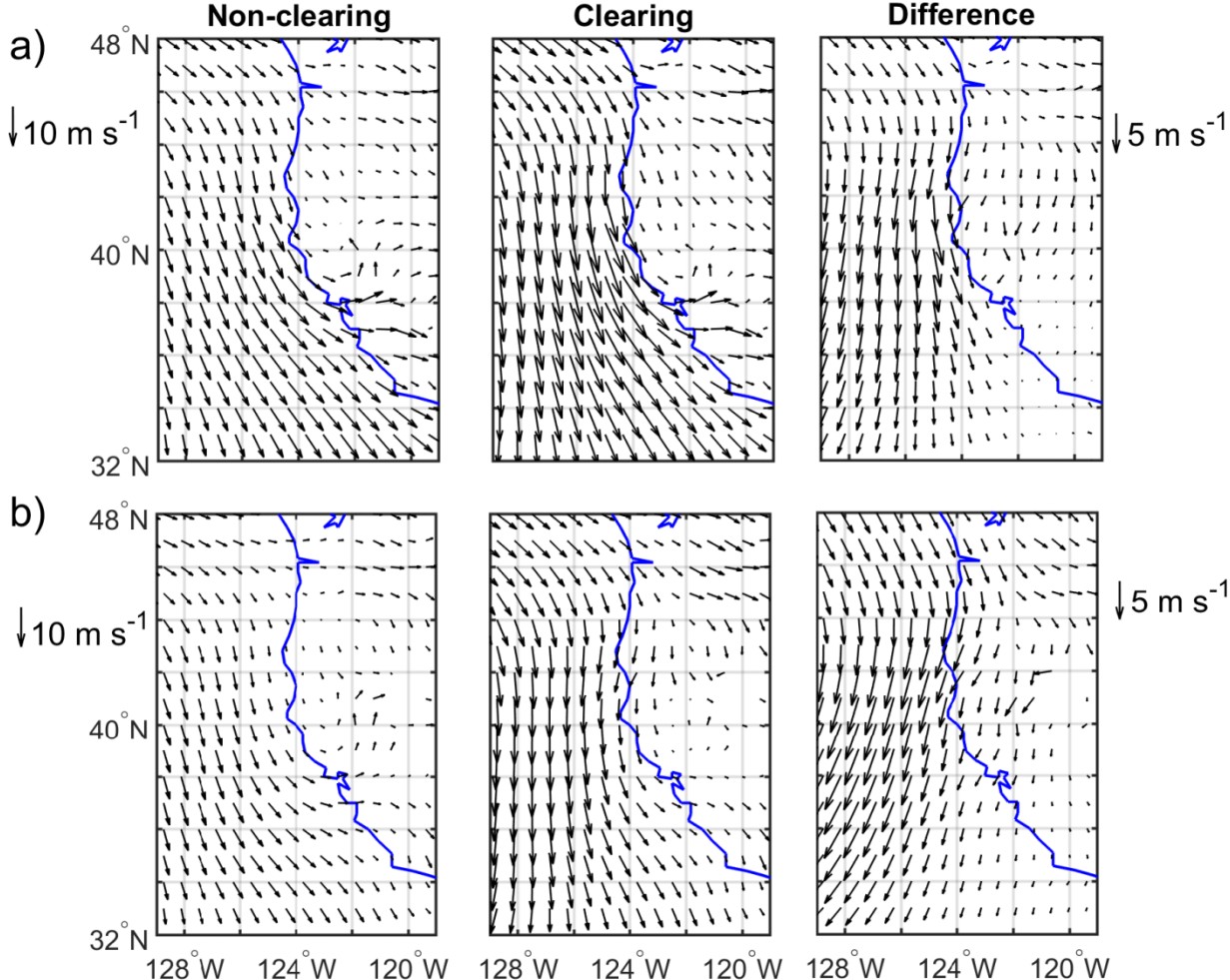

**Figure 7.** Same as Fig. 6 but for wind speed at the a) surface and b) 850 hPa. Reference wind vectors are shown on the far left for the left two columns, with separately defined vectors on the far right for the difference (clearing minus non-clearing) plots in the farthest right column.

1483

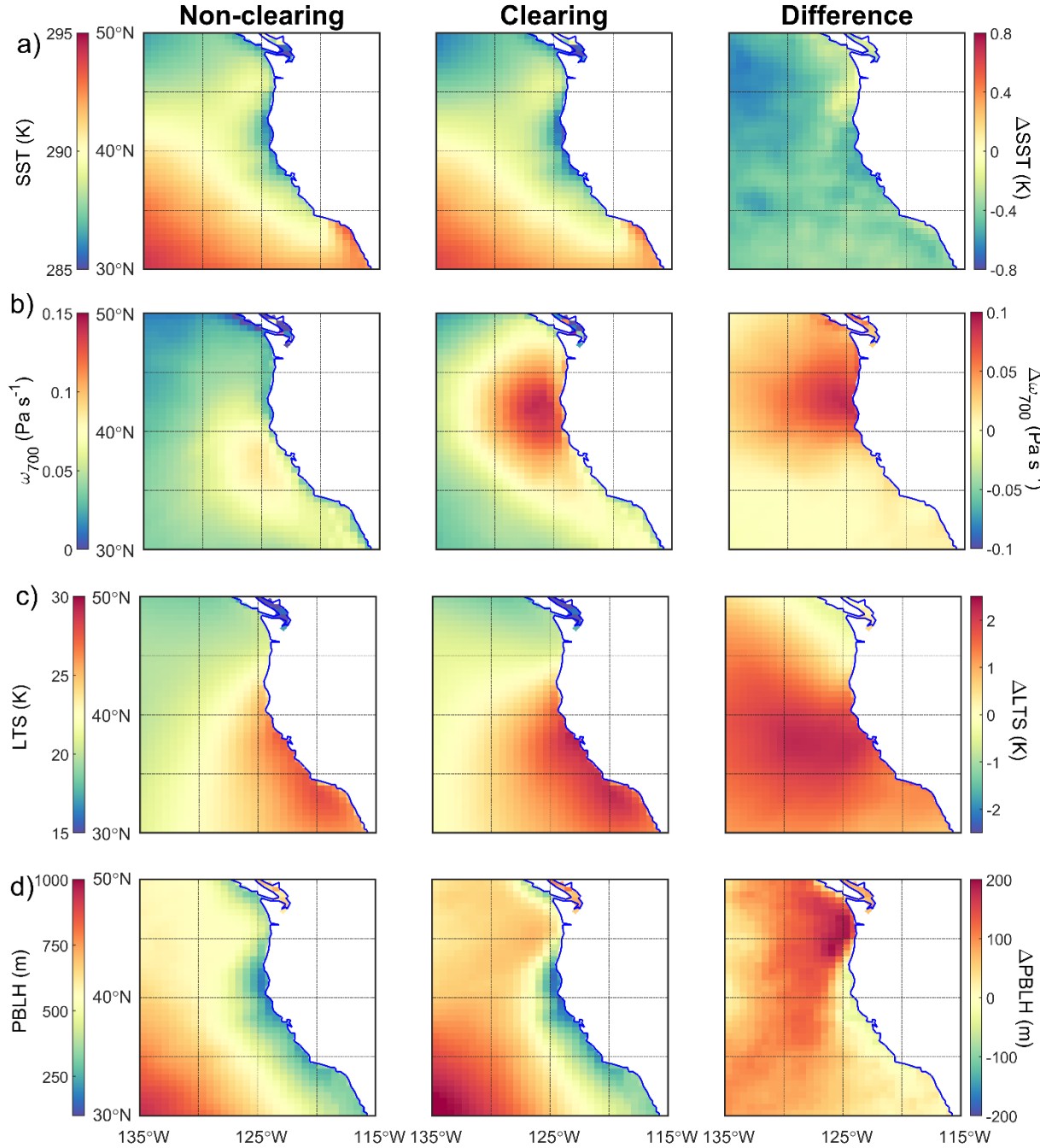

1484
**Figure 8.** Spatial map of environmental parameters controlling properties of stratocumulus clouds
for non-clearing and clearing events: a) sea surface temperature (*SST*), b) vertical pressure velocity
at 700 hPa ($\omega_{700}$), c) lower-tropospheric stability (*LTS*), d) planetary boundary layer height
(*PBLH*), e) specific humidity at 10 m ($q_{10m}$), f) specific humidity at 850 hPa ($q_{850}$), and g) aerosol
optical depth (*AOD*). Differences (clearing minus non-clearing) are shown in the farthest right
column with separate color scales.


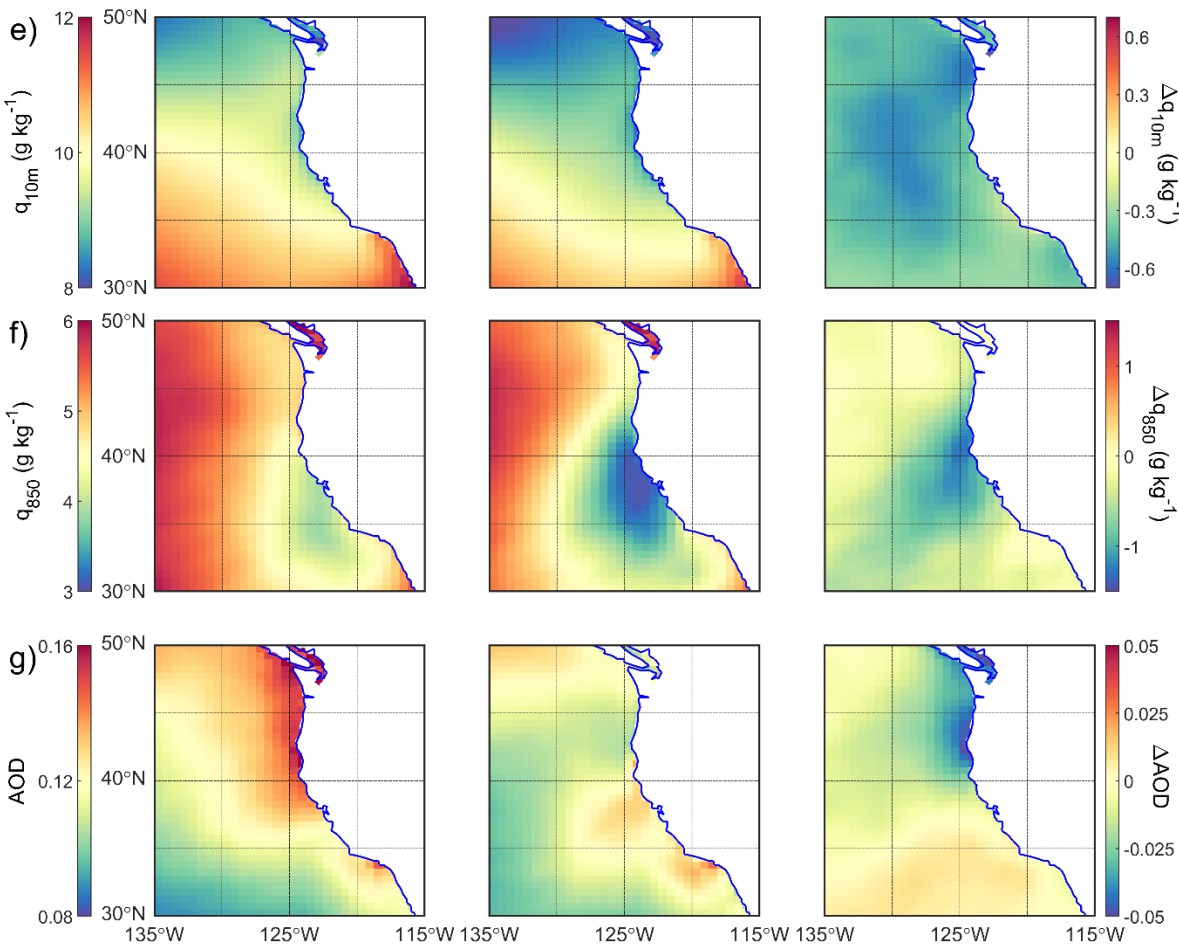

**Figure 8 (continued).**

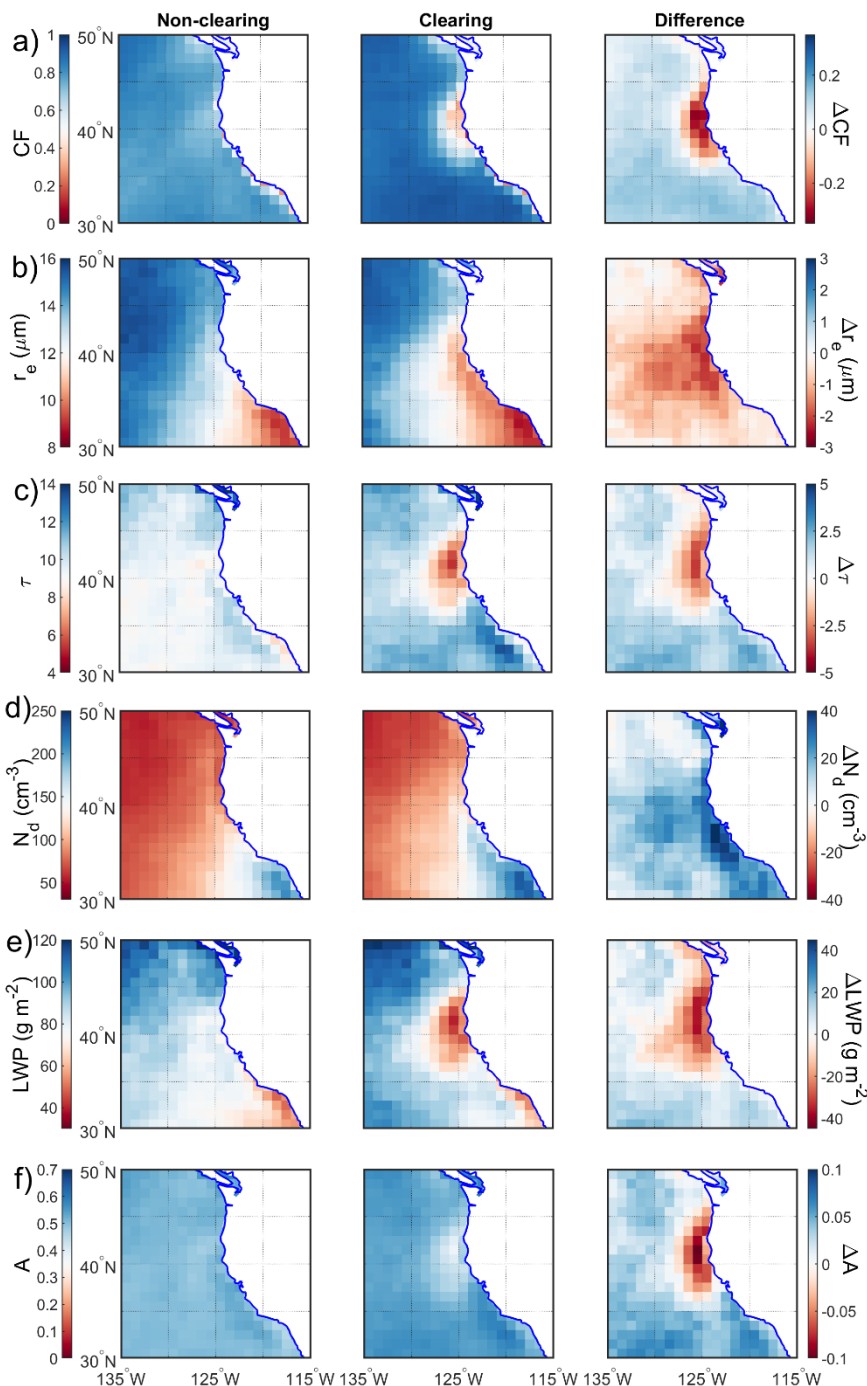


**Figure 9.** Average cloud parameters for non-clearing and clearing days obtained from MODIS Terra Level 3 (Collection 6.1) data: a) cloud fraction day (*CF*), b) cloud top droplet effective radius (*r_e*), c) cloud optical thickness (*τ*), d) cloud droplet number concentration (*N_d*), e) cloud liquid water path (*LWP*), and f) cloud albedo (*A*). Differences (clearing minus non-clearing) are shown in the farthest right column with separate color scales. Values from any instances of clear pixels were omitted from the analysis to produce panels b-f. Fig. S6 is an analogous figure based on MODIS Aqua data.

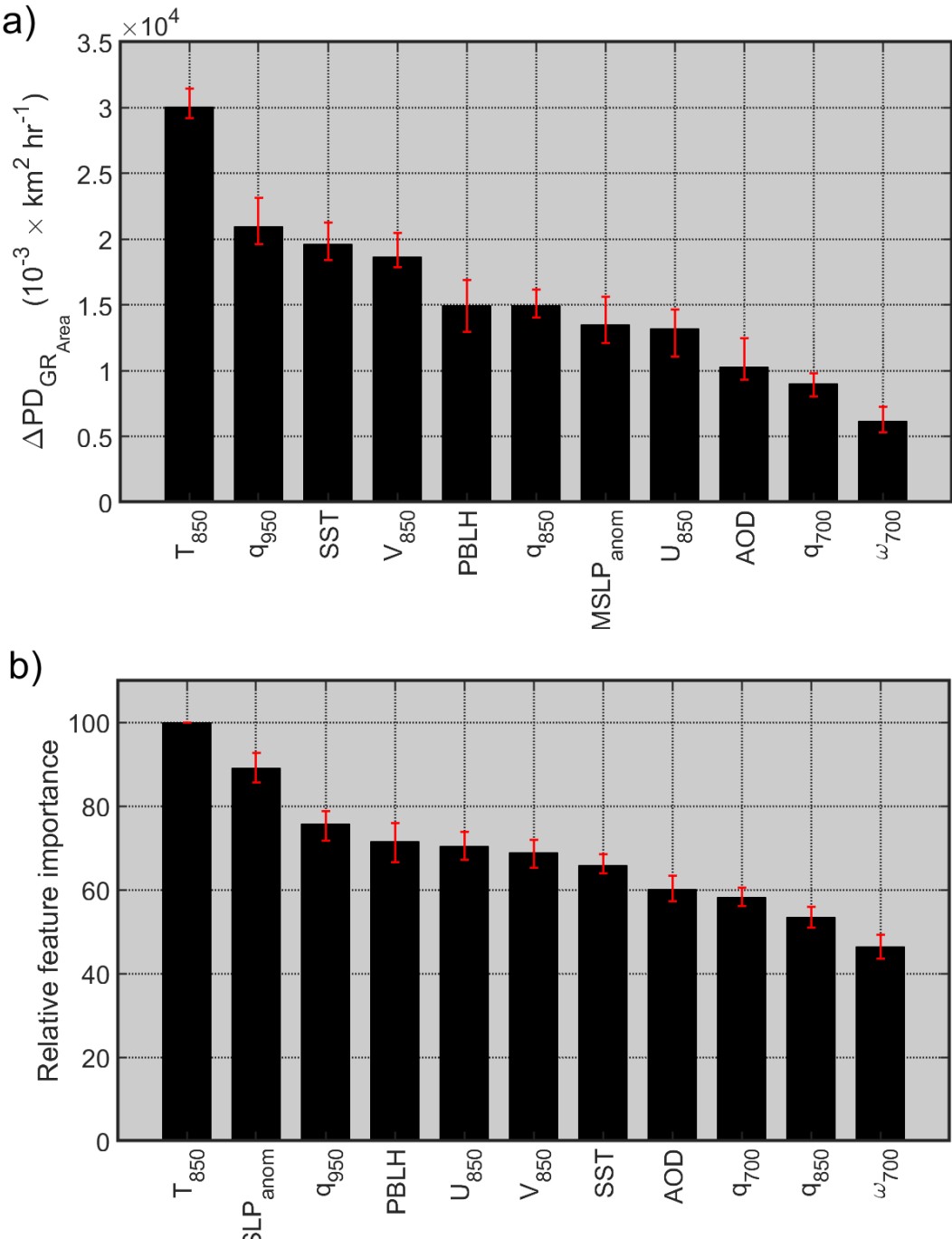

Figure 10. Two scoring methods used for measuring the relative influence of input variables in the GBRT model: a) the median difference of maximum and minimum partial dependence ($PD$) of clearing growth rate ($GR_{Area}$), and b) the median of relative feature importance calculated based on the method developed by Friedman (2001). Error bars represent the range of variability in 30 model runs. Note that GBRT simulations were performed using clearing growth rates obtained from the analysis of first and second GOES images (~09:00 – 12:00 PST) for all 306 clearing events examined.

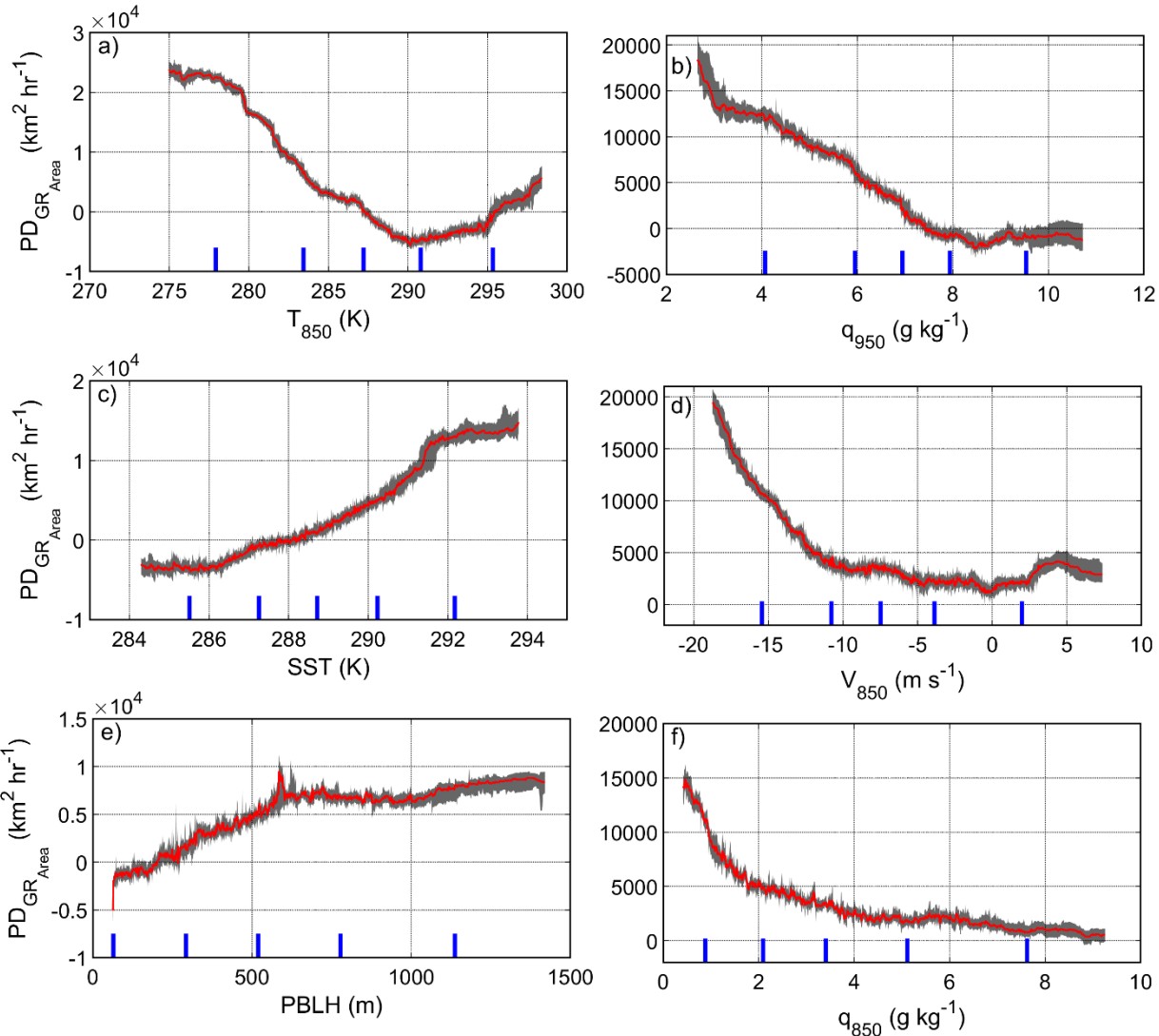

1514

**Figure 11.** The median partial dependence (*PD*) of clearing growth rate (*GR$_{Area}$*) on the following
parameters: a) air temperature at 850 hPa (*T$_{850}$*), b) air specific humidity at 950 hPa (*q$_{950}$*), c) sea
surface temperature (*SST*), d) meridional wind speed at 850 hPa (*V$_{850}$*), e) planetary boundary layer
height (*PBLH*), f) air specific humidity at 850 hPa (*q$_{950}$*), g) mean sea level pressure anomaly
(*MSLP$_{anom}$*), h) zonal wind speed at 850 hPa (*U$_{850}$*), i) aerosol optical depth (*AOD*), j) air specific
humidity at 700 hPa (q$_{700}$), and k) vertical pressure velocity at 700 hPa (*ω$_{700}$*). Grey shaded areas
represent the range of variability of *PD* for 30 model runs. Blue lines represent the values of the
(left to right) 5$^{th}$, 25$^{th}$, 50$^{th}$, 75$^{th}$, and 95$^{th}$ percentiles of the input parameter. GBRT simulations
were performed using clearing growth rates obtained from the analysis of first and second GOES
images (09:00 – 12:00 PST) for all 306 clearing events examined.



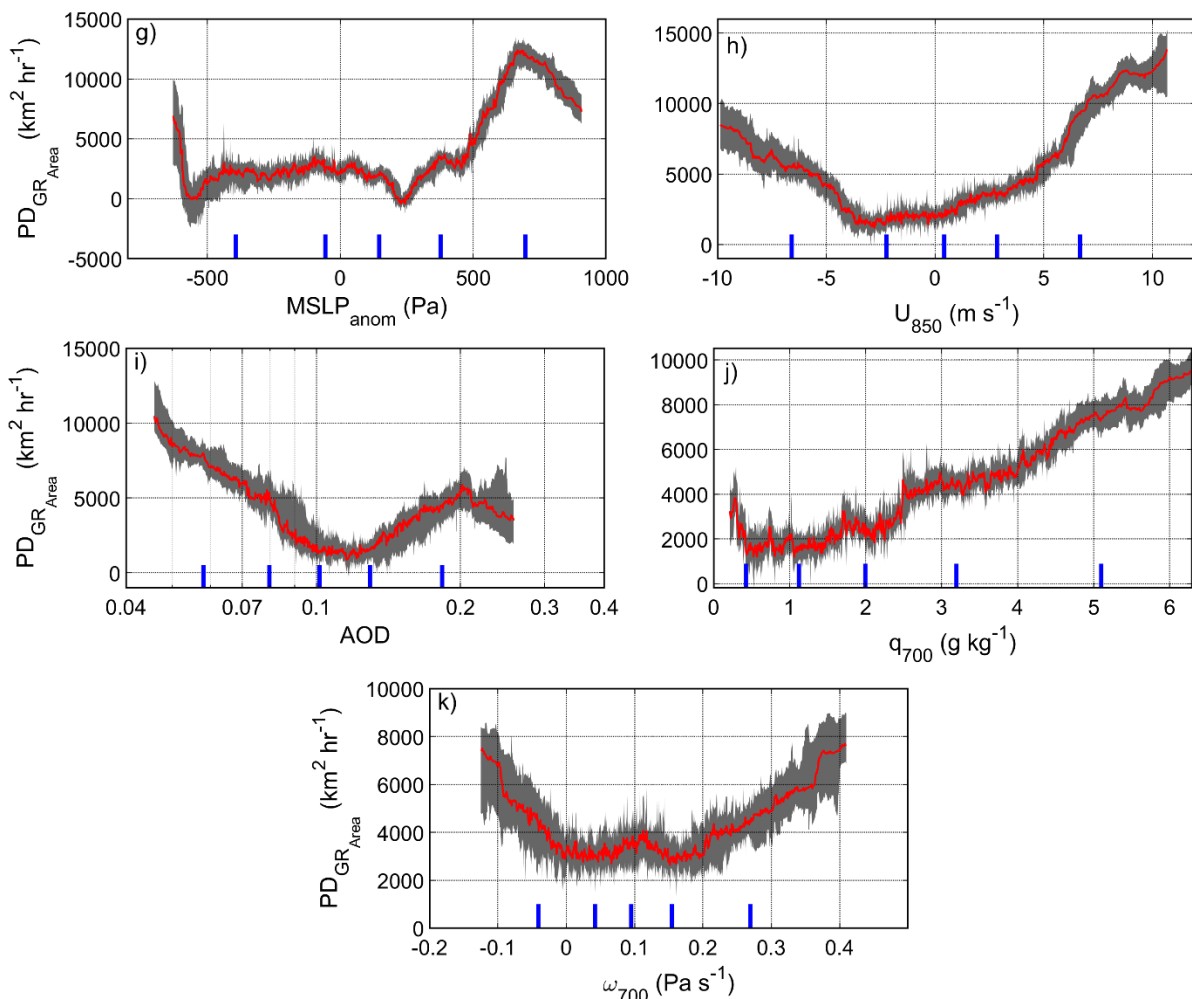

**Figure 11 (continued).**

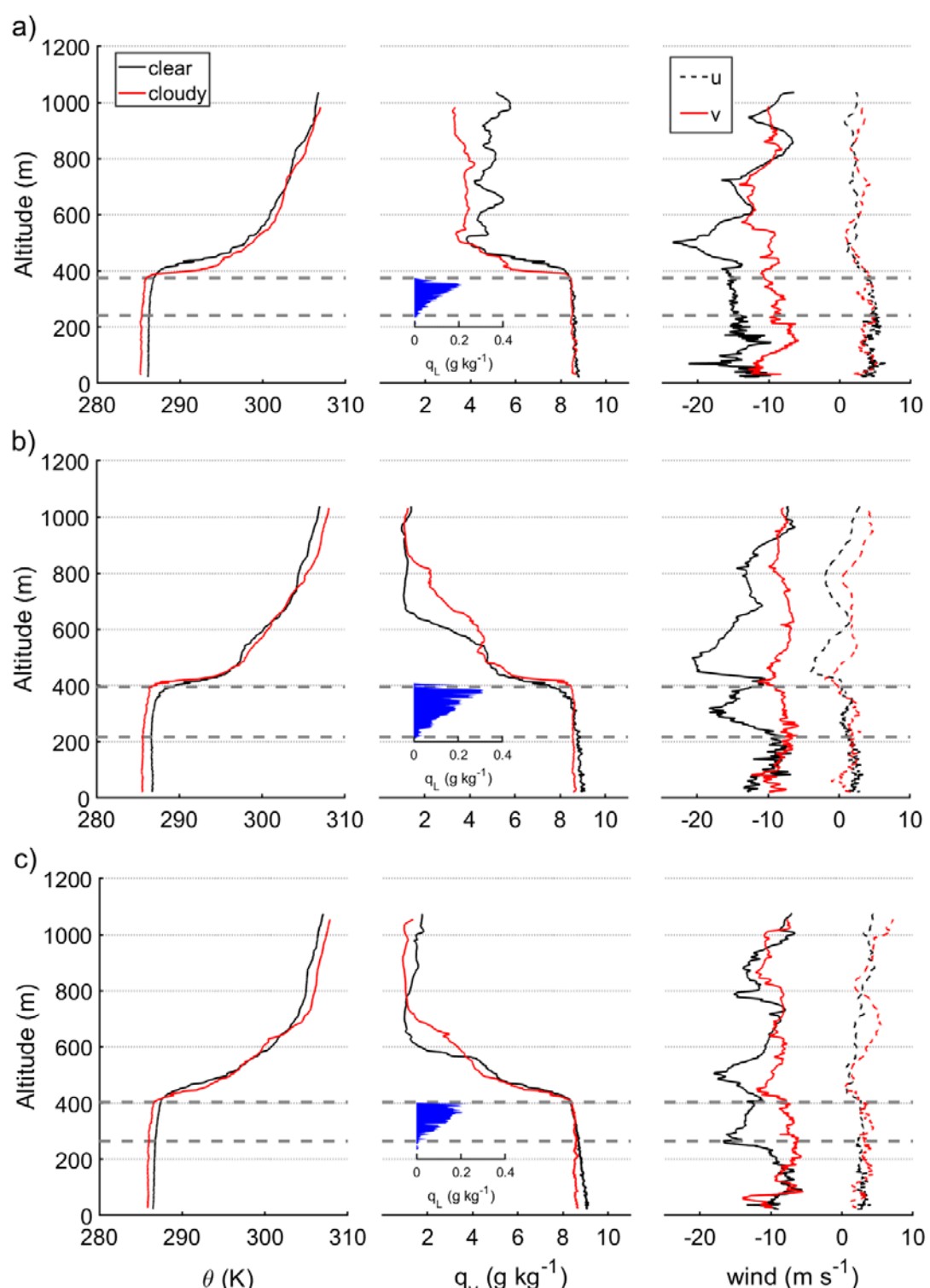

**Figure 12.** Sounding profiles of clear and cloudy columns for three case research flights examined in the FASE campaign: a) RF08, b) RF09A, c) RF09B. Horizontal wind speeds are decomposed into two components, (*u*) perpendicular and (*v*) parallel, relative to the cloud edge. Cloud base and top borders are marked with dashed lines.

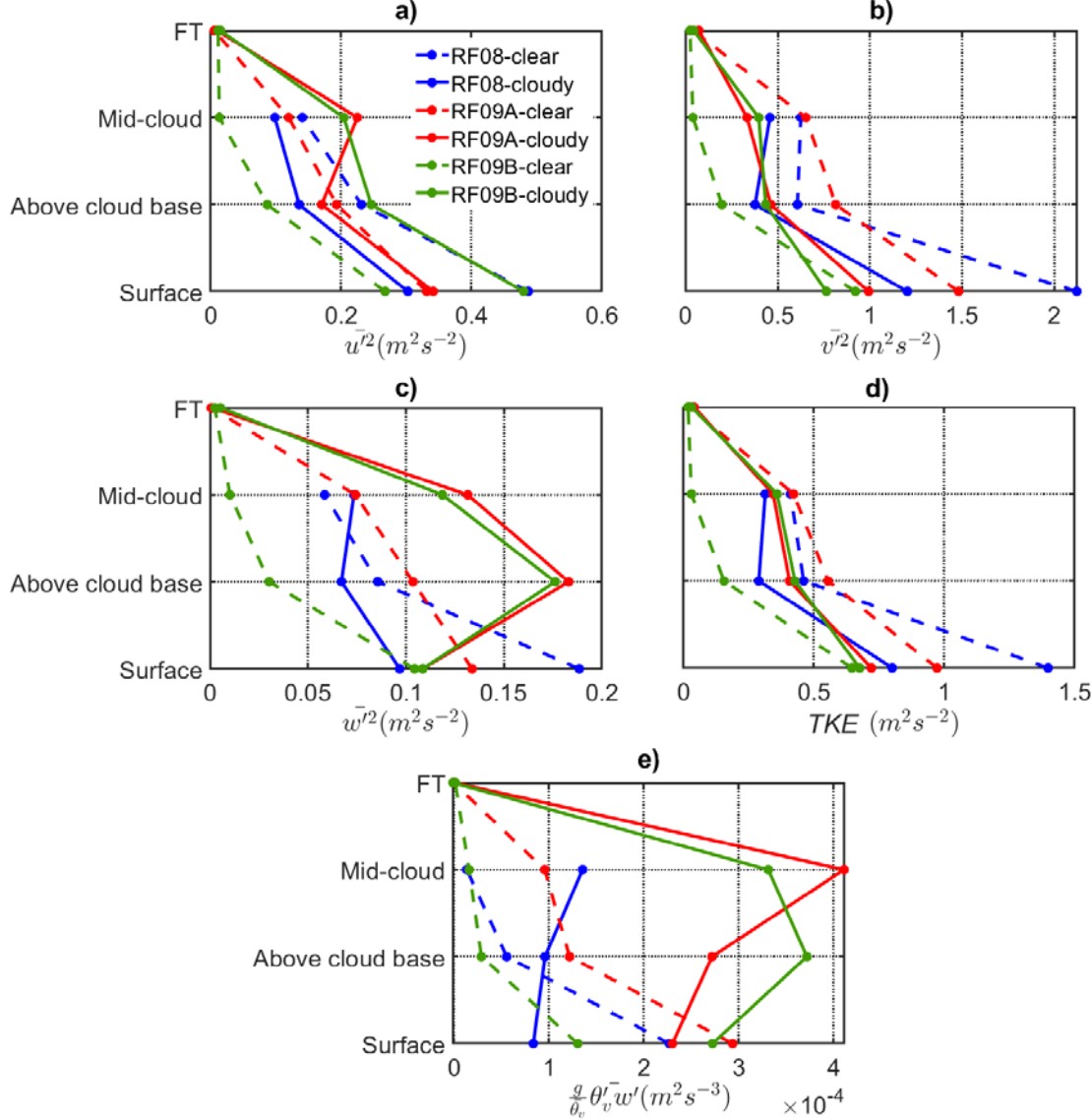

**Figure 13.** Selected dynamic parameters for the clear (dash lines) and cloudy (solid lines) parts of the legs performed at different altitudes for three FASE case research flights: Panels a-c) exhibit squared average velocity fluctuations of wind speeds components (*u* and *v* horizontal components, *w* vertical component). Horizontal wind speeds are decomposed into two components, (*u*) perpendicular and (*v*) parallel, relative to the cloud edge. Panels d) and e) display turbulent kinetic energy and buoyancy flux profiles, respectively, for the three flights.