# Peer review of "Stratocumulus Cloud Clearings: Statistics from Satellites, Reanalysis Models, and Airborne Measurements"

_Atmospheric Chemistry and Physics, 2019_

## Referee Comment (RC1) · Johannes Mohrmann (Referee) · 4 Feb 2020

General comments: This paper presents a new dataset of stratocumulus cloud clearings off the California coast derived from satellite observations, and examines this dataset with a variety of perspectives, including composites of satellite and reanalysis data, aircraft case studies, and a machine learning-based examination of clearing growth rates. The multitude of approaches is thorough and effective at providing a very in-depth characterization of clearing events. The paper is well-written, the text well-supported by the provided figures, and related work is sufficiently cited and referenced.

[Figure]

With regards to interpretation, there are a few areas where I feel the authors can improve and clarify the message of this paper. The most general is in the interpretation of how the large-scale conditions relate to cloud clearings (mainly sections 3.2, 3.3). For a clearing event to take place and be manually identified as described in section 2.1, two conditions must be met: there must be a cloud deck present, and then there must be a coastal clearing that occurs. In other words, the environment must be initially great for a cloudy MBL, and also eventually (at least coastally) poor for a cloudy MBL. The authors spend much of their interpretation arguing (and convincingly so) why certain factors (e.g. offshore winds) would be detrimental to clouds and result in a clearing, but not much on the first condition. For example, when it comes to interpreting the link between clearing days and enhanced stability (Fig 9b), I would expect that it is not so much that the stability is causing a clearing, but rather the link between strong LTS and cloudiness that allows there to be a cloud deck to erode in the first place. Whether a particular environmental factor is predictive of there being a cloud deck, or predictive of it being eroded, is something that can help understand some of the less explained results in the paper, in particular when comparing clearing vs non-clearing days. An obvious one would be the overall higher cloud fraction on clearing days. Presumably, a day with no stratocumulus deck in which to identify a clearing would be classified as "non-clearing day" (if this is incorrect and non-cloudy days are discarded, this should be clarified in section 2.1), and therefore days in which the large-scale conditions in the NEP were unfavourable for clouds would be mixed together with cases which were very favourable to clouds and no clearing occurred in the 'non-clearing day' category. While it would be sufficient to see this discussed in the interpretation with no additional figures, for their own interest the authors might consider splitting their 'non-clearing days' (of which there are approximately twice as many as clearing days anyways) into two sets, based on some criteria of overall cloudiness, and a three-way comparison between 'overall clear days', 'cloudy days-with clearing' and 'cloudy days-no clearing' might prove more interpretable.

This same point is also relevant for the growth rate discussion. The authors show that

the initial growth rate is strongest. A high growth rate would obviously correlate with a larger final clearing area, and this perspective is taken throughout the discussion of growth rate influences, but also a high growth rate may be associated with initially smaller clearings (this is supposition, though the authors could easily investigate in their dataset by examining whether the fastest growing clearings tended to have smaller-than-average initial sizes). Figures 4a supports this however; the presence of a longer lower tail on 9 a.m. size and absence of a longer upper tail on 12 p.m. size (though the log scale might be overemphasizing this) indicates that small initial clearings and not large final clearings are more likely to be the result of a high growth rate. In this case, it would be equally valid to explain why certain predictors of growth rate might be associated with enhanced nighttime cloudiness (again, such as the 1 parameter, $T_{850}$ or possibly LTS), and therefore a well (re-)formed initial deck that is then subsequently susceptible to breakup. Again, this point can largely be addressed in the discussion of results or by author rebuttal and does not require additional figures.

Specific comments:

Section 2.1, line 119: Can you describe in slightly more detail what was necessary for the visual identification of a clearing event? Approximately how large, how distinct, how much cloud had to be adjacent to the clearing? Were days when the Sc deck was completely detached from the coast or absent considered?

Section 3.2 (Clearing vs Non-Clearing)

The difference in subsidence between clearing and non-clearing days seems stark and geographically well-matched to the clearing locations, and yet it comes out as minimally important in the PD analysis. Is the only effect of subsidence to lead to a drier lower FT and therefore all it its signal is captured in $T_{850}$? The $w_{700}$ discussion seemed very brief.

The difference in AOD (low AOD on clearing days, mainly from 43N and up) may be explainable by the circulations shown in figure 8, with anomalously northerly and westerly flow bringing in relatively cleaner air from the marine midlatitudes. That being said, there is no obvious connection between the AOD and $N_d$ maps (low AOD but high $N_d$ on clearing days, though not collocated) that would suggest that the AOD anomalies are having any significant microphysical effect in terms of increasing available CCN, even north of the clearing region. One remedy would be backtrajectory analysis from the low AOD anomaly region, or else looking at the species of aerosol in MERRA-2 to see whether summertime wildfires (which have a large effect on AOD) are impacting the AOD results. The authors state that this may be left for future work, which I would agree with.

Section 3.3 (Growth Rates): It's not clear to me that the condition of requiring only that $r^2 < 0.5$ is a sufficient independence constraint to allow for accurate interpretation of the PD results. For true independence, the authors could have performed an EOF decomposition of all mentioned variables, including those that would clearly correlate strongly with other variables (e.g. LTS, EIS, which as the authors point out are crucial MBL cloud variables), perform the GBRT regression and PD analysis, and additionally the correlation of leading EOFs with input variables. I admit that this would add a level of interpretation, but it would more effectively deal with the tricky problem that so many of these variables are correlated. As it stands the selection of variables seems a little arbitrary, and it is not clear that the resulting ranking of the variables in Figure 11 is physically meaningful. It might be helpful to see another relative ordering of the importance of these variables in accurately determining the growth rate, such as permutation feature importance. Machine learning results are inherently difficult to interpret and the authors have done a more thorough job than many, but one way to improve robustness of interpretation is using multiple evaluation methods.

One area where I think the authors may have stretched the interpretation past the limits of PD analysis is lines 546-558, for instance with the discussion of MSLP and GR. The problem with using PD and correlated variables is that you risk simulating completely nonphysical states which produce nonsensical results. The high and low tails of the

PD sensitivity to MSLP could be a result of the breaking of assumed independence. This could be ameliorated with the addition of a rug plot/histogram to each Figure 12 subplot, showing some kind of likelihood or frequency of occurrence of that particular state (how often a -500 Pa MSLP anomaly occurred in the region affects the degree to which the interpretation of that portion of the PD plot is nonphysical), or the addition of some ICE (individual conditional expectation) plots, both of which are commonly used to help with the interpretation of PD plots.

Technical corrections/suggestions:

Figure 12 caption (line 1250): grey shaded areas, not red.

Figure 13: It would be helpful to see the inversion levels from Table 3 marked on these plots.

---

## Referee Comment (RC2) · Anonymous Referee #2 · 7 Feb 2020

Review of "Stratocumulus cloud clearing: Statistics from satellites, reanalysis models, and airborne measurements" by Dadashazar et al.

Using several data sources and a machine learning technique, this paper examines the topic of marine boundary layer stratiform cloud clearings over the northeastern Pacific Ocean. The study uses a holistic approach by considering spatial scales ranging from the synpoptic-scale to the microscale. The authors' do a nice job of utilizing satellite retrievals, reanalysis grids, and airborne measurements to highlight the complexity of the problem which involves interactions between the western United States coastline and the marine environment – a region which has historically received much attention

in the literature.

I think that the results stemming from this work are certainly interesting and worthy of publication. Because the authors' cover so many topics, I do have several major comments and many minor comments. The major comments concern one of the techniques used for the MODIS processing in addition to interpretation of some of the results. Overall, I recommend that the paper be accepted for publication once the authors' address my comments.

Major/general comments: 1. I am slightly concerned about the methods used to estimate cloud droplet number concentration, Nd. Because the authors' compare plots of Nd between clearing and non-clearing days, certainly there are differences in cloud base temperature and pressure (as implied by several figures shown in this study) that would affect the adiabatic lapse rate of LWC. Therefore, using an average value of the adiabatic lapse rate of LWC, which is derived from measurements concentrated near the central California coastline (Braun et al., 2018), may not be representative of the much larger domain on which the present study focuses. I recommend that the authors' calculate the adiabatic lapse rate of LWC using the MODIS retrievals of cloud top temperature and pressure. I do not mean to sound nitpicky here, but estimation of Nd already carries relatively large uncertainty, so I think that it is only fair that you estimate it as accurately as possible. It will be interesting to see how sensitive the Nd estimate is to this lapse rate calculation.

2. I think that the arguments presented in Section 3.2 regarding the spatial differences in PBLH (P11, L420-425) require additional explanation. Firstly, citations are needed to support the presented hypotheses. More importantly, why do you think that CF is higher for the broad study region on clearing days? What about the synoptic scale scenarios and the role of offshore flow? Advection of warm air combined with compressional warming near the coastline will increase layer thickness and therefore thin out the MBL below. This seems like a chicken-egg problem. Is it actually cloud processes that are responsible for the shallower PBLHs or are the large-scale dynamics/thermodynamics reducing clouds and therefore causing the shallower PBLHs or perhaps some combination of the two mechanisms?

3. The discussion in Section 3.2 connecting the MERRA-2 and MODIS results raises numerous questions that the authors' should address. For example, on P11, L447-448: This is an interesting yet surprising result. I am wondering how aerosol are treated in MERRA-2. Which aerosol types are included in the reanalysis? Is AOD calculated differently when clouds are present in a column? I must say that I am quite surprised that between clearing and non-clearing days, the MODIS retrievals show a clear difference in microphysical variables suggestive of aerosol influence, but MERRA-2 AOD does not show a clear deference in aerosol loading. While the authors' do provide a possible explanation for this confounding result, I am wondering if it is possible to look at precipitation rates from the MERRA-2 outputs? Or use the MODIS retrievals and the RCB-LWP-Nd relationship derived in Comstock et al. (2004) to estimate cloud base precipitation rate? I think that some general investigative work here would be nice to help shed light.

Reference: Comstock, K.K., Wood, R., Yuter, S.E. and Bretherton, C.S. (2004), Reflectivity and rain rate in and below drizzling stratocumulus. Q.J.R. Meteorol. Soc., 130: 2891-2918. doi:10.1256/qj.03.187

Minor/specific comments: 1. P2, L41: Do you mean model simulations from this study or previous studies? Please clarify.

2. P3, L54-56: This statement deserves citations; please cite some papers here.

3. P3, L85-86: Introduce abbreviations for cloud fraction and cloud liquid water path here?

4. P4, L110-112: Are there differences in retrieval and/or post-processing techniques between GOES-11 and GOES-15 that could impact interpretation/comparison of their results?

5. P4, 119-121: Please explain how you identified a clearing event using visual inspection.

6. P5, L146: From which wavelength retrieval are you using data?

7. P5, L147: Is any day that is not a clearing day lumped in with non-clearing days? Or were some days not considered in the analysis?

8. P5, L148: Why use 1 deg x 1 deg data rather than the higher resolution data that are available? I imagine that the resolution of the GOES data are much higher than 1 deg x 1 deg.

9. P5, L150-153: Why are all of these cloud microphysical properties important in the context of cloud clearings? Some justification in this section would be nice.

10. P5, L151-153, L156: Please italicize variables here and throughout the remaining text.

11. P5, L167-170: Does this need to be its own paragraph?

12. P5, Section 2.2: Similar to the previous section, it would be nice to hear some justification as to why you choose the listed parameters/vertical levels. Why are these parameters/vertical levels important to the analysis? Were other variables considered and found to be not useful?

13. Figure 2: The gray shading in panels c and d are a bit deceiving. Is the cloud base/top/depth in panel c truly that horizontally homogeneous? Panel d makes it seem as though cloud extends from the surface to 1000 m. I think that I understand what you are trying to show, but perhaps showing it a bit differently would be less confusing.

14. P6-7, L222-234: Please explain how all of these turbulence measurements will aid in understanding the physical mechanism(s) that contribute to cloud clearing processes.

15. P6, L224: Why use a 2-km wide high pass filter? I imagine this is influenced by the

aircraft speed? By the way, what is the typical aircraft speed?

16. P7, L236: Is Fig. 2c supposed to show where the inversion sits?

17. P7, L236-238: Why use temperature rather than potential temperature?

18. P7, L238-240: This sentence is a bit confusing; please reword.

19. P7, L247-248: Please reference the GBRT method for unfamiliar readers.

20. P8, L284: How is this $r^2$ threshold determined? Are the results sensitive to this choice?

21. P8, L298-299: What about the other MERRA-2 variables listed in Table 1 that are not listed here?

22. P9, L322-323: Please reference a figure here.

23. Figure 5: Because this plot is relatively straightforward, and only two sentences are written about it, I think that it makes more sense to add it to Figure 4, which also shows related variables as a function of time.

24. P9, L354-356: What about near Point Conception? Are similar mechanisms responsible for the reduction of CF here?

25. P9, L356-361: Is it possible to plot low-level (maybe 100 m) wind arrows over the CF contours in Fig. 6 to support/refute this hypothesis?

26. P9, L361-363: You mention southerly wind, but what about northerly wind along the coastline, which is much more common. Are expansion fan dynamics still present?

27. Figure 7: In the difference plot in panel a, are there truly no regions where the SLP is lower in clearing cases?

28. P10, L369: How might using nearly 2 times more non-clearing days influence your results?

29. P10, L383: When you reference Fig. 8a, should this instead be a reference to Fig. 8b?

30. P10, L395-396: A few more citations would be nice for a statement that is "well-documented".

31. P11, L411-413: Can you speculate as to why you observe this?

32. P11, L414-415: Why does PBLH exhibit this trend? Is this is a well-known feature of the MBL offshore the western U.S.?

33. P11, L467: Lower LWP values because the clouds are thinner, LWCs are lower, or both?

34. Section 3.3: Generally speaking, how do sample sizes influence the interpretation of these results? Many of the steep slopes shown in Fig. 12 occur at the low or high ends of the parameter spaces which is likely where the fewest number of samples lie. Are the results robust in these areas?

35. P13, L514-516: Are the local changes in slope of the PD-T850 relationship important? For example, from 275 to 280 K, the slope is relatively small, but from 281 to 282 K, the slope is relatively large.

36. P13, L524-534: Please reference the various panels in this section to help the reader.

37. P13, L540-543: Please provide a citation for this phenomenon. An example of previous work in this region may be found in Rahn et al. (2016, Observations of Large Wind Shear above the Marine Boundary Layer near Point Buchon, California, JAS).

38. P14, L557-558: A negative U850 promoting cloud clearing makes sense due to the offshore flow component, but can you hypothesize as to why strong positive U850 values also promote cloud clearing?

39. P14, L566: Might these vertical motions also induce dynamical circulations and

thereby influence shear/turbulence/entrainment processes near cloud top?

40. P15, L592: Specific or relative humidity?

41. P15, L614-627: I like this portion of the analysis, and the topic of horizontal wind shear is one that probably does not receive enough attention. I think that perhaps a line plot showing how the horizontal shear changes with distance for each of the vertical levels may be very useful.

42. P16, L648-650: I do not understand this sentence; please reword.

43. P16, L660: How is the cloud base rain rate determined?

44. P16-17, L677-681: Are you able to hypothesize why, in all three flights, surface PCASP concentrations are higher on the cloudy side even though the surface wind speeds are higher on the clear side? Is it possible that drizzle drops evaporate after the wet scavenging processes and therefore concentrate aerosol near the surface, whereas aerosol are well-mixed in the MBL on the clear side? If available, vertical profiles may help here.

45. P17, L683: Do you mean stronger gradients in horizontal wind speed?

46. P17, L683-685: What about the role of positive (cyclonic) vorticity that is generated by this horizontal shear? Could this influence cloud properties near the cloudy-clear interface?

47. P18, L749-765: I think that in order for the authors' to argue whether buoyancy or shear production of turbulence is more important, they should calculate the terms according to the TKE equation (e.g., see Eq. 5.1a in Stull, An Introduction to Boundary Layer Meteorology, 1988).

48. P18, L754-755: Adding vertical profiles of TKE would be very useful.

49. P18, L759: What do you mean by "stabilizing effect"?

[Figure]

50. P19, L803-805: Can new remote sensing platforms, such as GOES-16/17, help with the diurnal analysis of cloud properties?

Grammatical/wording recommendations: 1. P6, L198: Please change "Of the relevance to this study" to "Of relevance to this study".

2. P7, L254: Please change "or each of the 306 events." to "for each of the 306 days.".

3. P8, L313: Please change "between 2009 and 2018" to "from 2009 through 2018".

4. P10, L366: Please change "Large-scale characteristics of a dynamic and thermodynamic nature were contrasted" to "Large-scale dynamic and thermodynamic characteristics were contrasted".

5. P10, L401: Please change "likely contribute" to "likely contributes".

6. P11, L410: Please change "geographical coincident" to "geographically coincident".

7. P12, L494: Consider changing "GBRT model to model clearing" to "GBRT model to reproduce clearing".

8. P12, L500: Please remove "partial dependence" as this acronym has already been defined.

9. P16, L656: Please change "lesser effect" to "reduced effect".

10. P19, L780-781: Consider changing "clearings visible from space" to "clearings as suggested by satellite retrievals".

11. P19, L782: Please change "centroid of clearings is centered" to "centroid of clearings is located"

12. P19, L808: Please change "sea spray fluxes, which subsequently can impact clouds" to "sea spray fluxes and can subsequently impact clouds".

---

## Author Comment (AC1) · 18 Mar 2020

Author Response to Both Referee Comments:

Response: We thank the two reviewers for thoughtful suggestions and constructive criticism that have helped us improve our manuscript. Below we provide responses to reviewer concerns and suggestions in blue font.

Reviewer 1:
General comments: This paper presents a new dataset of stratocumulus cloud clearings off the California coast derived from satellite observations, and examines this dataset with a variety of perspectives, including composites of satellite and reanalysis data, aircraft case studies, and a machine learning-based examination of clearing growth rates. The multitude of approaches is thorough and effective at providing a very in-depth characterization of clearing events. The paper is well-written, the text well-supported by the provided figures, and related work is sufficiently cited and referenced.

With regards to interpretation, there are a few areas where I feel the authors can improve and clarify the message of this paper. The most general is in the interpretation of how the large-scale conditions relate to cloud clearings (mainly sections 3.2, 3.3). For a clearing event to take place and be manually identified as described in section 2.1, two conditions must be met: there must be a cloud deck present, and then there must be a coastal clearing that occurs. In other words, the environment must be initially great for a cloudy MBL, and also eventually (at least coastally) poor for a cloudy MBL. The authors spend much of their interpretation arguing (and convincingly so) why certain factors (e.g. offshore winds) would be detrimental to clouds and result in a clearing, but not much on the first condition. For example, when it comes to interpreting the link between clearing days and enhanced stability (Fig 9b), I would expect that it is not so much that the stability is causing a clearing, but rather the link between strong LTS and cloudiness that allows there to be a cloud deck to erode in the first place. Whether a particular environmental factor is predictive of there being a cloud deck, or predictive of it being eroded, is something that can help understand some of the less explained results in the paper, in particular when comparing clearing vs non-clearing days. An obvious one would be the overall higher cloud fraction on clearing days. Presumably, a day with no stratocumulus deck in which to identify a clearing would be classified as "non-clearing day" (if this is incorrect and non-cloudy days are discarded, this should be clarified in section 2.1), and therefore days in which the large-scale conditions in the NEP were unfavourable for clouds would be mixed together with cases which were very favourable to clouds and no clearing occurred in the 'non-clearing day' category. While it would be sufficient to see this discussed in the interpretation with no additional figures, for their own interest the authors might consider splitting their 'non-clearing days' (of which there are approximately twice as many as clearing days anyways) into two sets, based on some criteria of overall cloudiness, and a three-way comparison between 'overall clear days', 'cloudy days-with clearing' and 'cloudy days-no clearing' might prove more interpretable.

Response:

According to the reviewer's comment, we decided to split non-clearing events into two sub-categories of clear and non-clear based on an overall cloud fraction threshold 0.5. Based on this

criterion, 529 cases out of total 614 non-clearing days were further classified as cloudy non-clearing cases. As a result, the influence of events with unfavorable large scale conditions for low-level cloud formation are minimized. Then, we constructed the climatology comparisons of important large scale parameters between clearing and non-clearing (cloudy) conditions similar to Fig.7 in the manuscript. The results are shown in the following figure:

[Figure]

Figure: Climatology of non-clearing (cloudy with CF > 0.5 for study region between 135-115° W and 30-50° N) and clearing days as well as their differences (clearing minus non-clearing) during the summers (JJA) between 2009 and 2018 for a) mean sea level pressure (contours in hPa) and air temperature (color map) at sea surface, b) 850 hPa geopotential heights (contours in m) and air temperature (color map), and c) 500 hPa geopotential heights (contours in m) and air temperature (color map). The data were obtained from MERRA-2 reanalysis. Differences (clearing minus non-clearing) are shown in the farthest right column with separate color scales. White areas indicate no data were available.

As it turns out, the general features were preserved after subcategorizing non-clearing events based on cloud fraction. This result convinced us that the general mechanisms including the

displacement/enhancement of the Pacific high associated with clearing events stem from the nature of clearings and not from our analysis method. We have decided to not include this analysis in the manuscript as it might distract the discussion presented in the body of the paper. However, we revised the discussion of Section 3.2 to address reviewer's comment regarding clarifying if certain parameters (like greater *LTS*) are responsible for clearing formation. We refer the reviewer to edits in Section 3.2 for the concern raised in this comment.

This same point is also relevant for the growth rate discussion. The authors show that the initial growth rate is strongest. A high growth rate would obviously correlate with a larger final clearing area, and this perspective is taken throughout the discussion of growth rate influences, but also a high growth rate may be associated with initially smaller clearings (this is supposition, though the authors could easily investigate in their dataset by examining whether the fastest growing clearings tended to have smaller-than-average initial sizes). Figures 4a supports this however; the presence of a longer lower tail on 9 a.m. size and absence of a longer upper tail on 12 p.m. size (though the log scale might be overemphasizing this) indicates that small initial clearings and not large final clearings are more likely to be the result of a high growth rate. In this case, it would be equally valid to explain why certain predictors of growth rate might be associated with enhanced nighttime cloudiness (again, such as the 1 parameter, T850 or possibly LTS), and therefore a well (re-)formed initial deck that is then subsequently susceptible to breakup. Again, this point can largely be addressed in the discussion of results or by author rebuttal and does not require additional figures.

Response:

We addressed this comment by computing the average initial size of clearings (at the time relevant to image 1) which had growing rates (between image 1 and 2) faster than the 95$^{th}$ percentile of all growth rates. This analysis reveals that in fact the average initial size of the aforementioned subset of clearings is 239,100 km$^2$, while the average size of all clearings is 118,150 km$^2$. This suggests that the reviewer's speculation is not the case as the fastest growing clearings did not tend to have smaller than average initial sizes. Thus, we have decided to not change any part of manuscript based on this comment.

Specific comments:
Section 2.1, line 119: Can you describe in slightly more detail what was necessary for the visual identification of a clearing event? Approximately how large, how distinct, how much cloud had to be adjacent to the clearing? Were days when the Sc deck was completely detached from the coast or absent considered?

Response:

We added the following description in Section 2.1 in response to this comment:

(i)     "Each day's sequence of GOES images were visually inspected to identify if a clearing event was present. This involved utilizing the following general guidelines: (i) There had to be sufficient cloud surrounding the clearing area that the clearing's borders could be approximately identified, which excluded cases with highly broken cloud deck; (ii)

Clearings that were not connected to land between 30°-50° N in any of daily images were excluded; (iii) Days with the cloud deck completely detached from the coast between 30°-50° N were not considered; and (iv) Only clearings with a maximum daily area of greater than 15,000 km$^2$ (which translates to a clearing length on the order of 100 km) were considered. Consequently, the statistics presented in Section 3.1.1 represent a lower limit of clearing occurrence in the study region. However, it is expected that the qualitative trends discussed in Section 3.1.1 are representative of clearing behavior in the study region."

Section 3.2 (Clearing vs Non-Clearing)

The difference in subsidence between clearing and non-clearing days seems stark and geographically well-matched to the clearing locations, and yet it comes out as minimally important in the PD analysis. Is the only effect of subsidence to lead to a drier lower FT and therefore all it its signal is captured in T850? The w700 discussion seemed very brief.

Response: The influence of subsidence on clearing growth is further explained in Section 3.3 as follows:

"The relationship between $\omega$ at 700 hPa and $PD_{GRArea}$ is complex. Brueck et al. (2015) suggested that enhanced $\omega_{700}$ promotes cloudiness due to its link to higher $LTS$. Myers and Norris (2013) further showed that stronger subsidence can reduce $CF$ (at fixed inversion strength) by pushing down the top of the MBL, which is also supported by Bretherton et al. (2013). The $PD_{GRArea}$ profile of $\omega_{700}$ exhibited a minimum point near a value of $0 – 0.2$ Pa s$^{-1}$, with increases in $GR_{Area}$ below and above that range. The increase in $PD_{GRArea}$ with $\omega$ values above 0.2 Pa s$^{-1}$ can be attributed to the negative influence of subsidence on lower $CF$ (via pushing down the top of the MBL) as discussed by Myers and Norris (2013). Conversely, the increase in $GR_{Area}$ with decreasing $\omega$ values below 0 Pa s$^{-1}$ can be due to upward motion reducing the strength of the inversion capping the MBL, which is important to sustain the cloud deck. Vertical motions represented by the $\omega_{700}$ parameter could also induce dynamical circulations affecting cloud top processes such as shear and entrainment."

The difference in AOD (low AOD on clearing days, mainly from 43N and up) may be explainable by the circulations shown in figure 8, with anomalously northerly and westerly flow bringing in relatively cleaner air from the marine midlatitudes. That being said, there is no obvious connection between the AOD and Nd maps (low AOD but high Nd on clearing days, though not collocated) that would suggest that the AOD anomalies are having any significant microphysical effect in terms of increasing available CCN, even north of the clearing region. One remedy would be backtrajectory analysis from the low AOD anomaly region, or else looking at the species of aerosol in MERRA-2 to see whether summertime wildfires (which have a large effect on AOD) are impacting the AOD results. The authors state that this may be left for future work, which I would agree with.

Response: This could be subject of future work. Too much for this current paper in our view.

Section 3.3 (Growth Rates): It's not clear to me that the condition of requiring only that r2 < 0.5 is a sufficient independence constraint to allow for accurate interpretation of the PD results. For

true independence, the authors could have performed an EOF decomposition of all mentioned variables, including those that would clearly correlate strongly with other variables (e.g. LTS, EIS, which as the authors point out are crucial MBL cloud variables), perform the GBRT regression and PD analysis, and additionally the correlation of leading EOFs with input variables. I admit that this would add a level of interpretation, but it would more effectively deal with the tricky problem that so many of these variables are correlated. As it stands the selection of variables seems a little arbitrary, and it is not clear that the resulting ranking of the variables in Figure 11 is physically meaningful. It might be helpful to see another relative ordering of the importance of these variables in accurately determining the growth rate, such as permutation feature importance. Machine learning results are inherently difficult to interpret and the authors have done a more thorough job than many, but one way to improve robustness of interpretation is using multiple evaluation methods.

Response:
We revised the test regarding $r^2$ criterion to emphasize that the threshold value of 0.5 is chosen based on trial and error and it will only reduce the negative impact of correlated variables and will not completely remove undesired effects:

[revised manuscript text omitted]

One area where I think the authors may have stretched the interpretation past the limits of PD analysis is lines 546-558, for instance with the discussion of MSLP and GR. The problem with using PD and correlated variables is that you risk simulating completely nonphysical states which produce nonsensical results. The high and low tails of the PD sensitivity to MSLP could be a result of the breaking of assumed independence. This could be ameliorated with the addition of a rug plot/histogram to each Figure 12 subplot, showing some kind of likelihood or frequency of occurrence of that particular state (how often a -500 Pa MSLP anomaly occurred in the region affects the degree to which the interpretation of that portion of the PD plot is nonphysical), or the addition of some ICE (individual conditional expectation) plots, both of which are commonly used to help with the interpretation of PD plots.

Response:
To address this comment, we updated Fig. 11 with the information about the distributions of data by marking the various percentiles (5[th], 25[th], 50[th], 75[th], and 95[th]) of input data as blue markers.

[Figure]

**Figure 11.** The median partial dependence ($PD$) of clearing growth rate ($GR_{Area}$) on the following parameters: a) air temperature at 850 hPa ($T_{850}$), b) air specific humidity at 950 hPa ($q_{950}$), c) sea

surface temperature (*SST*), d) meridional wind speed at 850 hPa ($V_{850}$), e) planetary boundary layer height (*PBLH*), f) air specific humidity at 850 hPa ($q_{950}$), g) mean sea level pressure anomaly (*MSLP$_{anom}$*), h) zonal wind speed at 850 hPa ($U_{850}$), i) aerosol optical depth (*AOD*), j) air specific humidity at 700 hPa ($q_{700}$), and k) vertical pressure velocity at 700 hPa ($\omega_{700}$). Grey Shaded areas represent the range of variability of *PD* for 30 model runs. Blue lines represent the values of the (left to right) 5th, 25th, 50th, 75th, and 95th percentiles of the input parameter. GBRT simulations were performed using clearing growth rates obtained from the analysis of first and second GOES images (09:00 – 12:00 PST) for all 306 clearing events examined.

[Figure]

**Figure 12 (continued).**

Technical corrections/suggestions:
Figure 12 caption (line 1250): grey shaded areas, not red.

Response: Fixed.

Figure 13: It would be helpful to see the inversion levels from Table 3 marked on these plots.

Response: We think adding the inversion heights to Fig. 13 may confuse readers as they can easily find them in Table 3 and their values are different for cloudy and clear columns. Also, readers can spot the base of inversion according to the cloud top marked in Fig. 13.

Reviewer 2:
Review of "Stratocumulus cloud clearing: Statistics from satellites, reanalysis models, and airborne measurements" by Dadashazar et al.

Using several data sources and a machine learning technique, this paper examines the topic of marine boundary layer stratiform cloud clearings over the northeastern Pacific Ocean. The study uses a holistic approach by considering spatial scales ranging from the synpoptic-scale to the microscale. The authors' do a nice job of utilizing satellite retrievals, reanalysis grids, and airborne measurements to highlight the complexity of the problem which involves interactions between the western United States coastline and the marine environment – a region which has historically received much attention in the literature.

I think that the results stemming from this work are certainly interesting and worthy of publication. Because the authors' cover so many topics, I do have several major comments and many minor comments. The major comments concern one of thetechniques used for the MODIS processing in addition to interpretation of some of the results. Overall, I recommend that the paper be accepted for publication once the authors' address my comments.

Major/general comments:
1. I am slightly concerned about the methods used to estimate cloud droplet number concentration, Nd. Because the authors' compare plots of Nd between clearing and non-clearing days, certainly there are differences in cloud base temperature and pressure (as implied by several figures shown in this study) that would affect the adiabatic lapse rate of LWC. Therefore, using an average value of the adiabatic lapse rate of LWC, which is derived from measurements concentrated near the central California coastline (Braun et al., 2018), may not be representative of the much larger domain on which the present study focuses. I recommend that the authors' calculate the adiabatic lapse rate of LWC using the MODIS retrievals of cloud top temperature and pressure. I do not mean to sound nitpicky here, but estimation of Nd already carries relatively large uncertainty, so I think that it is only fair that you estimate it as accurately as possible. It will be interesting to see how sensitive the Nd estimate is to this lapse rate calculation.

Response:

Addressing the reviewer concern about using a constant value for adiabatic ($\Gamma_{ad}$) lapse rate of $LWC$, we recalculated $N_d$ values for both MODIS-Aqua and Terra using $\Gamma_{ad}$ that are dependent of cloud top temperature and pressures. Panel d of Figures 9 and S6 are also updated accordingly. It turns out the above modification had negligible effects on the average spatial distribution of $N_d$ over the region of interest on both clearing and non-clearing days. As such, we have decided to not change any discussion regarding Fig. 9 in the manuscript. We have also revised a few lines in Section 2.1 to describe the methodology of estimating $N_d$ from MODIS observations as follow:

"…where $\rho_w$ is the density of liquid water, $\Gamma_{ad}$ is the adiabatic lapse rate of liquid water content (LWC), and the parameter $k$ is representative of droplet spectral shape as the cube of the ratio between the volume mean radius and the effective radius. $\Gamma_{ad}$ is a function of temperature and pressure (Albrecht et al., 1990). In this study, cloud top temperature and pressure, provided by

MODIS, are used to estimate $\Gamma_{ad}$ following the methodology described in Braun et al. (2018). A constant value of 0.8 (Martin et al. 1994) is assigned to $k$ in Equation 1."

[Figure]

**Figure 9.** Average cloud parameters for non-clearing and clearing days obtained from MODIS Terra Level 3 (Collection 6.1) data: a) cloud fraction day (*CF*), b) cloud top droplet effective radius ($r_e$), c) cloud optical thickness ($\tau$), d) cloud droplet number concentration ($N_d$), e) cloud liquid water path (*LWP*), and f) cloud albedo (*A*). Differences (clearing minus non-clearing) are shown in the farthest right column with separate color scales. Values from any instances of clear pixels

were omitted from the analysis to produce panels b-f. Fig. S6 is an analogous figure based on MODIS Aqua data.

[Figure]

**Figure S6.** Average cloud parameters for non-clearing and clearing days obtained from MODIS Aqua Level 3 (Collection 6.1) data: a) cloud fraction day ($CF$), b) cloud top droplet effective radius ($r_e$), c) cloud optical thickness ($\tau$), d) cloud droplet number concentration ($N_d$), e) cloud liquid water path ($LWP$), and f) cloud albedo ($A$). Differences (clearing minus non-clearing) are

shown in the farthest right column with separate color scales. Values from any instances of clear pixels were omitted from the analysis to produce these figures.

2. I think that the arguments presented in Section 3.2 regarding the spatial differences in PBLH (P11, L420-425) require additional explanation. Firstly, citations are needed to support the presented hypotheses. More importantly, why do you think that CF is higher for the broad study region on clearing days? What about the synoptic scale scenarios and the role of offshore flow? Advection of warm air combined with compressional warming near the coastline will increase layer thickness and therefore thin out the MBL below. This seems like a chicken-egg problem. Is it actually cloud processes that are responsible for the shallower PBLHs or are the large-scale dynamics/thermodynamics reducing clouds and therefore causing the shallower PBLHs or perhaps some combination of the two mechanisms?

Response:

We addressed the comment by revising/updating the noted argument presented in Section 3.2 as follows below. We also added new references to support our discussion.

"Another key environmental parameter related to MBL cloud coverage is the *PBLH*. Consistent with previous studies (Neiburger et al., 1961; Wood and Bretherton 2004), regardless of whether clearings were present, *PBLH* generally increases with distance from the coast (Fig. 8d), where warmer *SSTs* lead to deeper MBLs by weakening the inversion (Bretherton and Wyant 1997). The shallowing of the MBL near the California coast is also notable with enhanced gradients in clearing days. The aforementioned MBL shallowing is believed to be a crucial element in development of coastal jet off the California coast (Zemba and Friehe 1987; Parish 2000). Previous studies (Beardsley et al., 1987; Edwards et al., 2001; Parish 2000; Zuidema et al., 2009) also reported MBL height adjustment in the vicinity of coast due to hydraulic adaptation to coastal topography, thermally driven circulation, and geostrophic adjustment in the cross-coast direction in response to the contrast in surface heating between ocean and land. There is also a strong gradient in *PBLH* along the shoreline in the vicinity of Cape Blanco (Fig. 8d). While the presence of a similar gradient in *SST* (Fig. 8a) may partly explain the observed gradient in *PBLH*, coastally induced processes could also play a role.

Comparing clearing with non-clearing days, *PBLH* tends to be higher on clearing days, with the largest differences (~200 m) observed to the north off the coasts of Washington and British Columbia, which re-emphasizes the important role of coastal topography near Cape Blanco and Cape Mendocino in mesoscale dynamics (Beardsley et al., 1987; Haack et al., 2001). Zuidema et al. (2009) suggested that dynamical blocking of the surface winds by the southern Peruvian Andes contributed to boundary layer thickening by encouraging mesoscale convergence. Enhanced dynamical blocking of surface winds by coastal topography near Cape Blanco, as suggested by greater wind speeds on clearing days (Fig. 7a), can lead to a deeper MBL in the coastal regions north and northwest of Cape Blanco. In contrast, coastal areas south of Cape Blanco, exhibit negligible differences in *PBLH* between clearing and non-clearing days. In the aforementioned regions, enhanced hydraulic response (i.e., expansion fan (Parish et al., 2016)) to coastal topography, may cause slightly shallower MBL on clearing days.

Higher MBL depths in the offshore regions of clearing days is noteworthy to discuss. Parameters influencing MBL depth include entrainment rates, vertical velocity at the top of MBL, and horizontal advection of MBL (Wood and Bretherton 2004; Rahn and Garreaud 2010). Although on clearing days there may be greater subsidence rates offshore (Fig. 8c) promoting a shallower MBL, the sum of entrainment and horizontal advection terms counteract the aforementioned effect resulting in a deeper MBL. Wood and Bretherton (2004) showed for the Northeast and Southeast Pacific that entrainment and subsidence were the most influential terms in the MBL prognostic equation, which acted in the opposite manner. It is also likely that entrainment processes resulting from changes in small scale turbulence contributed to elevated *PBLH* on clearing days (Randall 1984, Rahn and Garreaud 2010). The maps of *CF* from MODIS Terra (Fig. 9a) can provide at least one possible explanation for the spatial differences in *PBLH* between clearing and non-clearing days. Cloud fraction is generally higher for the broad study region on clearing days, which leads to more opportunity for cloud top radiative cooling to then fuel turbulence in MBL (Wood 2012). Greater turbulence can lead to a deeper MBL by promoting greater entrainment at the top of MBL (Randall 1984; Wood 2007)."

3. The discussion in Section 3.2 connecting the MERRA-2 and MODIS results raises numerous questions that the authors' should address. For example, on P11, L447-448: This is an interesting yet surprising result. I am wondering how aerosol are treated in MERRA-2. Which aerosol types are included in the reanalysis? Is AOD calculated differently when clouds are present in a column? I must say that I am quite surprised that between clearing and non-clearing days, the MODIS retrievals show a clear difference in microphysical variables suggestive of aerosol influence, but MERRA-2 AOD does not show a clear deference in aerosol loading. While the authors' do provide a possible explanation for this confounding result, I am wondering if it is possible to look at precipitation rates from the MERRA-2 outputs? Or use the MODIS retrievals and the RCB-LWP-Nd relationship derived in Comstock et al. (2004) to estimate cloud base precipitation rate? I think that some general investigative work here would be nice to help shed light.

Reference: Comstock, K.K., Wood, R., Yuter, S.E. and Bretherton, C.S. (2004), Reflectivity and rain rate in and below drizzling stratocumulus. Q.J.R. Meteorol. Soc., 130: 2891-2918. doi:10.1256/qj.03.187

Response: This is an excellent point and gets a bit more into the weeds of the critical details of how MERRA-2 and MODIS compare. We share here a bit more about MERRA-2:

The MERRA-2 aerosol reanalysis (Buchard et al., 2017; Randles et al., 2017) relies on the GEOS-5 Goddard Aerosol Assimilation System (Buchard et al., 2015) where the Goddard Chemistry, Aerosol, Radiation, and Transport (GOCART) (Chin et al., 2002) model is used to simulate 15 externally mixed aerosol tracers including hydrophobic and hydrophilic black carbon and organic carbon, dust (five size bins), sea salt (five size bins), and $SO_4^{-2}$. Sea salt and dust emissions are driven by wind speed in the GOCART model. Other species are treated using various emissions from combustion, biomass burning, biogenic sources, and volcanic emissions. The dominant removal mechanisms for aerosols include gravitational settling, dry deposition, and wet scavenging. MERRA-2 assimilates AOD from ground and satellite-based remote sensors, including AVHRR, AERONET, MISR, and MODIS.

Buchard, V., da Silva, A. M., Colarco, P. R., Darmenov, A., Randles, C. A., Govindaraju, R., . . . Spurr, R. (2015). Using the OMI Aerosol Index and Absorption Aerosol Optical Depth to Evaluate the NASA MERRA Aerosol Reanalysis. Atmospheric Chemistry and Physics, 15(10), 5743-5760. doi:10.5194/acp-15-5743-2015

Buchard, V., Randles, C. A., da Silva, A. M., Darmenov, A., Colarco, P. R., Govindaraju, R., Ferrare, R., Hair, J., Beyersdorf, A. J., Ziemba, L. D., and Yu, H.: The MERRA-2 Aerosol Reanalysis, 1980 Onward. Part II: Evaluation and Case Studies, J Climate, 30, 6851-6872, 10.1175/Jcli-D-16-0613.1, 2017.

Chin, M., Ginoux, P., Kinne, S., Torres, O., Holben, B. N., Duncan, B. N., Martin, R. V., Logan, J. A., Higurashi, A., and Nakajima, T.: Tropospheric aerosol optical thickness from the GOCART model and comparisons with satellite and Sun photometer measurements, J Atmos Sci, 59, 461-483, Doi 10.1175/1520-0469(2002)059<0461:Taotft>2.0.Co;2, 2002.

Randles, C. A., da Silva, A. M., Buchard, V., Colarco, P. R., Darmenov, A., Govindaraju, R., Smirnov, A., Holben, B., Ferrare, R., Hair, J., Shinozuka, Y., and Flynn, C. J.: The MERRA-2 Aerosol Reanalysis, 1980 Onward. Part I: System Description and Data Assimilation Evaluation, J Climate, 30, 6823-6850, 10.1175/Jcli-D-16-0609.1, 2017.

We added the following text to the paper based on the lengthier description above, which we feel is adequate to articulate how MERRA-2 handles aerosols:

"Of note is that the MERRA-2 aerosol reanalysis relies on the GEOS-5 Goddard Aerosol Assimilation System (Buchard et al., 2015) for which the Goddard Chemistry, Aerosol, Radiation, and Transport (GOCART) model (Chin et al., 2002) simulates 15 externally mixed aerosol tracers including sulfate, dust (five size bins), sea salt (five size bins), and hydrophobic and hydrophilic black carbon and organic carbon. Of relevance to this study, GOCART applies wind-speed dependent emissions for sea salt. Furthermore, the dominant removal mechanisms for aerosols include gravitational settling, dry deposition, and wet scavenging."

Also, we feel as though deeper examination into precipitation rates is best left for future work. We did not want to get too deep in this current manuscript into the aerosol-related aspects but believe that there are enough compelling results to investigate the aerosol-related aspects in subsequent work.

Minor/specific comments:

1. P2, L41: Do you mean model simulations from this study or previous studies? Please clarify.

Response: This study. Revised sentence:

"Measurements were compared on both sides of the clear-cloudy border of clearings at multiple altitudes in the boundary layer and free troposphere, with results helping to support links suggested by this study's model simulations."

2. P3, L54-56: This statement deserves citations; please cite some papers here.

Response: Added:

"Stratocumulus clouds also play an important role in the global radiation budget due to their high albedo contrast with the underlying ocean surface (Hartmann and Short, 1980; Herman et al., 1980; Stephens and Greenwald, 1991)."

Hartmann, D. L., and Short, D. A.: On the Use of Earth Radiation Budget Statistics for Studies of Clouds and Climate, J Atmos Sci, 37, 1233-1250, Doi 10.1175/1520-0469(1980)037<1233:Otuoer>2.0.Co;2, 1980.

Herman, G. F., Wu, M. L. C., and Johnson, W. T.: The Effect of Clouds on the Earths Solar and Infrared Radiation Budgets, J Atmos Sci, 37, 1251-1261, Doi 10.1175/1520-0469(1980)037<1251:Teocot>2.0.Co;2, 1980.

Stephens, G. L., and Greenwald, T. J.: The Earths Radiation Budget and Its Relation to Atmospheric Hydrology .2. Observations of Cloud Effects, J Geophys Res-Atmos, 96, 15325-15340, Doi 10.1029/91jd00972, 1991.

3. P3, L85-86: Introduce abbreviations for cloud fraction and cloud liquid water path here?

Response: Done

4. P4, L110-112: Are there differences in retrieval and/or post-processing techniques between GOES-11 and GOES-15 that could impact interpretation/comparison of their results?

Response: Not to our knowledge.

5. P4, 119-121: Please explain how you identified a clearing event using visual inspection.

Response: We added the following description in Section 2.1 in response to this comment:

"Each day's sequence of GOES images were visually inspected to identify if a clearing event was present. This involved utilizing the following general guidelines: (i) There had to be sufficient cloud surrounding the clearing area that the clearing's borders could be approximately identified, which excluded cases with highly broken cloud deck; (ii) Clearings that were not connected to land between 30°-50° N in any of daily images were excluded; (iii) Days with the cloud deck completely detached from the coast between 30°-50° N were not considered; and (iv) Only clearings with a maximum daily area of greater than 15,000 km$^2$ (which translates to a clearing length on the order of 100 km) were considered. Consequently, the statistics presented in Section 3.1.1 represent a lower limit of clearing occurrence in the study region. However, it is expected that the qualitative trends discussed in Section 3.1.1 are representative of clearing behavior in the study region."

6. P5, L146: From which wavelength retrieval are you using data?

Response: We added the requested information:

"The key daytime parameters (Table 1) retrieved for this study relevant to liquid clouds included the following, which were retrieved at 2.1 μm and selected based on their importance for marine boundary layer (MBL) cloud studies: *CF* obtained from the MODIS cloud mask algorithm (Platnick et al., 2003), cloud optical thickness (*τ*), *LWP*, and cloud droplet effective radius (*r_e*). Detailed information about these MODIS products is described elsewhere (Platnick et al., 2003; Platnick et al., 2017; Hubanks et al., 2019)."

7. P5, L147: Is any day that is not a clearing day lumped in with non-clearing days? Or were some days not considered in the analysis?

Response: A non-clearing day is defined as any summer day between 2009 and 2018 which was not identified as a clearing day. We clarify this in the first paragraph of Section 3.2 as follows:

"Large-scale dynamic and thermodynamic characteristics were contrasted (parameters in Table 1) between clearing and non-clearing days (Fig. 6). Sub-daily data were averaged up to daily resolution for parameters of interest, which were subsequently used to produce a climatology for non-clearing (614 days) and clearing (306 days) cases for the summers between 2009 and 2018. It is important to note that non-clearing cases include those summer days (e.g., June, July, and August) from 2009 through 2018 that were not categorized as clearing days. We further calculated the difference between clearing and non-clearing conditions."

8. P5, L148: Why use 1 deg x 1 deg data rather than the higher resolution data that are available? I imagine that the resolution of the GOES data are much higher than 1 deg x 1 deg.

Response: The reviewer is correct. A decision was made early to use the larger resolution data early in the study and the results we feel are robust and informative. Future work by anyone interested can certainly probe similar phenomena at higher resolution, but we decided not to have to re-do the entire analysis for this comment.

9. P5, L150-153: Why are all of these cloud microphysical properties important in the context of cloud clearings? Some justification in this section would be nice.

Response: Text added:

"The key daytime parameters (Table 1) retrieved for this study relevant to liquid clouds included the following, which were selected based on their importance for marine boundary layer (MBL) cloud studies:"

10. P5, L151-153, L156: Please italicize variables here and throughout the remaining text.

Response: Done

11. P5, L167-170: Does this need to be its own paragraph?

Response: We added the paragraph in question to the previous paragraph to address this issue.

12. P5, Section 2.2: Similar to the previous section, it would be nice to hear some justification as to why you choose the listed parameters/vertical levels. Why are these parameters/vertical levels important to the analysis? Were other variables considered and found to be not useful?

Response: We added text to explain our choice of parameters and levels:

"The parameters were chosen based on their ability to provide a sufficient view of atmospheric conditions in which MBL clouds form, evolve, and dissipate. Various vertical levels were used for some MERRA-2 products as a way of obtaining representative information for different layers of the MBL and free troposphere."

13. Figure 2: The gray shading in panels c and d are a bit deceiving. Is the cloud base/top/depth in panel c truly that horizontally homogeneous? Panel d makes it seem as though cloud extends from the surface to 1000 m. I think that I understand what you are trying to show, but perhaps showing it a bit differently would be less confusing.

Response: We typically show our clouds in this manner in past publications. We find our Fig. 2 caption to be sufficiently clear. And we trust that readers know that the gray box in panel c is meant to be representative of where clouds were, and they are not that clear-cut linear at the edges.

14. P6-7, L222-234: Please explain how all of these turbulence measurements will aid in understanding the physical mechanism(s) that contribute to cloud clearing processes.

Response: We discuss the actual results in the Results section and do not think an exhaustive discussion is needed here in the Methods section. Rather, we revise the first sentence of this paragraph:

"Ten Hz measurements of environmental parameters were used to estimate turbulent variance and covariance flux values, which may be relevant to the understanding of clearing formation and evolution based on past work (Crosbie et al., 2016)."

15. P6, L224: Why use a 2-km wide high pass filter? I imagine this is influenced by the aircraft speed? By the way, what is the typical aircraft speed?

Response: Typical aircraft speed is ~55 m s$^{-1}$. We add a line about this now: "The typical aircraft speed was 55 m s$^{-1}$."

We used a 2-km wide high pass filter for detrending signals, which is helpful for flux calculations. It is a common strategy employed in studies of this nature. A 2-km wide high-pass filter is conservatively picked to assure filtering of any signals that does not stem from MBL turbulent eddies (with the typical size being less than MBL depth). Given the aircraft speed of ~55 m s$^{-1}$, a 2-km wide filter translates to a filter with passband frequency of 0.0275 Hz. No change made for this comment.

16. P7, L236: Is Fig. 2c supposed to show where the inversion sits?

Response: The sentence in question says the inversion base typically coincides with cloud top. Thus, Fig. 2c gives a representative view of where the inversion base sits. No change made for this comment.

17. P7, L236-238: Why use temperature rather than potential temperature?

Response: Both could work. We used temperature as has been done in past work. No change made for this comment.

18. P7, L238-240: This sentence is a bit confusing; please reword.

Response: Revised:

"Inversion top was defined as the highest altitude at which $d\theta_l/dz$ exceeded 0.1 K m$^{-1}$, where $\theta_l$ is liquid water potential temperature and $z$ is altitude."

19. P7, L247-248: Please reference the GBRT method for unfamiliar readers.

Response: We already did in the second sentence of the paragraph. But we now added another one to the first sentence if that helps:

"A Gradient Boosted Regression Tree (GBRT) model approach was implemented to investigate the impact of environmental parameters on the evolution of clearing events (Friedman 2001)."

20. P8, L284: How is this r2 threshold determined? Are the results sensitive to this choice?

Response: The $r^2$ threshold was determined by choice. Sensitivity tests were done with different combinations of parameters and the general conclusions were preserved. We updated some text in the manuscript to address this comment:

"While *PD* plots are not flawless in capturing the influence of each variable in the model, especially if the input variables are strongly correlated, they provide useful information for interpretation of GBRT results (Friedman and Meulman 2003; Elith et al., 2008). To decrease the undesired influence of correlated variables on *PD* profiles, an arbitrary $r^2$ threshold of 0.5 was used based on the linear regressions between prospective input parameters. For instance, there were three choices of air temperature (i.e., at 950, 850, and 700 hPa), but based on the $r^2$ criterion, only one ($T_{850}$) was used in the model to minimize the unwanted impact of dependent input parameters. Lower tropospheric stability (*LTS*: defined as the difference between the potential temperature of the free troposphere (700 hPa) and the surface) is the stability parameter that has been widely used as a key factor controlling the coverage of stratocumulus clouds. However, in this study, the effects of stability were examined by putting $T_{850}$ and *SST* into the model without explicitly including *LTS*. The correlation between *LTS* and $T_{850}$ prevented them to be used as input parameters simultaneously. Using $T_{850}$ and *SST* instead of *LTS* is advantageous because the results

can be more informative by revealing different impacts of the two individual parameters on the model's output rather than just one parameter in the form of *LTS*. In addition, the mean sea level pressure anomaly ($MSLP_{anom}$) was used as an input parameter, which was calculated in reference to the average values of *MSLP* for the summer months for the study period. In the end, the following 11 predicting variables from MERRA-2 were used as input parameters for the GBRT simulations, with data product details summarized in Table 1: *AOD*, $T_{850}$, $q_{950}$, $q_{850}$, $q_{700}$, *SST*, $MSLP_{anom}$, $U_{850}$, $V_{850}$, *PBLH*, and $\omega_{700}$. It is important to note that the results of extensive sensitivity tests led to the selection of the set of parameters presented in this study. Also, theses sensitivity tests confirmed that the general conclusions presented here were preserved regardless of using different sets of the input parameters."

21. P8, L298-299: What about the other MERRA-2 variables listed in Table 1 that are not listed here?

Response: Well, they are listed still in Table 1 for completeness to walk readers through our process of analysis to reach the point of Lines 298-299. No harm in doing that in our opinion.

22. P9, L322-323: Please reference a figure here.

Response: Done. Additionally, we now differentiate between Figure 3a and 3b in the text.

23. Figure 5: Because this plot is relatively straightforward, and only two sentences are written about it, I think that it makes more sense to add it to Figure 4, which also shows related variables as a function of time.

Response: We added Figure 5 to the Supplement as adding it to Fig. 4 made this figure hard to read.

24. P9, L354-356: What about near Point Conception? Are similar mechanisms responsible for the reduction of CF here?

Response: We added the following text and added that point to Figure 6a:

"Less pronounced is a centroid of reduced cloud fraction by Point Conception, where similar mechanisms may be at work."

25. P9, L356-361: Is it possible to plot low-level (maybe 100 m) wind arrows over the CF contours in Fig. 6 to support/refute this hypothesis?

Response: Figure 8 shows winds clearly and it would be redundant in our view to put them in Figure 6 too.

26. P9, L361-363: You mention southerly wind, but what about northerly wind along the coastline, which is much more common. Are expansion fan dynamics still present?

Response: The sentences are revised accordingly:

"The significance of these capes is discussed in many previous studies (Beardsley et al., 1987; Haack et al., 2001; Juliano et al., 2019a/b) pointing their ability to alter local dynamics, cloud depth, and various microphysical processes such as entrainment. Cloud thinning in the vicinity of the capes due to an expansion fan effect is reported for both northerly and southerly flow (Beardsley et al., 1987; Juliano et al., 2017)."

27. Figure 7: In the difference plot in panel a, are there truly no regions where the SLP is lower in clearing cases?

Response: This occurred because of the choice of spacing in the contour plot. Figure 7 has been updated to fix this issue.

28. P10, L369: How might using nearly 2 times more non-clearing days influence your results?

Response: It obviously provides more statistics and solidifies the non-clearing results. We do not expect this difference in days to affect the general conclusions.

29. P10, L383: When you reference Fig. 8a, should this instead be a reference to Fig. 8b?

Response: Correct, thanks. Change made.

30. P10, L395-396: A few more citations would be nice for a statement that is "well documented".

Response: Sentence revised as the "well-documented" is unnecessary and seems to be a distraction.

31. P11, L411-413: Can you speculate as to why you observe this?

Response: This paragraph is revised to provide a potential explanation for the observed trend:

"The changes in synoptic-scale conditions, including relocation/strengthening of the Pacific high, on clearing days in comparison to non-clearing days can alter large-scale subsidence. This is indeed confirmed in Fig. 8b using $\omega_{700}$ as the proxy variable, with the strongest difference between clearing and non-clearing days (up to ~ 0.1 Pa s$^{-1}$) off the coast by Cape Blanco and Cape Mendocino and geographically coincident with where the sharpest gradients occur for *MSLP* between clearing and non-clearing cases (Fig. 6). It is interesting to note that the maximum *LTS* values coincide spatially with enhanced values of $\omega_{700}$ on non-clearing days, in contrast to clearing days when the peak value of $\omega_{700}$ is farther north from where *LTS* peaks (Fig. 8c). Consistent with the results presented here (Fig. 8b), modeling studies (Burk and Thompson 1996; Munoz and Garreaud 2005) reported enhanced subsidence for the entrance regions of the Chilean and California CLLJs in response to coastal features. These studies also reported the generation of a warm layer above the MBL due to coastal mechanisms especially downstream of coastal points and capes. This is also the case in this study where higher air temperature at 850 hPa was observed to the south of Cape Blanco and Cape Mendocino on clearing days (Fig. 5b). In addition, higher

*LTS* values on clearing days by up to ~2 K (Fig. 8c) are largely associated with the presence of warmer layer above the MBL south of Cape Blanco and Cape Mendocino. It is likely that reduced *SST*s and greater subsidence contributed to generally higher *LTS* on clearing days versus non-clearing days (Fig. 8c). Other works have pointed to the connection between cooler *SST*s, higher boundary layer cloud amount, and increased stability in the lower atmosphere (Norris and Leovy 1994, Klein and Hartman 1993)."

32. P11, L414-415: Why does PBLH exhibit this trend? Is this is a well-known feature of the MBL offshore the western U.S.?

Response:  We edited this line to add a few references and to address the suspected reason for the observed trend in PBLH:

"Another key environmental parameter related to MBL cloud coverage is the *PBLH*. Consistent with previous studies (Neiburger et al., 1961; Wood and Bretherton 2004), regardless of whether clearings were present, *PBLH* generally increases with distance from the coast (Fig. 8d), where warmer *SST*s lead to deeper MBLs by weakening the inversion (Bretherton and Wyant 1997). The shallowing of the MBL near the California coast is also notable with enhanced gradients in clearing days. The aforementioned MBL shallowing is believed to be a crucial element in development of coastal jet off the California coast (Zemba and Friehe 1987; Parish 2000). Previous studies (Beardsley et al., 1987; Edwards et al., 2001; Parish 2000; Zuidema et al., 2009) also reported MBL height adjustment in the vicinity of coast due to hydraulic adaptation to coastal topography, thermally driven circulation, and geostrophic adjustment in the cross-coast direction in response to the contrast in surface heating between ocean and land. There is also a strong gradient in *PBLH* along the shoreline in the vicinity of Cape Blanco (Fig. 8d). While the presence of a similar gradient in *SST* (Fig. 8a) may partly explain the observed gradient in *PBLH*, coastally induced processes could also play a role."

33. P11, L467: Lower LWP values because the clouds are thinner, LWCs are lower, or  both?

Response: Presumably both. No change needed to text in our view.

34. Section 3.3: Generally speaking, how do sample sizes influence the interpretation of these results? Many of the steep slopes shown in Fig. 12 occur at the low or high ends of the parameter spaces which is likely where the fewest number of samples lie. Are the results robust in these areas?

Response: Added the following text:

"Note that the 5[th], 25[th], 50[th], 75[th], and 95[th] percentiles of input parameter values are denoted in Figure 12 to caution that sharp slopes in the bottom and top 5[th] percentiles are based on few data points and that robust conclusions should not stem from those outer bounds."

35. P13, L514-516: Are the local changes in slope of the PD-T850 relationship important? For example, from 275 to 280 K, the slope is relatively small, but from 281 to 282 K, the slope is relatively large.

Response: The best we thought to do was report our method of how these plots were generated and then leave it to readers to conclude using their own criteria how important local changes are. In our view, we are most interested in more macroscopic trends in these plots and also changes in signs of relationships.

36. P13, L524-534: Please reference the various panels in this section to help the reader.

Response: Done.

37. P13, L540-543: Please provide a citation for this phenomenon. An example of previous work in this region may be found in Rahn et al. (2016, Observations of LargeWind Shear above the Marine Boundary Layer near Point Buchon, California, JAS).

Response: Done:

"Stronger northerly flow is associated with offshore flow of dry and warm air that can reside above the cloud top, which can dissipate the cloud layer after entrainment and via enhanced shearing (via Kelvin-Helmholtz instability) and mixing of cloudy parcels with warm and dry air in the FT (e.g., Rahn et al., 2016). As will be shown later, aircraft data showed that typical wind speeds parallel to clear-cloudy interfaces were near or greater than 10 m s$^{-1}$ (Fig. 12)."

38. P14, L557-558: A negative U850 promoting cloud clearing makes sense due to the offshore flow component, but can you hypothesize as to why strong positive U850 values also promote cloud clearing?

Response: We are not sure and make this explicit:

"Clearing growth due to negative zonal winds can be explained by the offshore flow component, however, the reason for growth during periods of positive zonal winds is unclear."

39. P14, L566: Might these vertical motions also induce dynamical circulations and thereby influence shear/turbulence/entrainment processes near cloud top?

Response: Maybe so. We add text to give this idea some attention in the draft:

"Vertical motions represented by the $\omega_{700}$ parameter could also induce dynamical circulations affecting cloud top processes such as shear and entrainment."

40. P15, L592: Specific or relative humidity?

Response: Made it clear it is "specific".

41. P15, L614-627: I like this portion of the analysis, and the topic of horizontal wind shear is one that probably does not receive enough attention. I think that perhaps a line plot showing how the horizontal shear changes with distance for each of the vertical levels may be very useful.

Response:
We calculated horizontal shear for constant level legs and displayed it in SI file. Additionally, the paragraph has been updated accordingly:

"To extend upon the possibility of shearing effects, absolute changes in $v$ ($|v|$) were calculated for level legs performed at the clear-cloudy border for the three research flights (Table 2). For consistency, these calculations were based on level legs of a constant length of ~40 km with relatively equal spacing on both sides of the clear-cloudy border. $|v|$ was calculated by multiplying 40 km by the slope of the linear fit of $v$ versus distance from cloud edge, where negative (positive) x values represent distance away from the edge on the clear (cloud) side. The results reveal that the horizontal wind shear was strongest somewhere between mid-cloud and cloud top altitudes, with the lowest values at the FT level. The lowest values in the MBL were observed in the surface legs. This can be attributed to turbulent transport of the momentum (Zemba and Friehe 1987) to the surface and the consequent drop in CLLJ wind speeds in the clear column. In addition, Fig. S7 shows absolute horizontal shear ($|dv/dx|$) as a function of distance from the cloud boundary for the parallel component of horizontal wind speed. Horizontal shear profiles for all research flights (Fig. S7) are slightly noisy especially at the surface legs, but they show the presence of the greatest horizontal wind gradient within 5 km length away from clear-cloudy edge. Shear at the clear-cloudy edge, especially at cloud levels, can support clearing growth through enhancing the mixing of cloudy and clear air. Crosbie et al. (2016) also showed using the case of NiCE RF19 that that mixing of cloudy air with adjacent clear air can be an important contributor to cloud erosion and thus expansion of clearings. To probe deeper into the clearing cases, the subsequent discussion compares vertically-resolved data on both sides of the clear-cloudy border based on soundings and level legs."

[Figure]

**Figure S7.** Absolute variations in horizontal shear as a function of distance from the cloud boundary for the parallel component of horizontal wind speed for three case research flights: a) RF08, b) RF09A, and c) RF09B. These variations were shown only for constant altitude legs (surface, above cloud base, and mid-cloud legs). Cloudy columns are highlighted in grey.

42. P16, L648-650: I do not understand this sentence; please reword.

Response: We revised the sentence:

"The wind maximum in the clearing also enhanced moisture advection, which counteracted the accumulation of moisture caused by mixing induced by vertical shear."

43. P16, L660: How is the cloud base rain rate determined?

Response: Text added:

"Cloud base rain rate was quantified using the size distributions of drizzle drop ($D_P > 40$ μm) obtained from CIP in the bottom third of clouds along with documented relationships between fall velocity and drop size (Wood 2005b)."

44. P16-17, L677-681: Are you able to hypothesize why, in all three flights, surface PCASP concentrations are higher on the cloudy side even though the surface wind speeds are higher on the clear side? Is it possible that drizzle drops evaporate after the wet scavenging processes and therefore concentrate aerosol near the surface, whereas aerosol are well-mixed in the MBL on the clear side? If available, vertical profiles may help here.

Response: We went ahead and made vertical profiles as shown in the new Figure S8. It is too difficult to reach the speculation above with a high level of confidence provided by the reviewer based on the available dataset in our opinion. Entrainment of free tropospheric aerosol particles is likely a possible explanation too. We added the following text:

"Figure S8 shows vertical profiles of aerosol concentrations on both sides of the clearing border, highlighting differences above cloud top level especially in RF09A and RF09B with higher values in the cloudy column. Higher aerosol concentrations were also observed in the cloud column in the sub-cloud layer even though surface wind speeds were always higher in the clear column for all three flights. Surface winds and thus sea spray production do not exclusively influence the aerosol concentrations. A likely explanation of higher concentrations in the MBL in the cloudy column is that there could be entrainment of more polluted free tropospheric aerosol as has been reported to be a common occurrence during the FASE flights (e.g., Mardi et al., 2019). As also reported during FASE, there can be sub-cloud evaporation of drizzle resulting in droplet residual particles that contribute to the aerosol concentration budget in the cloudy column (Dadashazar et al., 2018)."

[Figure]

**Figure S8.** PCASP profiles obtained from soundings performed in clear and cloudy columns for three case research flights: a) RF08, b) RF09A, and c) RF09B. The altitude range where the cloud deck was present is highlited in grey. PCASP data are unreliable in cloud due to droplet shatter artifacts and thus not shown.

45. P17, L683: Do you mean stronger gradients in horizontal wind speed?

Response: Yes, that is correct. Text revised.

"Stronger horizontal wind speed gradients,…"

46. P17, L683-685: What about the role of positive (cyclonic) vorticity that is generated by this horizontal shear? Could this influence cloud properties near the cloudy-clear interface?

Response: We suppose that is a possibility but we felt it was not necessary to address this in the text to avoid having too many speculations without unambiguous support.

47. P18, L749-765: I think that in order for the authors' to argue whether buoyancy or shear production of turbulence is more important, they should calculate the terms according to the TKE equation (e.g., see Eq. 5.1a in Stull, An Introduction to Boundary Layer Meteorology, 1988).

Response: This was in fact attempted already but the data looked noisy and inconclusive; this is mainly a limitation of the aircraft data. We do not feel this is really necessary to respond to as a result.

48. P18, L754-755: Adding vertical profiles of TKE would be very useful.

Response: The vertical profiles of TKE have been added to Fig. 13. We also updated the text as follows:

"Profiles of $\overline{u'^2}$ and $\overline{v'^2}$ exhibited downward trends with increasing altitude for RF09A and RF09B, in general agreement with the findings for RF08. One contrasting aspect was the comparison of $\overline{v'^2}$ between clear and cloudy columns, which mirrored RF08 during RF09A, while in RF09B, the values of $\overline{v'^2}$ for the clear side were substantially lower. In addition, $\overline{w'^2}$ profiles during RF09A and RF09B are substantially enhanced in the cloudy column as compared to RF08, with maxima in the cloud layer. There is an accompanying increase in the buoyancy flux for these profiles suggestive of a more significant contribution of buoyancy to *TKE* production (Fig. 13e). Although more subtle, $\overline{u'^2}$ values also showed an increase in the cloudy column of RF09A and RF09B relative to the clear column, also supportive of the role of buoyancy in these cases. In addition, *TKE* profiles (Fig. 13d) were largely influenced by variances in the horizontal component of wind speed ($\overline{u'^2}$ and $\overline{v'^2}$) which led to overall greater *TKE* values in the clear column except for RF09B."

[Figure]

**Figure 13.** Selected dynamic parameters for the clear (dash lines) and cloudy (solid lines) parts of the legs performed at different altitudes for three FASE case research flights: Panels a-c) exhibit squared average velocity fluctuations of wind speeds components ($u$ and $v$ horizontal components, w vertical component). Horizontal wind speeds are decomposed into two components, ($u$) perpendicular and ($v$) parallel, relative to the cloud edge. Panels d) and e) display turbulent kinetic energy and buoyancy flux profiles, respectively, for the three flights.

49. P18, L759: What do you mean by "stabilizing effect"?

Response: Those words were removed.

50. P19, L803-805: Can new remote sensing platforms, such as GOES-16/17, help with the diurnal analysis of cloud properties?

Response: Yes. Text added:

"More data such as those provided by GOES platforms can help understand processes occurring at the microscale that scale up to more climatologically relevant scales."

Grammatical/wording recommendations:
1. P6, L198: Please change "Of the relevance to this study" to "Of relevance to this study".

Response: Edited.

2. P7, L254: Please change "or each of the 306 events." to "for each of the 306 days.".

Response: Fixed.

3. P8, L313: Please change "between 2009 and 2018" to "from 2009 through 2018".

Response: Fixed.

4. P10, L366: Please change "Large-scale characteristics of a dynamic and thermodynamic nature were contrasted" to "Large-scale dynamic and thermodynamic characteristics were contrasted".

Response: Edited.

5. P10, L401: Please change "likely contribute" to "likely contributes".

Response: Edited.
6. P11, L410: Please change "geographical coincident" to "geographically coincident".

Response: Edited.

7. P12, L494: Consider changing "GBRT model to model clearing" to "GBRT model to reproduce clearing".

Response: Changed.

8. P12, L500: Please remove "partial dependence" as this acronym has already been defined.

Response: Edited.

9. P16, L656: Please change "lesser effect" to "reduced effect".

Response: Edited.

10. P19, L780-781: Consider changing "clearings visible from space" to "clearings as suggested by satellite retrievals".

Response: Changed.

11. P19, L782: Please change "centroid of clearings is centered" to "centroid of clearings is located"

Response: Edited.

12. P19, L808: Please change "sea spray fluxes, which subsequently can impact clouds" to "sea spray fluxes and can subsequently impact clouds".

Response: Edited.